



**1  Unraveling Pathways of Elevated Ozone Induced by the 2020**

**2  Lockdown in Europe by an Observationally Constrained Regional**

**3  Model: Non-Linear Joint Inversion of $NO_x$ and VOC Emissions**

**4  using TROPOMI**

Amir H. Souri[1]*, Kelly Chance[1], Juseon Bak[2], Caroline R. Nowlan[1], Gonzalo González Abad[1],
Yeonjin Jung[1], David C. Wong[3], Jingqiu Mao[4,5], and Xiong Liu[1]
[1]Atomic and Molecular Physics (AMP) Division, Harvard–Smithsonian Center for Astrophysics, Cambridge, MA,
USA
[2] Institute of Environmental Studies, Pusan National University, Busan, South Korea
[3]U.S. Environmental Protection Agency, Center for Environmental Measurement & Modeling, Research Triangle
Park, NC, USA
[4]Geophysical Institute, University of Alaska Fairbanks, Fairbanks, AK, USA
[5]Department of Chemistry and Biochemistry, University of Alaska Fairbanks, Fairbanks, AK, USA
* Corresponding author: ahsouri@cfa.harvard.edu
**Abstract.** Questions about how emissions are changing during the COVID-19 lockdown
periods cannot be answered by observations of atmospheric trace gas concentrations alone, in part
due to simultaneous changes in atmospheric transport, emissions, dynamics, photochemistry, and
chemical feedback. A chemical transport model simulation benefiting from a multi-species
inversion framework using well-characterized observations should differentiate those influences
enabling to closely examine changes in emissions. This approach has another advantage in that we
can, to a certain extent, disentangle the chemical and physical processes involved in the formation
of ozone. Accordingly, we jointly constrain $NO_x$ and VOC emissions using well-characterized
TROPOMI HCHO and $NO_2$ columns during the months of March, April, and May 2020
(lockdown) and 2019 (baseline). We observe a noticeable decline in the magnitude of $NO_x$
emissions in March 2020 (14-31%) in several major cities including Paris, London, Madrid, and
Milan expanding further to Rome, Brussels, Frankfurt, Warsaw, Belgrade, Kyiv, and Moscow (34-
51%) in April. The large variability of changes in $NO_x$ emissions is indicative of different dates
and the degree of restrictions enacted to prevent the spread of the virus. For instance, $NO_x$
emissions remain at somewhat similar values or even higher in northern Germany and Moscow in
March 2020 compared to the baseline. Comparisons against surface monitoring stations indicate





that the model estimate of the NO₂ reduction is underestimated, a picture that correlates with the TROPOMI frequency impacted by cloudiness. During the month of April, when ample TROPOMI samples are present, the surface NO₂ reductions occurring in polluted areas are described fairly well by the model (model: -21±17%, observation: -29±21%). Changes in VOC emissions are dominated by eastern European biomass burning activities and biogenic isoprene emissions. In March, however, TROPOMI HCHO sets an upper limit for HCHO changes such that the chemical feedback of NO$_x$ on HCHO constrained by TROPOMI NO₂ reveals a non-negligible decline in anthropogenic VOC emissions in Paris (-9%), Milan (-29%), London (-5%), and Rome (-5%). Results support an increase in surface ozone during the lockdown. In April, the constrained model features a reasonable agreement with maximum daily 8 h average (MDA8) ozone changes observed at the surface ($r$=0.43), specifically over central Europe where ozone enhancements prevail (model: +3.73±3.94%, +1.79 ppbv, observation: +7.35±11.27%, +3.76 ppbv). Results of integrated process rates of MDA8 surface ozone over central Europe in the same month suggest that physical processes (dry deposition, advection and diffusion) decrease ozone on average by -4.83 ppbv, while ozone production rates dampened by largely negative $J_{NO2}[NO_2]$-$k_{NO+O3}[NO][O_3]$ become less negative, leading ozone to increase by +5.89 ppbv. Experiments involving fixed anthropogenic emissions suggest that meteorology (mainly as air temperature and photolysis) contributes to 42% enhancement in MDA8 surface ozone over the same region with the remaining part (58%) coming from changes in anthropogenic emissions. Results illustrate the capability of satellite data of major ozone precursors to help atmospheric models capture the essential character of ozone changes induced by abrupt emission anomalies.

## 1. Introduction

Continuous monitoring of air pollution by satellites can help our understanding of both anthropogenic and biogenic variability and change caused by rapid economic recession [Castellanos and Boersma, 2012] and regulations [Krotkov et al., 2016; Souri et al., 2020a]. Earth's atmosphere has exponentially become more polluted during previous decades because of rapid industrialization increasing anthropogenic emissions [Li and Lin, 2015], thus any abrupt hiatus in these emissions should result in an impulsive and sweeping impact on relatively short lifetime pollutants such as nitrogen dioxide (NO₂), formaldehyde (HCHO), and tropospheric ozone (O₃). The beginning of the global COVID-19 pandemic in early 2020 [Fauci et al., 2020] provided such an abrupt change in human activities [Le Quéré et al., 2020]. A first step to fully understand how





much of these impacts are related to the pandemic lockdowns is to disentangle the physical and
chemical processes determining their ambient concentrations. Unraveling those processes require
precise, continuous observations of physical and chemical states and emission rates, which are not
routinely available on global, continental and regional scales. Therefore, we resort to using a model
realization attempting to reproduce such an intricate system. Models without observational
guidance are incapable of numerically representing the real world [Lorenz, 1963], so our best
option to improve a model is to constrain some of its prognostic inputs using well-characterized
observations. Accordingly, the framework of this study is centered around inverse modeling and
data assimilation.
Significant attention has been given to documenting the lockdown-related changes in
atmospheric compositions around the world using both in-situ and satellite observations [e.g.,
Sicard et al., 2020; Shi and Brasseur, 2020; Lee et al., 2020; Salma et al., 2020; Le Quéré et al.,
2020; He et al., 2020; Le et al., 2020; Miyazaki et al., 2020; Liu et al., 2020; Barré et al., 2020;
Goldberg et al., 2020; Ordóñez et al., 2020; Wyche et al., 2021]. The broad picture is consistent
among these studies; the lockdown drastically reduced the concentrations of $NO_x$, CO, and $SO_2$
and some types of particulate matters, whereas the concentrations of several secondarily formed
compounds such as ozone increased due to emissions and/or meteorology. To the best of our
knowledge, changes in volatile organic compounds (VOCs) due to the lockdown over Europe have
not been reported.
The motivations of this study are to determine the capability of a regional model
constrained by satellite HCHO and $NO_2$ columns to capture near-surface pollution, and if local
ozone production rates are the driving factors for heightening ozone pollution during the 2020
lockdown. In other words, what chemical and physical processes are associated with the elevated
ozone? How representative are satellite observations at capturing surface air quality through an
inversion context? Is meteorology the primary factor in shaping elevated ozone as suggested by
Ordóñez et al. [2020]?
To address these pivotal questions, it is desirable to constrain models using multi-species
observations because relationships between the atmospheric compounds such as HCHO and $NO_2$
are importantly intertwined [Marais et al., 2012; Valin et al., 2016; Wolfe et al., 2016; Souri et al.,
2020a,b]. Accordingly we build our inversion framework upon a non-linear joint analytical
inversion of $NO_x$ and VOCs proposed in Souri et al. [2020a] using TROPOMI HCHO and $NO_2$





observations in Europe. Performing this type of inversion not only enables us to precisely quantify
the impact of the pandemic on emissions (along with its uncertainty, as the inversion framework
is analytical) but also paves the way for estimating the resulting changes on different pathways of
surface ozone.
**2.  Measurements, Modeling, and Methods**
***2.1. Satellite Observations***
*2.1.1.  TROPOMI NO$_2$*

We use daily offline S5P TROPOMI tropospheric NO$_2$ slant columns

[Copernicus Sentinel data processed by ESA and Koninklijk Nederlands

Meteorologisch Instituut (KNMI), 2019] derived from a two-step framework

involving DOAS spectral fitting in conjunction with a stratosphere/troposphere

decoupler [Boersma et al., 2018]. The time periods of this study are March, April

and May 2020 and 2019. The data provide Jacobians of light intensity with respect

to optical thickness (i.e., vertically-resolved scattering weights) which are

dependent on scene surface reflectivity, the cloudiness of the assumed Lambertian

clouds, and sensor viewing geometry. Aerosol effects on the scattering weights are

not taken into consideration. The 2019 TROPOMI observations used in this study

have a spatial resolution of 7×3.5 km$^2$, whereas those in 2020 have a spatial

resolution of 5.5×3.5 km$^2$. The NO$_2$ products for the study time period were

produced by processor versions v01.02.02 (1 March 2019 – 20 March 2019) and

v01.03.02 (20 March 2019 onward). The v01.03.02 processor includes an update

to the FRESCO-S cloud algorithm and improvements to a quality flag variable.

NO$_2$ validation from processors v01.02.02 and v01.03.02 shows similar biases and

dispersion [Lambert et al., 2020], as do comparisons from before and after the pixel

spatial resolution change [Verhoelst et al., 2021]. We extract good quality pixels

based on the main quality flag (qa_flag) > 0.75, which removes retrievals flagged

as bad and pixels over snow/ice or with cloud radiance fractions > 0.5, and resample

them to our 15-km regional model (discussed later) using bilinear interpolation.

Since vertical column densities (VCDs) depend on assumed gas profile shape (i.e.,

they are quasi-observations), we recalculate those shape factors using profiles from

our constrained chemical transport model.



Satellite remote sensing observations are usually far more stable than they
are accurate. This can make the data practical for measuring relative changes in
emissions, but may necessitate the use of a bias correction for absolute emissions
estimates. Moreover, the systematic and random errors associated with satellite
retrievals may differ markedly from location to location. It is therefore crucial to
thoroughly validate columns against independent observations. To this end, we
compile statistics reported in several validation studies focusing on the TROPOMI
tropospheric $NO_2$ product and summarize their findings in Table 1. The most
comprehensive global study to date is a comparison of TROPOMI tropospheric
$NO_2$ with that derived from 19 MAX-DOAS instruments [Verhoelst et al., 2021].
This study indicates there is a low bias in TROPOMI tropospheric $NO_2$ of -23 to -
37% relative to MAX-DOAS at clean to moderately polluted sites, and as large as
-51% at highly polluted sites. When considering all sites, the overall median bias
in this study was found to be -37%, with a RMSE of $3.5 \times 10^{15}$ molec/cm$^2$ (defined
as half of the 68% interpercentile). No obvious seasonal patterns were found in the
biases. These results are consistent with other validation studies which have
observed a low bias in TROPOMI tropospheric $NO_2$ [Chan et al., 2020; Griffin et
al., 2019; Judd et al., 2020]. A potential significant source of bias in polluted
regions is the relatively low-spatial resolution (1×1º) TM5-MP prior profiles used
in the TROPOMI air mass factor calculation. Several validation studies have shown
the low bias in TROPOMI $NO_2$ can be reduced in polluted regions by 5-17%
through the use of higher spatial resolution model a priori profiles or other
improvements in the AMF calculation [Chan et al., 2020; Griffin et al., 2019; Judd
et al., 2020; Zhao et al., 2020]. There are challenges in using the aforementioned
numbers, of which the largest relates to the lack of formulating errors as functions
of prognostic inputs used for the retrievals (e.g., albedo, scene radiance, etc.). This
in turn precludes a more general estimation of errors for all pixels. Given the fact
that our study will derive emissions primarily from information in moderate to
highly polluted areas [Souri et al., 2020a], we uniformly scale up $NO_2$ pixels by
25% based on the low bias determined by Verhoelst et al. [2021] while considering
the potential reduction in the bias through the use of higher spatial resolution trace





| | 156 | gas a priori profiles. We set the RMSE to $1.1\times10^{15}$ molec/cm$^2$ ($<6\times10^{15}$ molec/cm$^2$) |
|---|---|---|

gas a priori profiles. We set the RMSE to $1.1\times10^{15}$ molec/cm$^2$ ($<6\times10^{15}$ molec/cm$^2$)
in clear regions and $3.5\times10^{15}$ molec/cm$^2$ ($>=6\times10^{15}$ molec/cm$^2$) in moderately to
highly polluted regions.
*2.1.2.  TROPOMI HCHO*
We use daily offline S5P TROPOMI HCHO total slant columns
[Copernicus Sentinel data processed by ESA, German Aerospace Center (DLR),
2019]. A full description of the algorithm can be found in De Smedt et al. [2018].
The HCHO products for the study time period were produced by processor versions
v01.01.05 (1 March 2019 – 28 March 2019), v01.01.06 (28 March 2019 – 23 April
2019) and v01.01.07 (23 April 2019 onward). The newer versions have added
updates to the surface classification climatology and cloud products that might have
some effects on the magnitude of HCHO in cloudy scenes. We again remove bad
pixels based on qa_flag < 0.75 and recalculate shape factors using the simulated
profiles derived from our regional model.
Validation efforts reported in the sixth Quarterly Validation Report of the
Copernicus Sentinel-5 Precursor Operational Data Products [Lambert et al., 2020]
indicate varying biases depending on the magnitude of HCHO concentrations in
comparison to ground-based observations. Locations with HCHO concentrations
above $8\times10^{15}$ molec/cm$^2$ show a low bias of ~-31%. Conversely, clean sites with
HCHO concentrations below $2.5\times10^{15}$ molec/cm$^2$ undergo a high bias of 26%.
Those biases oscillate around $8\times10^{15}$ molec/cm$^2$. Vigouroux et al. [2020] expanded
the validation suite by including more than 25 FTIR stations majorly located over
pristine areas and 9 MAX-DOAS stations located in polluted sites. Results from
the comparison with FTIR measurements (over clean areas) indicate a high bias,
whereas those compared with MAX-DOAS measurements at Cabauw and De Bilt
(Netherlands) show a low bias of -44%. The agreement between MAX-DOAS and
satellite observations improved after adjusting TROPOMI shape factors based on
MAX-DOAS observations. By compiling numbers quoted in Lambert et al. [2020]
and Vigouroux et al. [2020], we correct the existing biases in TROPOMI HCHO
by scaling 25% ($<2.5\times10^{15}$ molec/cm$^2$) down columns in clear areas and 30%



($>=7.5\times10^{15}$ molec/cm$^2$) up in polluted areas. We set the magnitude-dependent
RMSE to be equal to 4% of HCHO total columns based on Vigouroux et al. [2020].
*2.1.3.  MODIS AOD*

To improve the simulation of total aerosol mass, we use the collection 6

MODIS aerosol optical depth (AOD) from both Aqua (~ 13:30 LT) and Terra (~
10:30 LT) platforms over both land and ocean [Levy et al., 2013] (available at
https://ladsweb.modaps.eosdis.nasa.gov, access May 2020). We independently
validate all three major products, namely the deep blue, the dark target and a
combined dark blue products by comparing to AOD values measured by
AERONET over Europe at the same time period of this study. Only good and very
good (qa$>=2$) pixels are selected for the comparison. The AERONET AOD data
are computed based on the values at 500 nm and Angstrom Exponent in the 440-
675 nm range. We collocate two datasets if they are within 10 km radius and less
than 30 mins apart. The dark blue product results in the best agreement ($r>0.87$)
with a high bias of <0.05 (Figure S1, and S2). We remove the bias and assign the
value of the covariance matrix of observations to the RMSE values obtained from
the comparison.
***2.2. Surface Measurements***

UV photometry and chemiluminescence surface ozone and NO$_2$ measurements all

over continental Europe are used to investigate possible changes in their concentrations
induced by the lockdown (https://discomap.eea.europa.eu/map/fme/AirQualityExport.htm,
access June 2020). The NO$_2$ chemiluminescence measurements are usually overestimated
due to interferences from the NO$_z$ family (PAN, organic nitrate, HNO$_3$, etc.). We assume
that the interferences are not significantly different between the baseline and lockdown
mainly due to relatively low photochemistry in early spring [Lamsal et al., 2008] compared
to summertime.

More than 6450 meteorological stations archived on NOAA's integrated surface

database (https://www.ncei.noaa.gov/data/global-hourly/, access April 2020) are used to
validate the performance of our weather model in terms of several prognostic inputs
including ambient air temperature, air humidity, and U and V wind components.



### 2.3. WRF-CMAQ Modeling

The regional air quality simulations at $15 \times 15$ km$^2$ are carried out with the widely
used CMAQ v5.2.1 (https://doi.org/10.5281/zenodo.1212601) in conjunction with WRF
v3.9.1 [Skamarock et al. 2008] models. The models overlap and cover continental Europe
and some portions of Africa and Middle East. The domain consists of 483 east-west, 383
north-south grids, and 37 unevenly spaced eta levels (Figure 1). The simulation time period
is from March to May 2019 and 2020 (six months). Since IC/BC are taken from already
spun-up National Centers for Environmental Prediction (NCEP) FNL (final) reanalysis and
GEOS-Chem v12.9.3 (10.5281/zenodo.3974569) runs, we only spin up the models for the
month of February. The chemistry configuration of the CMAQ model mainly consists of
CB05 with chlorine chemistry (gases) and AERO6 (aerosol). Biogenic emissions are
processed by the offline standalone Model of Emissions of Gases and Aerosols from Nature
(MEGAN) v2.1 model [Guenther et al., 2012] based on high-resolution plant functional
maps made by Ke et al. [2012]. Anthropogenic emissions are based on the Community
Emissions Data System (CEDS) inventory in 2014 [Hoesly et al., 2018]. We also output
the CMAQ integrated process analysis quantifying the contribution of each process to the
amount of compounds. The physical setting of WRF includes the Lin microphysics scheme
[Lin et al., 1983], the Grell 3-D ensemble cumulus scheme [Grell and Dévényi, 2002], the
RRTMG radiation scheme, ACM2 planetary boundary layer parametrization [Pleim,
2007], and Pleim-Xu land-surface scheme [Xiu and Pleim, 2001]. We nudge moisture,
wind and temperature fields toward the reanalysis data used only outside of the PBL layer.
Moreover, leaf area index and the sea surface temperature are updated every 6 hours based
on satellite measurements included in the reanalysis data. Extensive model evaluations
based upon surface observations show a striking correspondence (Table S1, S2) which is
indicative of fair energy budget and transport in our model.

### 2.4. Inverse Modeling and Data Assimilation

To adjust the bottom-up emission inventories, we follow a non-linear joint
inversion method proposed in Souri et al. [2020a]. Briefly, a Gauss-Newton algorithm is
utilized to incrementally solve the Bayes' quadratic function in analytical fashion. The
posterior emissions are then derived by

$$\mathbf{x}_{i+1} = \mathbf{x}_a + \mathbf{G}[\mathbf{y} - F(\mathbf{x}_i) + K_i(\mathbf{x}_i - \mathbf{x}_a)] \tag{1}$$





where $y$ is bias-corrected TROPOMI $NO_2$ and HCHO observations, $\mathbf{x}_a$ (or $\mathbf{x}_0$) is the prior
emissions, $\mathbf{x}_i$ is the posterior emission at the $i$th increment, $F$ is the forward model (here
WRF-CMAQ) to project the emissions onto columns space, $\mathbf{G}$ is the Kalman gain,

$$\mathbf{G} = \mathbf{S}_e \, K_i^T \left( K_i \mathbf{S}_e \, K_i^T + \mathbf{S}_o \right)^{-1} \tag{2}$$

and $K_i \,(= K(\mathbf{x}_i))$ is the Jacobian matrix calculated explicitly from the model using the finite
difference method. $\mathbf{S}_o$ and $\mathbf{S}_e$ are the error covariance matrices of the observations and
emissions. In terms of the prior errors, we use the numbers reported in Souri et al. [2020a].
The instrument covariance matrices are populated with squared-sum of the aforementioned
RMSEs based on the compilation of the validation studies and precision errors provided
with the data. Both error matrices are diagonal. The inversion window is monthly. The
covariance matrix of the a posteriori is calculated by:

$$\hat{\mathbf{S}}_e = (\mathbf{I} - \mathbf{G}\widehat{K})\mathbf{S}_e \tag{3}$$

where $\widehat{K}$ is the Jacobian from the $i$th iteration. Here we iterate Eq.1 three times. The
averaging kernels (**A**) are given by:

$$\mathbf{A} = \mathbf{I} - \hat{\mathbf{S}}_e \mathbf{S}_e^{-1} \tag{4}$$

Not only does this method considers non-linear chemical feedback among $NO_2$-
HCHO-$NO_x$-VOC by simultaneously incorporating the HCHO and $NO_2$ in the inversion
framework, it also permits quantification of **A** that explicitly explains the amount of
information obtained from the observation.
We also correct total aerosol mass by daily assimilating the MODIS dark blue AOD
observations following the algorithm discussed in Jung et al. [2019]. Briefly, the
assimilation framework uses a modified optimal interpolation method adjusting uniformly
all relevant aerosol masses in a column as a function of a weighted-distance and appropriate
errors.
**3. Results and Discussion**
*3.1. Variability of HCHO and $NO_2$ columns seen by TROPOMI*
We assess difference maps of TROPOMI HCHO and $NO_2$ columns in 2020 with respect
to those in 2019 during the months of March, April and May. The difference maps along with the
absolute values of the tropospheric $NO_2$ columns are shown in Figure 2. Regardless of the year,
we observe a noticeable reduction in $NO_2$ as we approach warmer months which can be explained



by increases in OH concentrations (higher water vapor content, solar radiation, and $O_3$ levels),
faster vertical mixing due to larger sensible fluxes (more diluted columns due to stronger advection
in higher altitudes), and a reduction in temperature-dependent light-duty diesel $NO_x$ emissions
[Grange et al., 2019]. Two unintended consequences of this sequential decline, noted by Silvern
et al. [2019] and Souri et al. [2020a], are first the free-tropospheric region complications and
second a barrier to obtaining high amount of information from the sensor which manifests itself in
lower averaging kernels of emission estimates (shown later). The anomaly map in March indicates
pronounced decreases in tropospheric $NO_2$ columns over several countries including France,
Spain, Italy, and Germany (box A). In contrast, we see negligible reductions in the magnitude of
the $NO_2$ columns over some portions of the UK excluding London (box B), northern Germany
(box C), and Moscow, Russia (box D). A very recent study [Barré et al. 2020] observed roughly
the same tendency which was attributable to meteorological changes. While those changes are
indeed an important piece of information that will be investigated later in this study, we should
recognize that the degree of the enforced restrictions varies both spatially and temporally;
moreover changes in emission heavily rely on the dominant emission sector (e.g., mobile or
industry) and population. For instance, northern Germany is associated with less populated areas
and industrial areas which might be less impacted by the shutdown (see Figure 2 in Le Quéré et
al. [2020]), and as a result, we would expect a weaker signal in the reduction of $NO_2$. According
to TASS press [https://tass.com/society/1144123, accessed Sep 2020], Russian governments did
not take significant measures to control the virus before April 15, immediately evident in the large
$NO_2$ enhancement over Moscow in March (box D). During the next two months (April and May),
we observe a major turnaround over this city (box F and H). In May, the anomaly of the
tropospheric $NO_2$ suggests an abrupt hiatus in the ongoing reduced $NO_x$ emissions in central
Europe (box G). However it is crucial to note that these maps are based upon sporadic clear-sky
pixels that might obscure the full portrayal of emissions changes happening throughout the period
(discussed later).

We further investigate potential changes in HCHO total columns shown in Figure 3 in the

same context as we discussed for $NO_2$. Various VOCs with different sources contribute to the
formation of HCHO (see Figure 2 in Chan Miller et al. [2016]) leading to striking HCHO column
patterns with large variations. In theory, we have a higher chance to single out anthropogenic-
derived HCHO concentration by looking at wintertime measurements, although temperature and



photochemistry are always key influencers of oxidizing/photolyzing all types of VOCs. The
inevitable trade-off for this is dealing with a weaker signal that is near to instrument detection
limit. Encouragingly, the TROPOMI HCHO retrieval offers a very low detection limit for
individual pixels ($7 \times 10^{15}$ molec/cm$^2$) that can be further lowered down by co-adding
measurements (roughly a factor of $1/\sqrt{n}$).  Accordingly, we observe a promising signal in March
over eastern European countries that is not explainable by biogenic emissions; but the magnitudes
of the difference over these areas ($<1.5 \times 10^{15}$ molec/cm$^2$) are below the detection limit ($\sim 2.4 \times 10^{15}$
molec/cm$^2$ given the co-added measurements over time) to relate them to the lockdown in a robust
manner; nonetheless TROPOMI sets an upper limit for these changes. In April, results show
elevated HCHO concentrations in high latitudes in 2019 (box I), mainly a result of biomass burning
activities in eastern Europe [e.g., Karlsson et al. 2013]. As temperature rises in May, the footprint
of biogenic emissions become more visible. This signal is not only induced by the inherent
temperature-dependency of biogenic isoprene emissions, but also stems from the fact that isoprene
reactivity significantly increases by rising temperature [Pusede et al. 2015]. The dipole anomaly
of HCHO columns suggested by TROPOMI (box J and K) pertains largely to variations in ambient
surface air temperature (shown later).
### 3.2. Top-Down estimates of $NO_x$ and VOC emissions
Following the inversion and the data assimilation frameworks, we adjust the total amounts
of VOC, $NO_x$ emissions, and aerosols mass using the well-characterized TROPOMI HCHO, $NO_2$
and MODIS AOD observations for the study time period. We focus on the topic of gas phase
chemistry (i.e., ozone and its precursors) implying that the aerosol data assimilation is carried out
to partially remove errors associated with radiation [e.g., Jung et al., 2019] or heterogenous
chemistry [Jacob, 2000], therefore, the aspect of aerosol changes induced by the lockdown will be
examined elsewhere. The spatial distributions of magnitude of the top-down $NO_x$ and VOC
emissions (i.e., constrained by the observations), their corresponding changes and averaging
kernels are shown in Figure 4 and Figure 5, respectively. It is worth emphasizing that we use
identical prior values in terms of anthropogenic emissions in both years; therefore, the differences
in the top-down emissions are primarily dictated by the observations used in the inversion.
According to Figure 4, large averaging kernels associated with $NO_x$ emissions are confined in
high-emitting regions suggesting that the most valid estimates can be found in areas undergoing
strong TROPOMI $NO_2$ signals. We observe a large reduction (31-45%) in the bias associated with



simulated surface $NO_2$ using the posterior emissions compared to the surface measurements in
Europe, although improvements in correlation were minimal (not shown). Similarly, as expected,
the discrepancies between the simulated tropospheric $NO_2$ columns versus TROPOMI are largely
mitigated by the inversion (Figure S3 and S4). Immediately apparent in Figure 4 is a strong
correlation between anomaly maps of TROPOMI tropospheric $NO_2$ (Figure 2) and those of top-
down emissions. However, in practical terms, the magnitude of these anomalies is not as drastic
as the ratio of observation to model ratio because of the consideration of observational errors and
chemical feedback [Souri et al., 2020a], which always leaves some doubt about the practicality of
direct mass balance methods. We observe reductions in $NO_x$ emissions in March (14-31%) in
several major cities including Paris, London, Madrid, and Milan; the reductions further expand to
Rome, Brussels, Frankfurt, Warsaw, Kyiv, Moscow, and Belgrade with higher magnitudes (34-
51%) in April. Table 2 summarizes the absolute and relative differences in total $NO_x$ emissions
estimated by the inversion binned to different regions in Europe based on country land borders. In
general, the level of $NO_x$ reduction is somewhat higher in April relative to months of March and
May owing to spatiotemporal variabilities associated with the restrictions; for example, UK and
Poland governments enforced the restrictions starting in the last week of March to the middle of
April, a situation that clearly shows up in our results. Interestingly, the decreased anthropogenic
$NO_x$ emissions in the strait of Gibraltar and Alboran Sea reveal reportedly reduced ship activities
[United Nations Conference on Trade and Development Report, Accessed Dec 2020]. The
numbers in May indicate that several countries in central and eastern Europe (shown in box G in
Figure 2) likely eased coronavirus lockdown restrictions, a picture that has yet to be verified by
surface measurements (discussed later).

As to VOC emissions, we observe a significant improvement in the magnitude and spatial

distribution of simulated HCHO columns after the inversion with respect to TROPOMI data
(Figure S5 and S6). It is very evident that the magnitudes of the emissions primarily follow
anthropogenic sources in March; expectedly, very low averaging kernels over major European
cities in this month are indicative of inadequacies of one-month averaged TROPOMI HCHO data.
However, we surprisingly observe a noticeable decline in the amount of VOC emissions (majorly
anthropogenic) in Paris (-9%), Milan (-29%), London (-5%), and Rome (-5%). All of these cities
emit considerable amounts of VOCs during wintertime [Schneidemesser et al., 2011; Possanzini
et al., 2002; Baudic et al., 2016]. This tendency, which is striking, mainly stems from the indirect





impacts of the reduced $NO_x$ emissions on HCHO formation [Marais et al., 2012; Valin et al., 2016;
Wolfe et al., 2016; Souri et al., 2020b]. The sensitivity of HCHO levels to VOC emissions is
controlled by the availability of OH that is impacted by $NO_x$. A decrease in $NO_x$ emissions in $NO_x$-
saturated areas frees up more OH to faster oxide VOCs [Souri et al., 2020b] resulting in a steeper
gradient of HCHO with respect to its sources. Likewise we observe larger Jacobians of HCHO
with respect to VOC emissions in 2020 over the cities mentioned (not shown). If we assume the
relative changes in HCHO levels between the two years to be insignificant, which are suggested
by TROPOMI HCHO (considering the errors in the retrieval), the steeper gradient of HCHO
concentrations with respect to VOC emissions should normally lead to a reduction in the VOC
emissions in 2020. In other words, it would require a smaller VOC emission rate to reach to the
same amount of HCHO. We note that the TROPOMI HCHO observations provide an upper limit
of the changes so that we can make this assumption. Table 3 summarizes the amount of VOCs
changing in the cities mentioned. The inversion partly corrects for the large underrepresentation
of biomass burning emissions in high latitudes occurring in April 2019 but due to large
uncertainties of the retrieval over this area, averaging kernels are low. We revisit the pronounced
dipole anomaly of dominantly biogenic VOC emissions in May. It is readily evident from the
averaging kernels that more realistic information from TROPOMI HCHO is attainable in warmer
months, contrary to the $NO_2$ case.
**3.3. Disparities and rationale behind the differences in near-surface concentrations**
**suggested by the constrained model versus those by in-situ measurements**
### 3.3.1. $NO_2$ and HCHO
We further investigate the effect of the lockdown on the surface HCHO and $NO_2$
concentrations based on the constrained simulations. Figure 6 gives the difference maps (lockdown
minus baseline) of daily-averaged surface $NO_2$ and HCHO overplotted with the differences of
surface wind vectors, planetary boundary layer heights (PBLHs), surface air temperature, and the
ratio of photolysis rates below clouds ($J_{below}$) to those in clear-sky conditions ($J_{clear}$) following
Madronich [1987]. The anomaly of emissions is on par with those of surface $NO_2$ and HCHO
surface concentrations, this is perhaps not surprising, since the emissions are mostly located near
the surface. Horizontal transport (shown as wind vectors) plays a critical role in explaining the
spatial variations in emissions downwind.


PBLH describes the level of vertical diffusion of air parcels [Jacobson, 2005]. The increase
(decrease) in the PBLH is an indication of more (less) diluted air, subject to assuming that the
pollutant concentration (in mixing ratio) would exponentially decrease aloft. The extent to which
PBLH can impact air pollution relative to advection is strongly dependent upon the wind speed
(see Figure 8 in Su et al., [2018]). The stronger the wind, the more likely PBLHs are going to be
of secondary importance. We subjectively identify calm conditions by assuming that wind speeds
should be below 1 m/s. We overlay the calm conditions as black dots over the PBLH contour.
Although model uncertainties exist, the less pronounced $NO_2$ reduction over UK and northern
Germany in March is unlikely to be resulting from shallower PBLHs in 2020 given how strong the
predominant winds are. A strong expansion of PBL over the central Europe in April and May 2020
relative to 2019 possibly contributes to a larger reduction of $NO_2$ concentration.
Because of relatively colder air and less photochemistry in March, VOCs become naturally
less reactive. This in turn will provide an opportunity for the volume of air to become dispersed.
Thus the reduced VOC emissions over several major cities influence larger areas and become less
distinctive. The temperature dependency of HCHO concentration progressively becomes more
pronounced with increasing temperature. Both photochemistry and biogenic derived emissions are
a function of shortwave solar radiation [Madronich, 1987; Guenther et al., 2012; Stavrakou et al.,
2014] that can vary significantly with cloud transmissivity and the solar zenith angle. The ratio of
$J_{below}/J_{clear}$ well describes such a relationship; positive (negative) differences in the ratio suggest
more (less) photochemistry. The strongly positive ratio of $J_{below}/J_{clear}$ over the central Europe in
April potentially overrides the fluctuations associated with surface temperature leading HCHO
levels to rise.
Clearly, with the help of the CMAQ process analysis, more quantitative work on relevant
physical/chemical processes pertaining to $NO_2$ and HCHO surface concentrations can be done
here, but before proceeding it is necessary to examine whether the constrained model can
adequately represent the changes observed by surface measurements. Unfortunately we limit the
analysis to $NO_2$ due to the lack of routinely measured HCHO observations. Several factors can
complicate this analysis: i) having overconfidence in the constrained model where the satellite
observations used were uncertain; this problem can be safely addressed by considering grid cells
whose averaging kernels are above a threshold (here 0.5), ii) ignoring spatial representivity
function to directly compare point measurements to the model grids; a statistical construction of





the spatial representivity function [Janic et al., 2016] requires a dense observational network so
that we can build a semivariogram; instead, we only consider model grid cells having more than
two stations; those observations then are then averaged, iii) interferences of $NO_z$ family on $NO_2$
chemiluminescence measurements [Dickerson et al., 2019] which can be partly discounted when
calculating differences, iv) model uncertainties, especially with respect to turbulent and convective
fluxes that are heavily determined by representing local heterogenicity of forces and non-
hydrostatic dynamics [Emanuel, 1994], all of which are challenging to properly resolve in a 15-
km resolution. With these caveats in mind, we plot the daily-averaged changes of surface $NO_2$
concentrations in 2020 relative to 2019 derived by the model and the European air quality network
for the months of March, April, and May (Figure 7). Large gaps in Figure 7 are caused by
considering grid cells with averaging kernels>0.5 and number of samples>2. The constrained
model correlates reasonably well with the changes observed by the surface measurements in March
and April, but it fails to reflect those in May. The surface measurements reinforce the less
pronounced reduction in $NO_2$ in northern Germany and UK, although the magnitudes are not as
large as those suggested by the model. The constrained model tends to consistently underrepresent
the decline in $NO_2$ in March (model: -11±21%, observation: -19±16%), April (model: -21±17%,
observation: -29±21%), and May (model: -12±18%, observation: -25±20%). The frequency of
TROPOMI data heavily impacted by cloudiness is another factor that can effectively lead to the
underrepresentation of the model in a course of a month. Figure 8 depicts the number of days that
TROPOMI was able to sample on. There is a strong degree of correlation between the frequency
of the data and the discrepancy between the model versus the surface observations. This is
especially the case for May when we see too few days to be able to realistically reproduce $NO_2$
changes. Given the reasonable performance of our model at reproducing the changes observed
over the surface in April, a result of abundant samples from TROPOMI, we only focus on this
month for the subsequent analysis.

***3.3.2. Ozone***

Figure 9 depicts the changes in maximum daily 8 h average (MDA8) surface ozone
concentrations suggested by the measurements and the constrained model in April. Immediately
obvious from the observations is the elevated surface ozone concentrations up to 32% in places
where $NO_x$ emissions drastically decreased such as Germany, Italy, France, UK, Switzerland, and
Belgium (shown as box L). This tendency potentially is driven by ozone chemistry [Sicard et al.,



2020a; Shi and Brasseur, 2020; Grange et al. 2020; Salma et al., 2020; Lee et al., 2020] and/or
meteorology [Lee et al., 2020; Wyche et al., 2021; Ordóñez et al., 2020] has drawn much attention.
The challenge is to simulate a model that is the characteristic of such a complex tendency [e.g.,
Parrish et al., 2014]. Encouragingly, our constrained model does have skill in describing the
essential character of the ozone enhancements over the whole domain ($r$=0.43). In the proximity
of central Europe (shown as box L), the enhanced MDA8 ozone concentration observed by the
observations is 7.35±11.27% (+3.76 ppbv) which is nearly a factor of two larger than that of the
model (3.73±3.94%, +1.79 ppbv).

While the remaining model uncertainty could be either improved or characterized by

including more observations (if available), reconfiguring the physical/chemical mechanisms used,
and constraining chemical boundary conditions, it is imperative to gauge the contribution of each
process (i.e., transport, chemistry, etc.) in forming ozone changes. Here we mainly make use of
the CMAQ process analysis. A direct use of the process analysis output (in unit of ppbv hr$^{-1}$) can
be confusing as both physical/chemical processes and underlying concentrations are inextricably
linked together. To be able to isolate each process (in unit of hr$^{-1}$), we normalize the outputs by
ozone concentrations. Here, we average each process at the same hours used in calculating MDA8.
Figure 10 shows the major model processes, namely as horizontal transport (horizontal advection
plus diffusion), vertical transport (vertical advection plus diffusion), dry deposition, and chemistry
in 2020, 2019, and their differences. Positive (negative) values indicate a source (sink) for ozone.
Regarding the horizontal transport, the values mostly follow the transport pattern and are
dependent on whether the advected air mass is more or less polluted. The vertical transport
correlates with the PBLH which is an indicator of the atmospheric stability and turbulence,
although we should not rule out the impact of the subgrid convective transport that can occur
sporadically. Low PBLHs are usually associated with more stable (or sometimes capping
inversion) and weaker vertical mixing [e.g., Nevius and Evans, 2018]. Vertical transport which is
majorly dictated by the vertical diffusion is by far the most influential factor in the magnitude of
ozone [e.g., Cuchiara et al., 2014]. In contrast to $NO_2$ and HCHO, a stronger vertical diffusion
increases surface ozone due to positive gradients of ozone with respect to altitude. However, the
aerodynamic resistance controlling dry deposition velocity [Seinfield and Pandis, 2006] is also a
function of turbulent transport. For example, during daytime, intensified turbulence exposes more
pollution to surface deposition. It is because of this reason that we see the dry deposition process



largely counteracting vertical transport. This will leave the chemistry process the major driver of
the ozone changes.
We separately sum the quantities of the physical processes and PO$_3$ contributing to MDA8
surface ozone changes binned to box L. The physical processes lead to -4.83 ppbv changes in the
MDA8 ozone mainly due to a relatively larger dry deposition in 2020, whereas P(O$_3$) contributes
to +5.89 ppbv. The net effect is +1.01 ppbv which is slightly smaller than the simulated changes
in MDA8 ozone in this region (+1.79 ppbv). This apparent discrepancy is caused by the differences
in boundary and initial conditions which are not quantifiable by the process analysis and would
require additional sensitivity test. Nonetheless, we believe these numbers should provide
convincing evidence on the fact that chemistry has promoted the enhancements of surface ozone
during the lockdown.
Chemistry is also a function of meteorology, specifically solar radiation and temperature.
A typical scenario to isolate emissions from meteorology is by running the model with fixed
anthropogenic emissions (and boundary conditions) and subtracting the outputs from the variable
emission output. Figure 11 shows the contribution of anthropogenic emissions (VOCs and NO$_x$)
to the changes seen over the surface. The anthropogenic emissions make up roughly 58% of the
changes. The map is strongly in line with the changes in NO$_x$ emissions constrained by TROPOMI.
The impact of meteorology plus biogenic changes (the former is dominant) highly correlates with
anomalies in both surface air temperature and photolysis rate (Figure 6). We observe negligible
ozone changes due to emissions over Iberian Peninsula reinforcing the significance of the
meteorological impacts [Ordóñez et al., 2020].
**3.4.Ozone chemistry**
Figure 12 shows the numerically-solved ozone production rates (PO$_3$) simulated by the
constrained model during the MDA8 hours period. We observe positive PO$_3$ in less polluted areas
and eastern Europe where biomass burning activities occurred in 2019, while negative PO$_3$ in
major cities. Negative values in PO$_3$ are indicative of either loss in O$_3$ or O$_3$-NO-NO$_2$ partitioning.
The difference in PO$_3$ between the two years suggests that the ozone enhancement in box L is
caused by a reduction in negative PO$_3$ in 2020 over major cities compared to 2019. To examine
which pathways are contributing to this pattern, we attempt to analytically reproduce the
numerically-solved PO$_3$ (Figure 12) through two different equations: the first equation widely
applied in photochemically active environments follows [Kleinman et al., 2002]:





$$P(O_3) = k_{HO_2+NO}[HO_2][NO] + \sum k_{RO_{2i}+NO}[RO_{2i}][NO]$$
$$- k_{OH+NO_2+M}[OH][NO_2][M] - k_{HO_2+O_3}[HO_2][O_3]$$
$$- k_{OH+O_3}[OH][O_3] - k_{O(^1D)+H_2O}[O(^1D)][H_2O] - L(O_3 \qquad (5)$$
$$+ VOCs)$$

This equation yields negative values only if the $O_3$ loss pathways including $NO_2$+OH, $HO_x$+$O_3$ ,
$O^1D$+$H_2O$ and $O_3$+VOCs dominate over the first two terms. The second equation which is
independent of $RO_2$ and $HO_2$ concentrations [Thornton et al., 2002], is:

$$P(O_3) = jNO_2[NO_2] - k_{OH+NO_2+M}[O_3][NO] \qquad (6)$$

In summer, this equation tends to be positive during early afternoon, almost zero during afternoon
(steady-state), and negative in early morning (or night) in which the second term ($O_3$ titration) is
leading. Any abrupt changes in $NO_x$ and VOC, and photolysis can directly affect equation 6
moving $PO_3$ out of the diel steady-state. The assumption of the steady-state ($PO_3$ from equation 6
equals to zero) is also not valid if an air parcel is in the vicinity of high-emitting $NO_x$ sources
[Thornton et al., 2002].
Figure 13 displays the reactions rates of each individual component involved in equation 5
averaged during the MDA8 hours. $HO_2$+NO is the dominant chemical source of ozone correlating
well with the changes in $NO_x$ and prevailing chemical conditions regimes ($NO_x$-sensitive vs VOC-
sensitive). Souri et al. [2020a] found the reaction of $RO_2$+NO to be primarily dependent on VOCs.
Likewise, we observe a strong degree of correlation between the anomaly of $RO_2$+NO and that of
VOCs (Figure 3). Figure 13 indicates that the chemical pathways of ozone loss are rather constant
between the two years; therefore the largely negative $PO_3$ over urban areas shown previously in
Figure 12 is not reproducible using this equation. Figure 14 shows the reactions rates of $J_{NO2}[NO_2]$,
$k_{NO+O3}[NO][O_3]$, and the difference during the MDA8 hours. The difference maps replicate the
largely negative $PO_3$ over cities suggesting that we are not in the diel steady-state, and $O_3$ titration
is prevailing due to relatively low photochemistry in the springtime. Table 4 lists the averaged
reactions rates involved in equation 5 and 6 along with the numerically-solved $PO_3$ shown in
Figure 12 over box L. These numbers suggest that the major chemical pathways of enhanced ozone
are through $J_{NO2}[NO_2]$ and $k_{NO+O3}[NO][O_3]$, implying that $O_3$-NO-$NO_2$ partitioning is more
consequential than other chemical pathways. This analysis strongly coincides with Lee et al.



[2020] and Wyche et al. [2021] who observed roughly constant $O_3+NO_2$ concentrations over the
UK before and during the lockdown 2020.
**4. Summary**
The slowdown in human activities due to the COVID-19 pandemic had an immediate and
sweeping impact on air pollution over Europe [Barré et al. 2020; Siccard et al., 2020]. Satellite
monitoring systems with large spatial coverage help shed light on the spatial and temporal extent
of those impacts. The relationships between satellite-derived HCHO and $NO_2$ columns and near-
surface emissions have proven difficult to fully establish without using realistic models, capable
of providing insights on the convoluted processes involving chemistry, dynamics, transport, and
photochemistry and therefore help with deciphering what anomaly maps of satellite concentrations
are suggesting [e.g., Goldberg et al., 2020]. To address these challenges, we jointly constrained
$NO_x$ and VOC emissions using TROPOMI HCHO and $NO_2$ columns following a non-linear Gauss
Newton method developed in Souri et al. [2020a], in addition to assimilating MODIS AOD
observations based on Jung et al. [2019]. The constrained emissions also permitted investigating
the simultaneous effects of physical and chemical processes contributing to ozone formation,
illuminating the complexities associated with non-linear chemistry,
Several implications of the derived emissions for the months of March, April, and May
2020 (lockdown) relative to those in 2019 (baseline) were investigated. First, as previously
reported [Sicard et al., 2020; Barré et al. 2020], we observed a significant reduction in $NO_x$ in
March (14-31%) in several major polluted regions including Paris, London, Madrid, and Milan.
The reductions were further seen in other cities such as Rome, Brussels, Frankfurt, Warsaw,
Belgrade, Kyiv, and Moscow (34-51%) in April. Second, a large spatial and temporal variability
associated with the reduction in $NO_x$ was evident, as each country might have different level and
timeline of restrictions. For instance, $NO_x$ emissions decreased drastically in April rather than
March in some portions of UK, northern Germany, Moscow, and Poland. Third, we showed that
anthropogenic VOC emissions over Paris (-9%), Milan (-29%), London (-5%), and Rome (-5%)
decreased in March, a picture that was achievable through jointly using $NO_2$ and HCHO
observations. The reduced anthropogenic VOC emissions were a result of two key assumptions:
the reduced $NO_x$ emissions in $NO_x$-rich areas increased HCHO made from VOCs (evident in larger
Jacobians derived from the regional model), and TROPOMI HCHO suggested a negligible
difference in HCHO concentration between the two years. This striking result emphasizes the


importance of building a multi-specie framework into inverse modeling studies, as the intertwined
chemical feedback between HCHO and $NO_2$ is quite important and shown in proof of concept by
Marais et al. [2012], Valin et al. [2016], Wolfe et al. [2016], and Souri et al. [2020b]. Fourth,
changes in VOC emissions were primarily dictated by biogenic and biomass burning sources in
April and May.
The constrained model calculations gave good representations of near-surface $NO_2$
changes in April (model: -21±17%, observation: -29±21%) in places where the top-down estimates
are reasonable (averaging kernels > 0.5), but inferior representations in other months, especially
in May (model: -12±18%, observation: -25±20%). This tendency mainly arose from TROPOMI
frequencies; too few days (10-26%) in May due to cloudiness precluded the determination of
realistic $NO_x$ emission changes.
We observed surface MDA8 ozone increase from both model and measurements in April
2020 with respect to the baseline. Comparisons of calculation by the constrained model in terms
of MDA8 surface ozone found reasonable agreement with observations in the proximity of central
Europe in April (model: +3.73±3.94%, +1.79 ppbv, observation: +7.35±11.27%, +3.76 ppbv).
These comparisons indicate that the performance of the constrained model to reproduce the ozone
enhancement feature is promising, suggesting fruitful information in TROPOMI $NO_2$ and HCHO,
although reasons behind the underestimation of the enhancement remained unexplained. It was
clear that the dominantly negative ozone production rates dictated by $O_3$-NO-$NO_2$ partitioning
($J_{NO2}[NO_2]$-$k_{NO+O3}[NO][O_3]$) became less negative primarily due to the reduced $NO_x$ emissions in
urban areas where $O_3$ titration occurred. This tendency was in agreement with studies of Lee et al.
[2020] and Wyche et al. [2021]. We found negligible differences in ozone production from
$[HO_2+RO_2][NO]$ and ozone loss from $O^1D+H_2O$ and $O_3+HO_x$ between the two years suggesting
photochemistry was rather low in the springtime over Europe.
We further quantified the contributions of physical processes (transport, diffusion and dry
deposition) and chemistry to the formation/loss of ozone using the integrated process rates. The
physical processes decreased MDA8 ozone by -4.83 ppbv resulting from relatively larger dry
deposition in 2020, whereas chemistry (ozone production) augmented ozone levels by +5.89 ppbv,
indicating that rising ozone was primarily impacted by changes in chemistry. Enhanced air
temperature and photolysis in 2020, both of which were well captured in our model, also affected
chemistry. Experiments with fixed anthropogenic emissions underwent significant enhancement





in surface MDA8 ozone over central Europe, but those only contribute to 42% of the total enhancement indicating that anthropogenic emissions were the major factor.

The results shown here reveal previously unquantified characteristics of ozone and its precursors emission changes during the lockdown 2020 in Europe. We have been able to measure the amount of changes along with the level of confidence in $NO_x$ and VOC emissions using a state-of-the-art inversion technique by leveraging well-characterized satellite observations, which in turn, allowed us to unravel the chemical and physical processes contributing to increased ozone in Europe. Unless a comprehensive air quality campaign targeting COVID-19 related lockdown is available, we recommend that the impact of lockdown on air pollution should be examined through the lens of well-established models constrained by publicly available data, especially those from space in less cloudy environments.

**Author contributions**

AHS designed the research, analyzed the data, conducted the inverse modeling and atmospheric modeling (for CMAQ, GEOS-Chem, WRF, and MEGAN), made all figures, and wrote the paper. JB validated WRF-CMAQ model and reformatted the surface observation files. CRN and GGA did literature review regarding the TROPOMI validation. YJ validated MODIS AOD. DW helped with implementing the AOD assimilation framework. KC, JM, and XL guided the discussion. All authors contributed to discussion and edited the paper.

**Data availability**

The atmospheric inversion data are publicly available from Souri et al. [2021]. The model outputs are available upon the request from ahsouri@cfa.harvard.edu. The links on where to download surface and satellite observations that are used in this study are already provided in the text.

**Acknowledgments**

Amir H. Souri acknowledges supports from the Smithsonian Astrophysical Observatory (SAO) Scholarly Award (40488100AA50203), MethaneSAT LLC, and Environmental Defense Fund. J. Bak acknowledges Basic Science Research Program through the National Research Foundation of Korea (NRF) funded by the Ministry of Education (2020R1A6A1A03044834). Both calculations and simulations are done  on the Smithsonian Institution High-Performance Cluster (SI/HPC) (https://doi.org/10.25572/SIHPC). The views expressed in this manuscript are those of the authors alone and do not necessarily reflect the views and policies of the U.S. Environmental Protection Agency. EPA does not endorse any products or commercial services mentioned in this publication.



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





**Table 1.** Statistics reported in several validations studies comparing TROPOMI
tropospheric $NO_2$ against independent observations.

| Study | Location | Time Period | Benchmark Instrument | Bias (low) | RMSE | Modification | Modified Bias (low) |
|---|---|---|---|---|---|---|---|
| Chan et al. 2020 | Munich | May 2018-Apr 2019 | MAXDOAS | 30% | N/A | In-situ MAX-DOAS profiles | 17% |
| Griffin et al. 2019 | Canadian Oil Sands | Mar-May 2018 (v1.01) | Pandora (direct Sun) | 15-30% | N/A | Higher resolution profiles (10 km) and albedo | 0-25% |
| Judd et al. 2020 | New York | Jun-Sep 2018 | GeoTASO | 19-33% | N/A | Higher resolution profiles (12 km) | 7-19% |
| Verhoelst et al. 2020 | Global | Apr 2018-Feb 2020 | MAXDOAS | 37% (average), 23-51% (range) | $3.5\times10^{15}$ molec/cm² | N/A | N/A |
| Wang P. et al. 2020 | Atlantic and Pacific Oceans | 4 campaigns during Dec 2018-Jul 2019 | MAXDOAS | Negligible | N/A | N/A | N/A |
| Zhao et al. 2020 | Greater Toronto Area | Mar 2018-Mar 2019 | Pandora (direct Sun) | 24-28% (suburban/urban) +4-10% (rural) | | Higher resolution profiles (10 km) and albedo | 13-24% (suburban/urban) +14-15% (rural) |






**Table 2.** Relative and absolute differences of top-down estimate of $NO_x$ emissions using
TROPOMI for different countries in Europe in March-May 2020 (lockdown) with respect to 2019
(baseline). Ton and d denote tonne and day, respectively.

| Countries | March (%, ton/d) | | April (%, ton/d) | | May (%, ton/d) | |
|---|---|---|---|---|---|---|
| Austria | -17.25 | -63.36 | -6.64 | -23.32 | -3.77 | -12.17 |
| Belarus | -12.99 | -67.91 | -15.39 | -88.86 | -4.22 | -19.85 |
| Belgium | -32.6 | -159.34 | -27.31 | -137.95 | -28.61 | -177.56 |
| Czech Republic | -23.66 | -113.31 | -9.74 | -43.50 | -2.85 | -11.31 |
| Denmark | -10.88 | -17.91 | -13.12 | -29.58 | -8.12 | -19.84 |
| Finland | -2.88 | -5.92 | -7.70 | -18.21 | -8.82 | -19.39 |
| France | -25.35 | -547.20 | -20.46 | -467.45 | -9.29 | -198.17 |
| Germany | -7.2 | -203.63 | -24.42 | -832.93 | -9.58 | -285.57 |
| Greece | -20.56 | -77.88 | -5.32 | -19.91 | -0.88 | -3.38 |
| Hungary | -12.24 | -34.31 | -6.21 | -18.57 | -5.01 | -12.25 |
| Ireland | -12.5 | -24.55 | -7.48 | -16.83 | -3.73 | -8.15 |
| Italy | -17.81 | -270.57 | -16.14 | -252.17 | +2.37 | 34.15 |
| Netherlands | +8.86 | 28.30 | -9.71 | -38.99 | -2.27 | -10.98 |
| Norway | -2.88 | -7.69 | -8.87 | -26.91 | -3.40 | -9.52 |
| Poland | -15.05 | -246.15 | -20.02 | -342.90 | -8.34 | -126.70 |
| Portugal | -8.83 | -24.42 | -8.80 | -23.31 | -3.39 | -10.16 |
| Romania | -12.93 | -70.83 | -1.13 | -5.80 | +1.12 | 5.25 |
| Spain | -10.13 | -156.21 | -12.53 | -192.19 | -2.12 | -32.45 |
| Sweden | -6.60 | -15.17 | -8.94 | -23.12 | -6.48 | -15.85 |
| Switzerland | -8.46 | -14.15 | -8.03 | -13.23 | -13.07 | -18.96 |
| Turkey | -10.46 | -224.27 | -3.97 | -76.61 | -5.22 | -98.66 |
| Ukraine | -13.64 | -224.24 | -12.31 | -198.00 | -13.82 | -207.06 |
| United Kingdom | -14.94 | -254.82 | -19.11 | -334.27 | -14.35 | -263.89 |
| The strait of Gibraltar and Alboran Sea | -7.17 | -77.32 | -8.58 | -86.69 | -14.34 | -10.67 |
| **All** | **-13.93 ± 8.44** | **-2795.65 ± 129.67** | **-15.4 ± 6.70** | **-3224.62 ± 194.51** | **-7.72 ± 6.52** | **-1522.46 ± 92.00** |







**Table 3.** Relative and absolute differences of top-down estimate of VOC emissions using
TROPOMI for different cities in Europe in March 2020 (lockdown) and 2019 (baseline).

| Countries | March 2020 (ton/d) | March 2019 (ton/d) | Diff (%, ton/d) | |
|---|---|---|---|---|
| Paris | 54.27 | 59.66 | -9.03 | -5.39 |
| Rome | 9.92 | 10.46 | -5.18 | -0.54 |
| Milan | 1.83 | 2.59 | -29.40 | -0.76 |
| London | 27.74 | 29.32 | -5.38 | -1.58 |

**Table 4.** Reaction rates relating to the chemical pathways of ozone formation and loss over box L
(proximity of central Europe).

| Reactions | Production (P) or loss (L) | April 2020 (ppbv/hr) | April 2019 (ppbv/hr) | Net diff [a](ppbv/hr) |
|---|---|---|---|---|
| $HO_2+NO$ | P | 0.85 | 0.91 | -0.06 |
| $RO_2+NO$ | P | 0.44 | 0.41 | +0.03 |
| $NO_2+OH$ | L | 0.10 | 0.14 | +0.04 |
| $O^1D+H_2O$ | L | 0.07 | 0.08 | +0.01 |
| $O_3+VOCs$ | L | 0.01 | 0.01 | 0.00 |
| $O_3+HO_x$ | L | 0.09 | 0.08 | -0.01 |
| $J_{NO2}[NO_2]$ | P | 14.61 | 27.28 | -12.67 |
| $k_{NO+O3}[NO][O_3]$ | L | 15.11 | 28.52 | +13.40 |
| $J_{NO2}[NO_2]- k_{NO+O3}[NO][O_3]$ | N/A | -0.50 | -1.24 | +0.74 |
| Numerically solved PO$_3$ | N/A | -0.79 | -1.53 | +0.74 |

[a] A positive net difference indicates higher (lower) production (loss) in 2020 with respect to 2019.




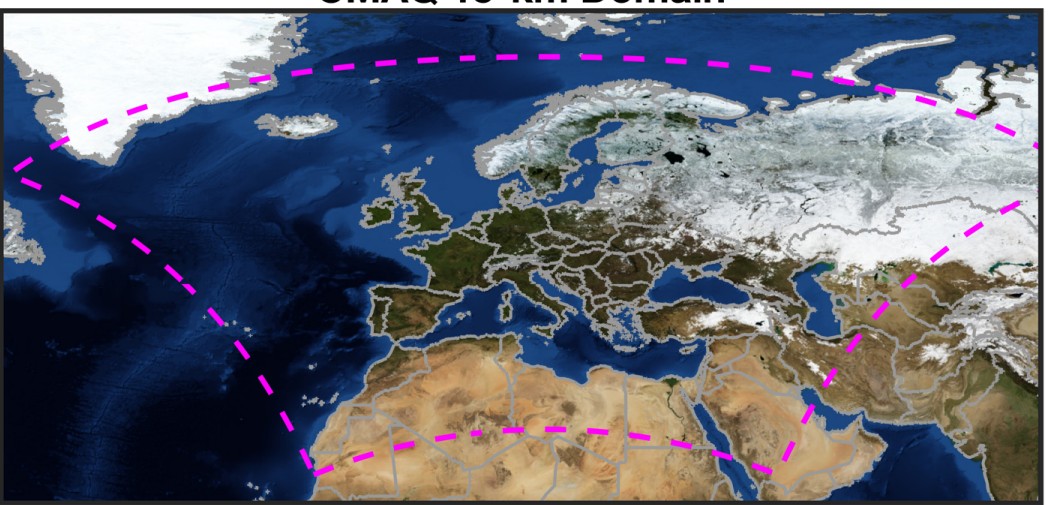


**Figure 1.** The WRF-CMAQ 15 km domain covering Europe. The background picture is based
on the publicly available NASA Blue Marble (© NASA).




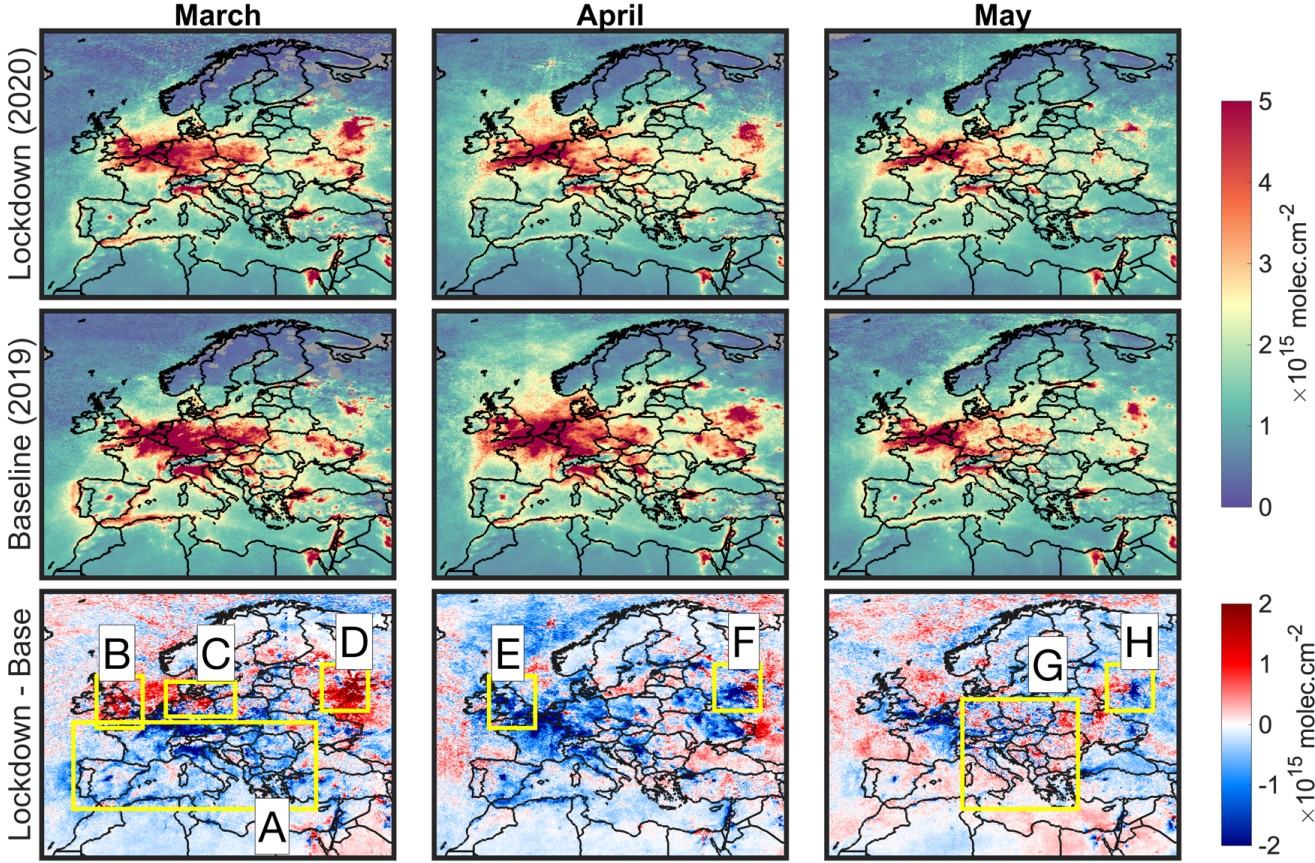

**Figure 2.** (first row) Contour maps of tropospheric $NO_2$ from the TROPOMI sensor during months
of March, April, and May in 2020 (lockdown). (second row) Same as the first row but for the
baseline year (2019). (last row), Difference of the columns in 2020 with respect to those of 2019.
All columns are corrected for the bias and their AMFs are recalculated iteratively based on the
posterior profiles derived from our inverse modeling practice. The satellite-derived columns are
subject to errors, so a direct interpretation of their magnitudes cannot be performed in a robust
manner.





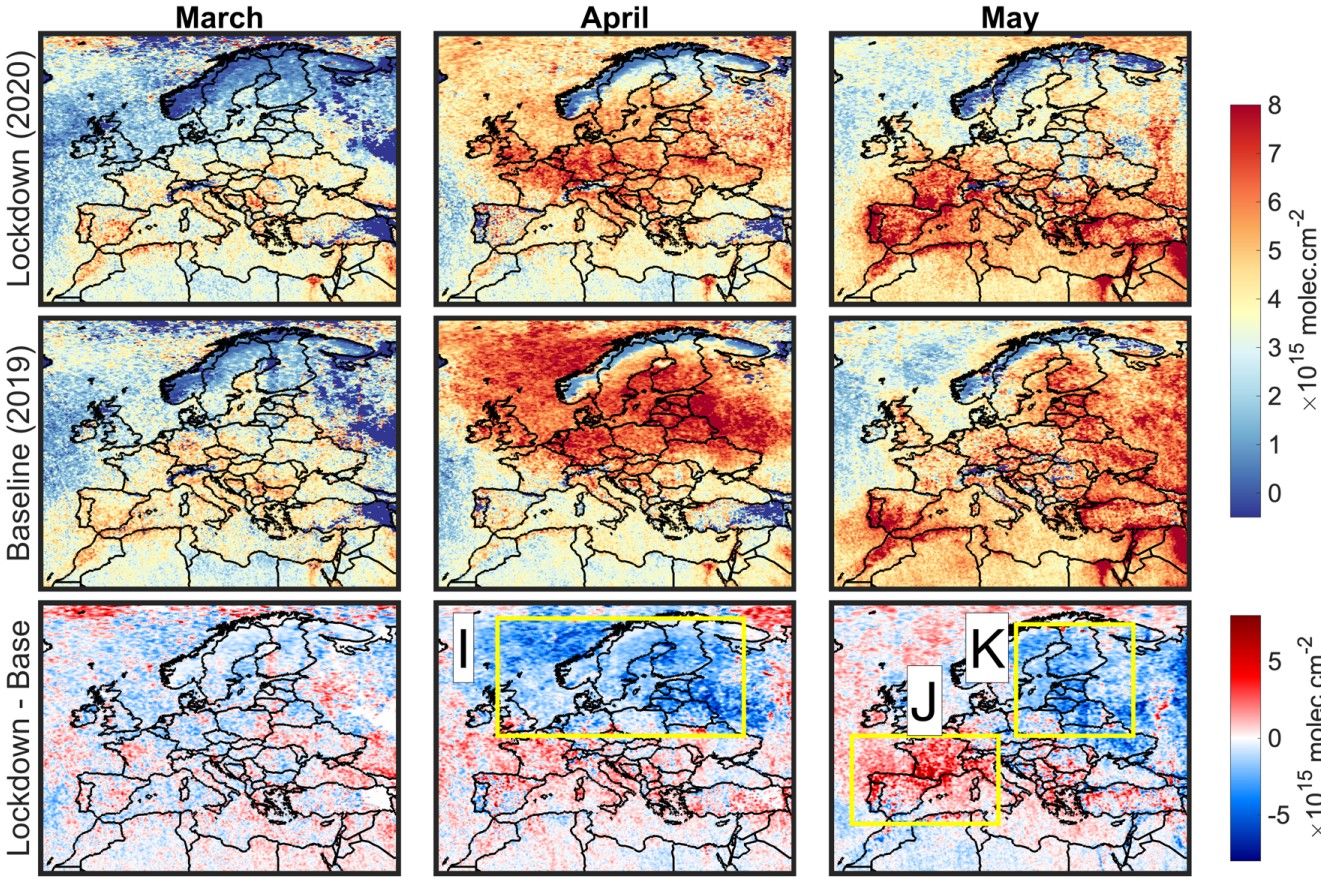

**Figure 3.** Same as Figure 2 but for the total HCHO columns.




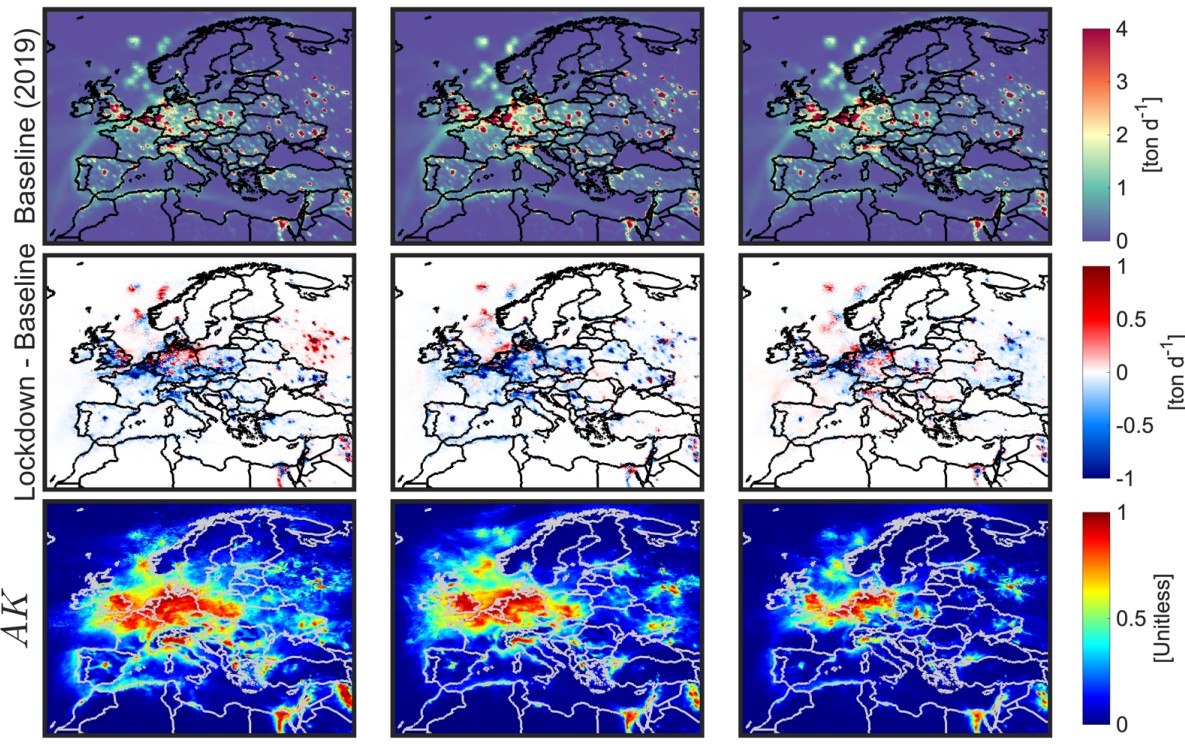

**Figure 4.** Top-down estimates of total NO$_x$ during months of March, April and May in 2019 (baseline) and the differences with respect to 2020. To infer the magnitude of emissions in 2020, the second row should be added to the first one. Both TROPOMI HCHO and NO$_2$ observations are jointly used to estimate these numbers. Averaging kernels (mean values based on both 2019 and 2020 estimates) explain the level of credibility of the estimate which is heavily dependent on the TROPOMI signal/noise ratios.






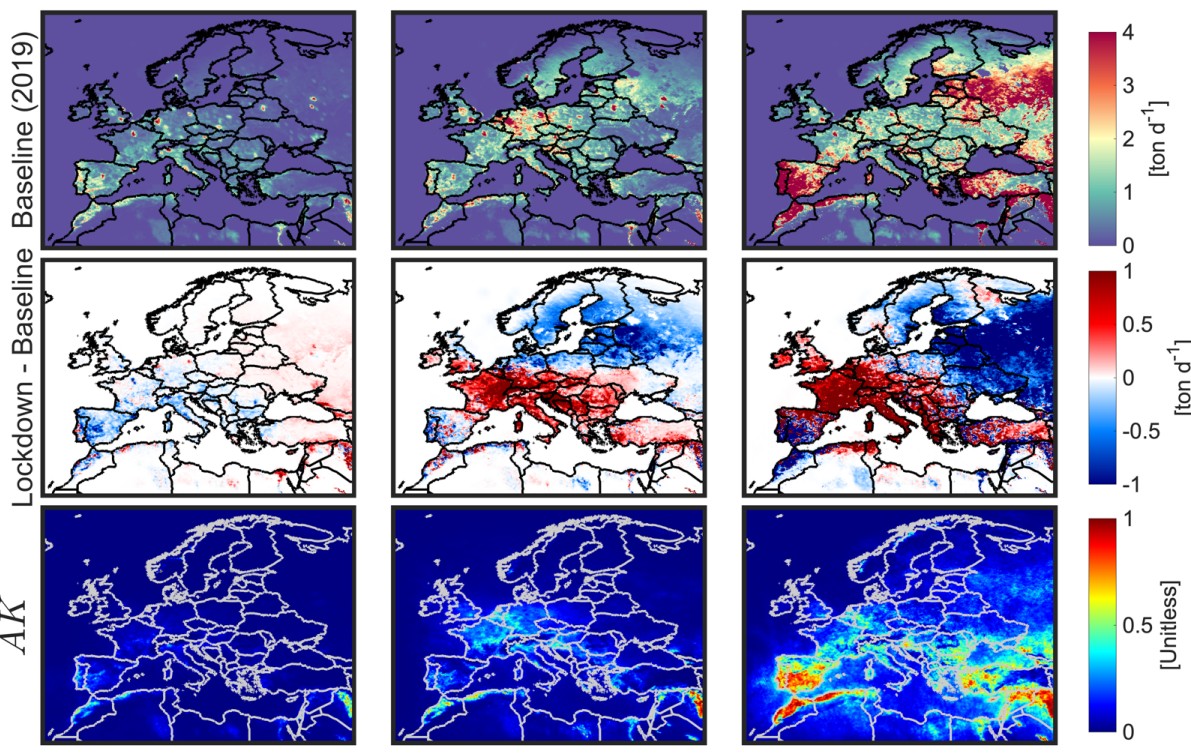


**Figure 5.** Same as Figure 4 but for the total VOC emissions.







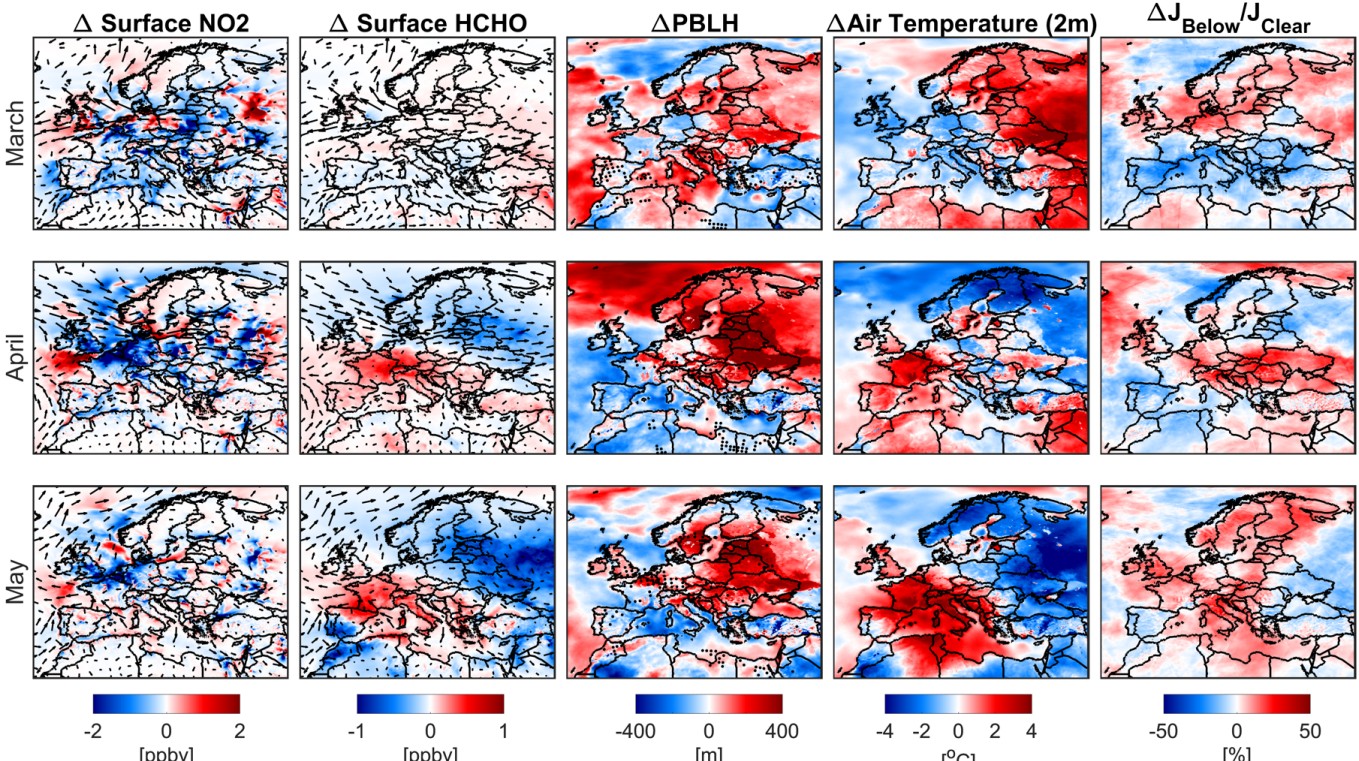

**Figure 6.** Daily-averaged differences in simulated surface $NO_2$, HCHO, surface wind vectors, PBLH, surface air temperature, and the ratio of photolysis rate below clouds to a clear-sky condition. The difference maps are computed by subtracting values in 2020 from those in 2019. Black dots overlaid in the PBLH subplot denote calm surface winds (<1 m/s).






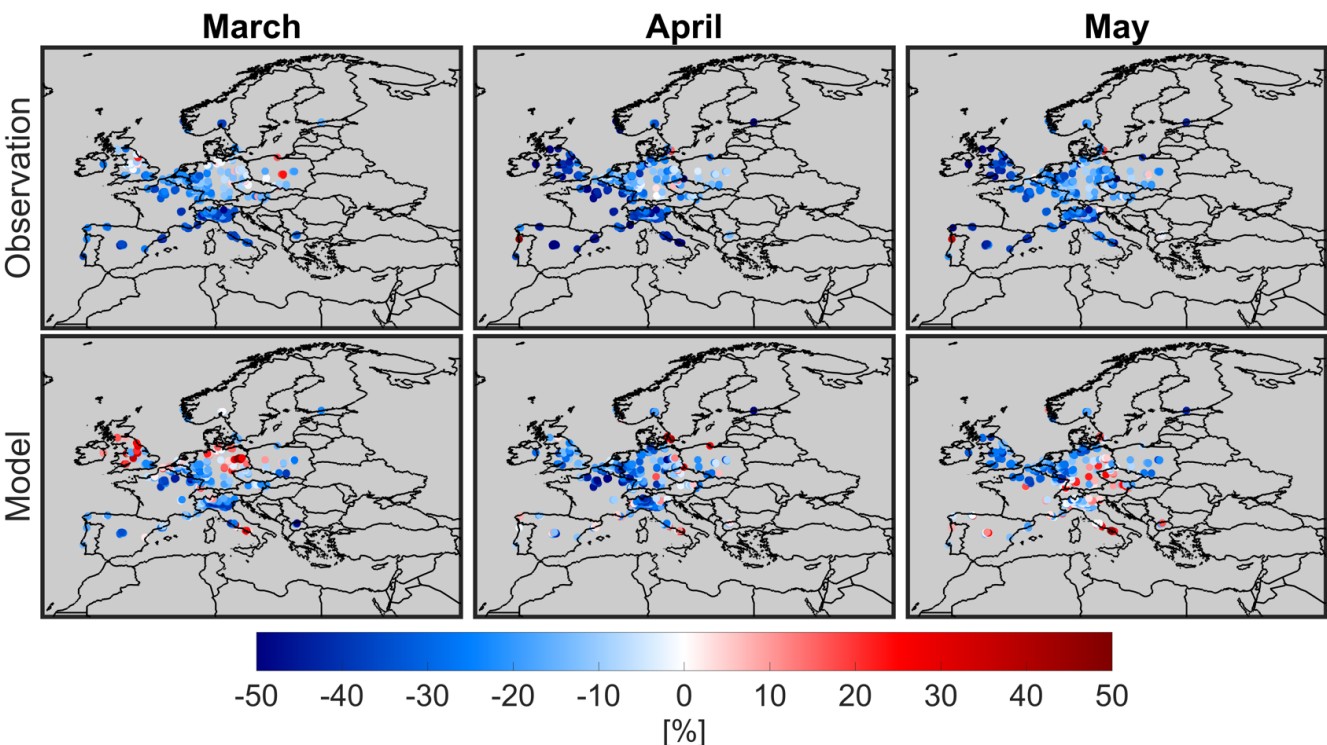

**Figure 7.** Scatter maps of relative changes in surface $NO_2$ concentrations suggested by the
European air quality network (first row), and the constrained model (second row). Results are
daily-averaged. We only consider grid cells whose averaging kernels (from the $NO_x$ inversion)
are above 0.5. Furthermore, grid cells having more than 2 stations are only included to partly
account for the spatial representivity factor. Surface concentrations are not accounted for the
$NO_z$ family interferences.





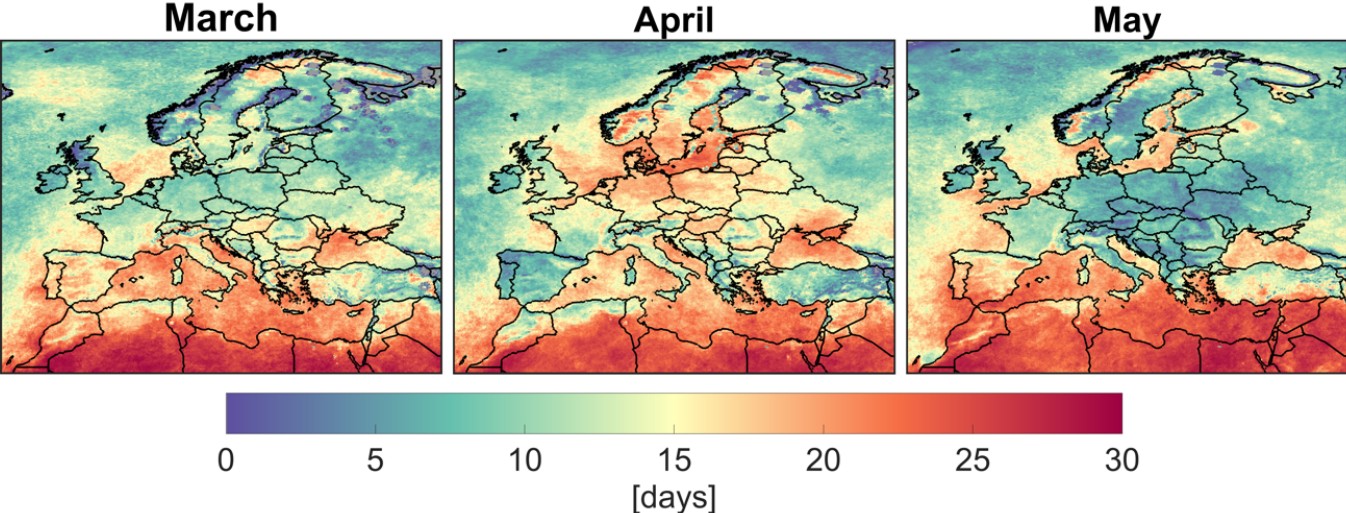

**Figure 8.** Number of good quality (qa_flag>0.75) TROPOMI tropospheric $NO_2$ days observed at
15×15 km$^2$. These numbers are heavily affected by cloudiness.







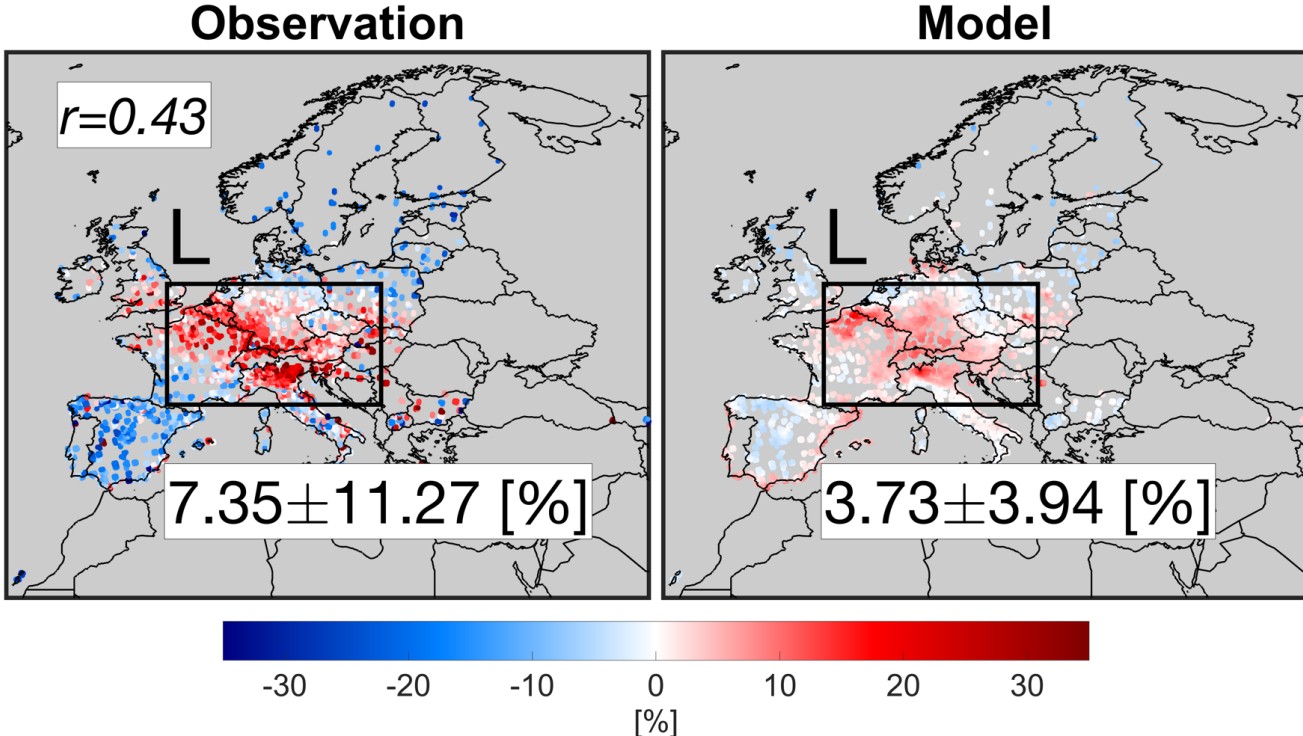

**Figure 9.** Changes in surface MDA8 ozone concentrations suggested by the observation (left),
and the constrained model (right) in April 2020 relative to those in 2019.









**Figure 10.** Surface process tendencies (hr$^{-1}$) including horizontal transport (advection plus
diffusion), vertical transport (advection plus diffusion), dry deposition, and chemistry. Positive
(negative) values mean source (sink) of ozone. These outputs are based on the constrained
model. Wind vectors are the difference.





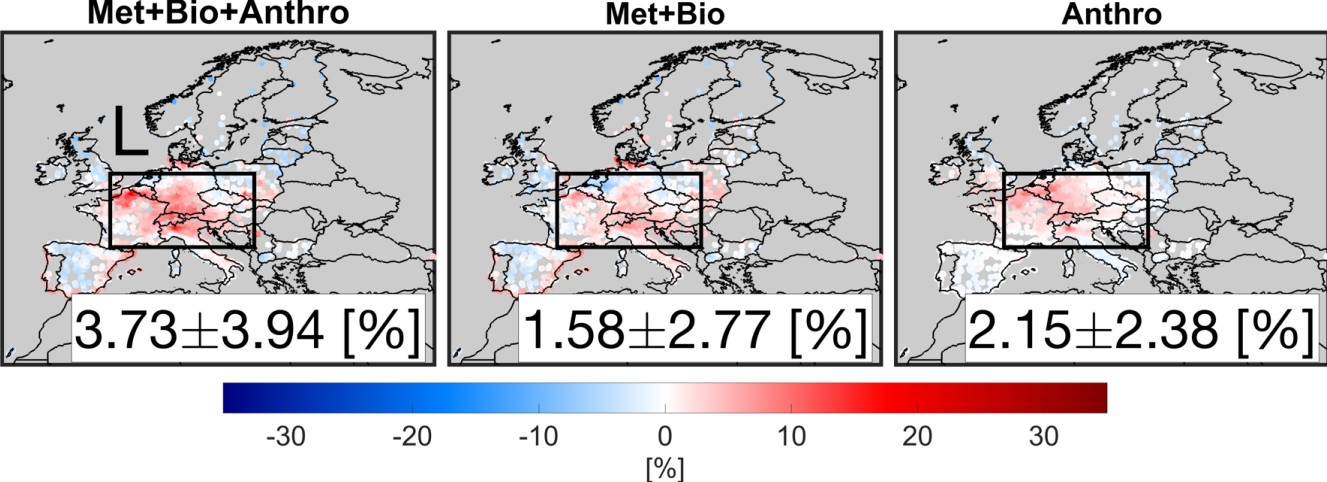

**Figure 11.** Simulated MDA8 surface ozone difference between April 2020 with respect to April
2019 including (left) dynamical meteorology, biogenic and anthropogenic emissions, (middle)
dynamical meteorology and biogenic emissions, and (right) the subtraction of the previous
scenarios isolating dynamical anthropogenic emissions.








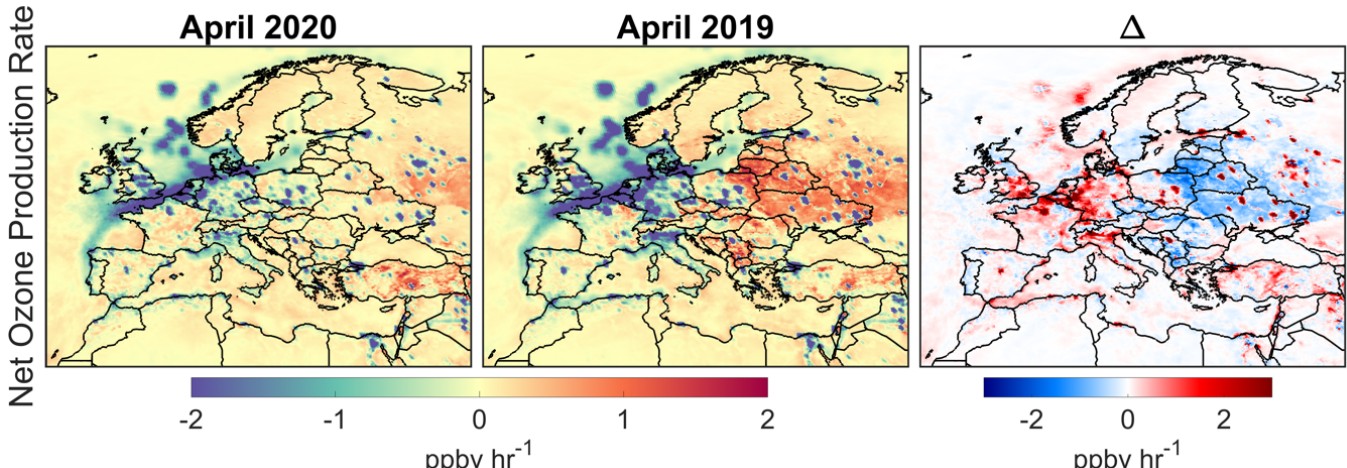

**Figure 12.** Numerically-solved net ozone production rates based on the WRF-CMAQ simulations
using constrained emissions in April 2020, 2019, and the difference. These values are over the
surface and are averaged during the MDA8 hours.





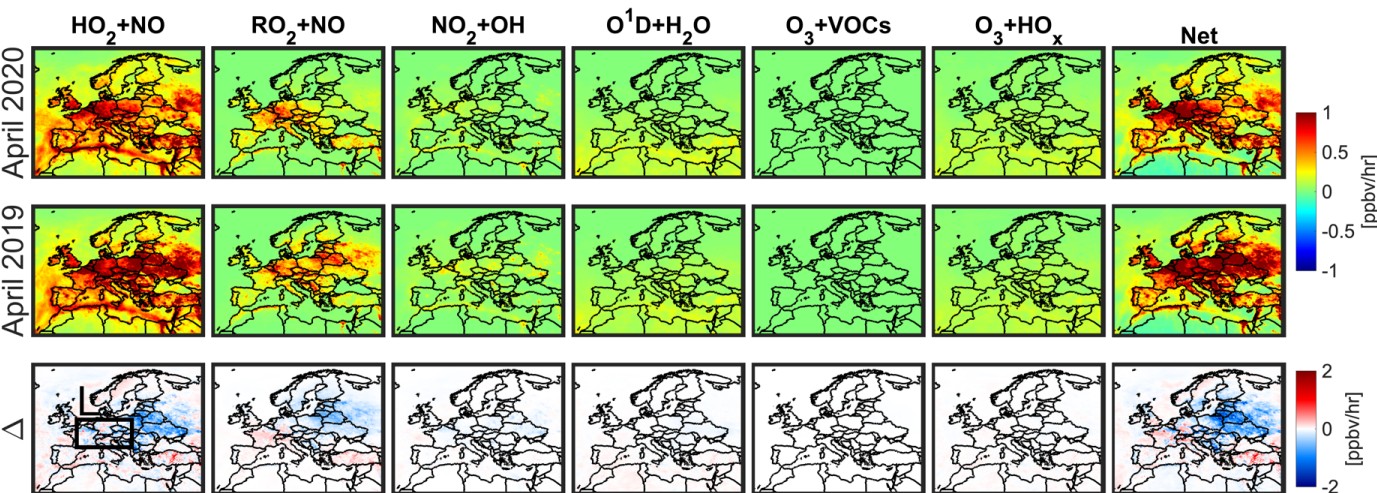

**Figure 13.** Surface chemical processes involved in equation 5 (ppbv hr$^{-1}$) pertaining to the production and loss of ozone in April 2020 (lockdown) and 2019 (baseline) during MDA8 hours.





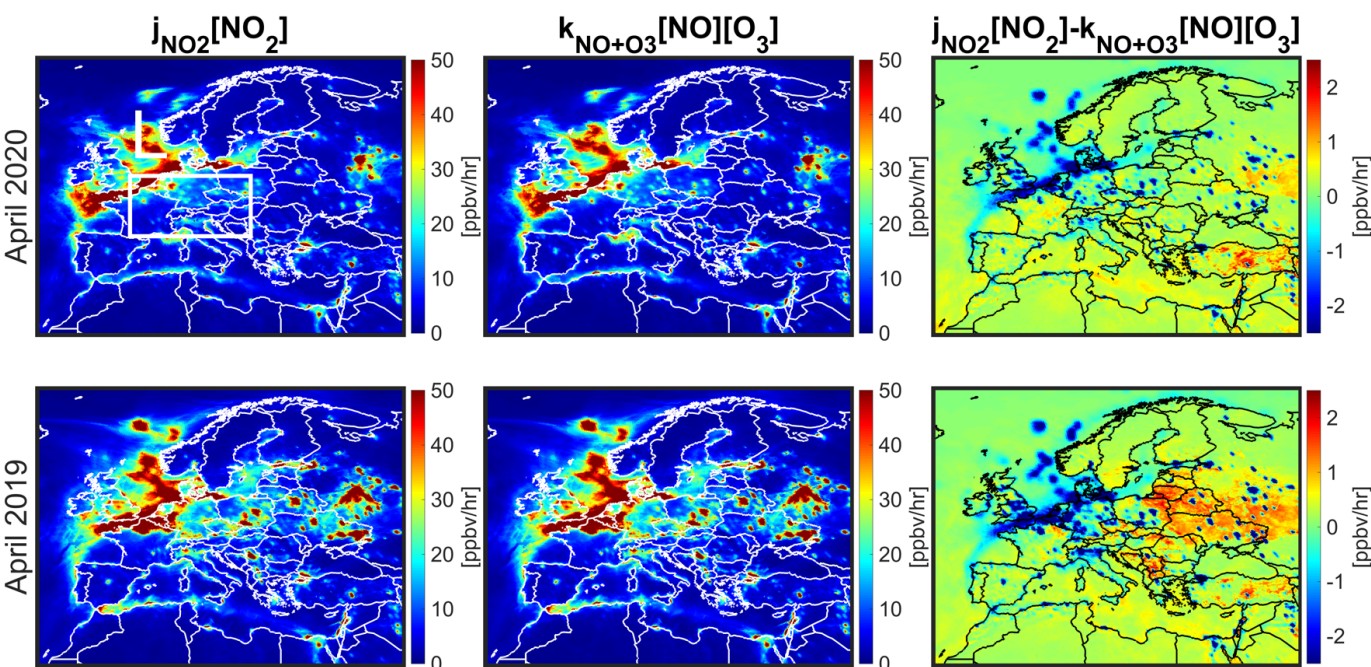

**Figure 14.** Surface chemical processes involved in equation 6 (ppbv hr$^{-1}$) pertaining to the O$_3$-
NO-NO$_2$ partitioning in April 2020 and 2019 during MDA8 hours.

