# Peer review of "Unraveling Pathways of Elevated Ozone Induced by the 2020"

_Atmospheric Chemistry and Physics, 2021_

## Author Comment (AC1)

*This paper is an extensive and intensive study of the NOx, VOC and O3 changes over Europe due to the Lockdown using WRF-CMAQ and TROPOMI data. In a sense, it is 2 papers combined into 1. The first part is the assimilation of satellite data to adjust emission inventories of NOx and VOC. The second part is a process analysis of ozone formation.*

*The paper seems rigorous and is well written, I am happy to recommend publication. Below are some minor comments that you may wish to consider.*

**We appreciate the time and thoughts this reviewer put into their review.**

*Minor Comments:*

*Table 2 and 3: Do you mean that these are the differences between 2019 and 2020 after assimilation?*

**These numbers are 2020 minus 2019 (in terms of 2020, it's normalized with respect to 2019) and yes, they are based on the inversion. We explicitly mentioned "top-down estimate" and "2020 with respect to 2019" in the caption.**

*The inconvenience of having so much material in a single paper is that the paper brushes over a fair amount of the information on the assimilation.*

*I think it would be good to show the emissions in the prior as well as the emissions in the 2019 posterior and the 2020 posterior.*

**We fully understand your concern. Emissions are usually a large source of error in CTMs. Because the accuracy of any experiment derived from models is subject to the errors of initial conditions (here emissions), we believe inversion/data assimilation procedures should be a prerequisite step in getting reasonable results from models.**

**To account for the reviewer's comment, we added both a priori and a posteriori of NOx and VOCs in the supplemental information and briefly mentioned them in the main text. We have decided to show these contour maps in the SI because we want to focus on the changes in emission happening during the pandemic.**

[Figure]

Figure S3. The a priori and the a postteriori of the total NOx emissions for the months of March (first column), April (second column), and May (last column) in 2020.

[Figure]

Figure S4. The a priori and the a postteriori of the total NOx emissions for the months of March (first column), April (second column), and May (last column) in 2019.

[Figure]

Figure S5. The a priori and the a postteriori of the total VOC emissions for the months of March (first column), April (second column), and May (last column) in 2020.

[Figure]

Figure S6. The a priori and the a postteriori of the total VOC emissions for the months of March (first column), April (second column), and May (last column) in 2019.

*The caption should clarify that these are estimates based on inversions using TROPOMI (vs. estimates based on ratios of TROPOMI data,for example).*

**Thanks, we added:**

**In Figure 11:** "Emissions used for these experiments are based on the top-down estimates."

**In Figure 12:** "the constrained emissions by the satellite data"

**In Figure 13.** "These outputs are based on the constrained model"

**In Figure 14:** "The constrained model by the satellite observations are used to derive these outputs."

**In other captions, we had stated the inversion as either "top down" or "the constrained model"**

*Fig. 10: There is a big difference in ozone production rates in Eastern Europe between rural and urban areas. This is discussed in Section 3.4. Given the importance of the question of ozone sensitivity, would it be possible to provide average values of ozone changes for NW Europe,*

*Rural East Europe and Urban East Europe? This could be integrated into a discussion of NOX/VOC sensitivity.*

**Thanks, this comment is addressed by including a figure described in the next comment.**

*Fig. 10: It would be instructive to see the corresponding average MDA8 Ozone maps. We see many difference plots, but without seeing the actual average values that these are departures from, it is hard to get a sense of what is going on.*

**Thanks, we now have included the MDA8 maps. The description of the new result follows:**

"We plot the simulated MDA8 surface ozone concentrations in April 2020 (lockdown), April 2019 (baseline), and their differences in Figure 9. Surface ozone concentrations show a strong latitudinal gradient with lower values in higher latitudes, underscoring the importance role of solar radiation in the formation of ozone. Meanwhile, the Mediterranean basin has always been prone to elevated concentrations of ozone resulting from different factors including calm weather, the transport from neighboring countries, atmospheric recirculation in coastal environments, and local emissions [Lelieveld et al., 2002]. While we observe a strong variability in the difference map, signaling various sources and sinks (discussed later), three distinctive features in 2020 in comparison to 2019 are evident: i) higher concentrations over the central Europe (up to 5 ppbv), ii) lower concentrations in eastern Europe (-2.67±1.65 ppbv) due to the 2019 biomass burning activities and larger snow cover fraction accelerating photolysis [e.g., Rappenglück et al., 2014], and iii) lower values in the Iberian Peninsula (-0.51±1.41 ppbv) [Ordóñez et al., 2020]."

[Figure]

**Figure 9.** Simulated surface MDA8 ozone concentration using the constrained model in the month of April 2020 (lockdown), April 2019 (baseline), and their difference.

*Technical Comments:*

*Fig. 3: "Estimate" not "Estmate"*

**Corrected.**

*Fig. 4: I think you mean Delta X = X_2020 – X_2019 – this is what you have in the text and elsewhere.*

**Corrected.**

*Fig. 5: Could zoom in on the area with data, which would make the figure more legible.*

**We want to be consistent throughout the paper about the location of case study.**

*Fig. 10: There is room to spell out Delta in the title. I think this would make it easier for the casual reader to follow, eg. Delta O3P = O3P_2020 - O3P_2019*

**Thanks, added.**

---

## Author Comment (AC2)

*Souri and co-authors present an analysis of lockdown-induced changes in NO2, HCHO, and O3 over Europe based on a data assimilation approach involving TROPOMI measurements (NO2, HCHO) and the WRF-CMAQ model. An advantage of the approach is that it explicitly accounts for meteorological influences and so forth in assessing the causes of AQ changes during the COVID period.*

*The paper topic is suitable for ACP and will make a useful contribution to the literature. There are some methodological and science comments and questions that I feel should be addressed before publication; these are listed below. In a number of cases things are described in a confusing way and need to be clarified. Finally, in many places the writing can be clearer or is overly wordy. For example, many of the paragraphs are about a page long and cover multiple topics, which really does not help communication. Once these issues are addressed I recommend publication in ACP.*

**We thank you taking the time to provide such detailed feedback.**

=====================================

*Science comments. Numbering refers to line numbers.*

=====================================

*1. The abstract is very long (almost 500 words!) which partly defeats the point of an abstract. I suggest reducing it by approximately half.*

**We have shortened the abstract.**

*56. "Earth's atmosphere has exponentially become more polluted during previous decades". Too vague / sweeping. What do you mean by "previous decades"? Some parts of the world have become significantly less polluted (for PM and ozone) over the last 2-3 decades. Elsewhere I do not think the word "exponentially" is necessarily accurate.*

**We agree that we wrote this part a bit too general. We have changed it to:**

**"**Earth's atmosphere has substantially become more polluted since the industrial era in comparison to its original environmental condition [Li and Lin, 2015], thus any abrupt hiatus in anthropogenic (man-made) emissions …"

**We simply want to emphasize on the magnitude of anthropogenic emissions being so large, which is essentially the main motivation of this study.**

*78. "whereas the concentrations of several secondarily formed compounds such as ozone increased due to emissions and/or meteorology". This is not universally true; e.g., see Bekbulat et al., https://doi.org/10.1016/j.scitotenv.2020.144693.*

**Thanks. We added the paper and wrote***: "whereas the concentrations of several secondarily formed compounds such as ozone behaved in non-linear ways due to emissions and/or meteorology."*

*767. For reports, please list references in a way that readers can readily access them, e.g. with a doi or persistent URL.*

**Added.**

*138-139. RMSE already has a definition; what you are reporting here is not the RMSE.*

**We changed it to "*dispersion*" in the text and the table. We defined the dispersion ("as half of the 68% interpercentile").**

*148-150. "There are challenges…" Unclear what you mean here. Please rewrite for clarity.*

**We made a new paragraph and rewrote this part:**

*"Directly incorporating these numbers into an inversion model is challenging, mainly because of spatiotemporal variability in the satellite errors. Ideally, the relationship between errors and retrieval inputs (e.g., albedo, scene radiance, profiles, etc.) would be used as an additional cost function in the inversion, commonly known as variational bias correction [e.g., Auligné et al., 2007]. In the absence of such relationships, we use the biases reported in the validation studies. "*

*153. "we uniformly scale up NO2 pixels by 25% based on the low bias determined by Verhoelst et al. [2021] while considering the potential reduction in the bias through the use of higher spatial resolution trace gas a priori profiles." Not clear what is meant here by "while considering". How was this considered? Do you mean it was considered by choosing 25% rather than the median value of 37% reported by Verhoelst? Or is something different being implied here? Please clarify.*

**Please see the next comment.**

*153. The choice of a 25% bias correction for NO2 seems a bit arbitrary. As I understand it, the argument being presented is as follows: "the bias was reported previously to vary from -23 to -51%, with a median of -37%. But the use of higher-resolution shape factors here should reduce the bias. So, we use +25%." I agree that higher-resolution shape factors will reduce the bias, but there is no quantification of that effect here, so the 25% value seems to be pulled out of a hat. There is also the fact that the TROPOMI bias was shown previously to vary between rural and polluted environments, but this is not accounted for here. Overall, there needs to be either a more rigorous justification for the bias correction being employed, and/or some sensitivity analysis to quantify the degree to which this assumption affects the results.*

**Thanks for your detailed feedback. We added the justification:**

In the case of NO$_2$, we uniformly scale up the satellite tropospheric columns by 25%. This bias estimate is derived by first assuming a 37% low bias in the columns over polluted regions as reported by Verhoelst et al. [2021]. In turn, this low bias can be mitigated somewhat by the application of high spatial resolution profiles in the air mass factor calculation, such as the ones used in this study. Table 1 summarizes the results from several TROPOMI validation studies at specific locations that calculated NO$_2$ using model profiles with higher spatial resolution than the operational TROPOMI (1°×1°) profiles (see Table 1 columns "Modification" and "Modified Bias"). In these studies, modified columns show increases ranging from 0 - 25%. Based on these results, we assume a low bias of 37% can be mitigated by ~12% through the use of high spatial resolution profiles, for a resulting total low bias of 25%. This bias is likely not valid over pristine areas, where validation studies show lower biases in TROPOMI NO$_2$ [Verhoelst et al., 2021, Wang et al., 2020, Zhao et al., 2020]; nonetheless, we previously observed in Souri et al. [2020a] that the low signal-to-noise ratios of those column amounts resulted in small changes in the top-down emissions.

**We also elaborated the error characterization:**

We assume the errors of observations originate from two main sources: i) the precision error provided with the data (e$_{precision}$) and ii) a fixed error estimated from comparisons to in-situ measurements (e$_{const}$). Mathematically, the final error is:

$$e_O^2 = e_{const}^2 + \frac{1}{n^2} \sum_{i=1}^{n} e_{precision,i}^2 \qquad (1)$$

where $n$ is the number of samples for a given grid and e$_{const}$ equals to $1.1\times10^{15}$ molec/cm$^2$ ($<6\times10^{15}$ molec/cm$^2$) in clean regions and $3.5\times10^{15}$ molec/cm$^2$ ($>=6\times10^{15}$ molec/cm$^2$) in moderately to highly polluted regions based on the wide ranges reported in Verhoelst et al. [2021] ($3-14\times10^{15}$ molec/cm$^2$ for moderately to highly polluted region).

*183-187. A similar comment applies to HCHO. I appreciate that the authors pay close attention to uncertainty and bias in the satellite data. But in the end the employed corrections are chosen a bit haphazardly from the range of reported biases. How can this choice be better justified, or if it is necessarily a little arbitrary, how can the impact of that assumption be quantified?*

**Those numbers are estimated based on the Figure 3 in Vigouroux et al. [2020]. Do we think this is the best way of accounting for the errors? No. In fact, our former studies exclusively focusing on AQ campaigns validated the measurements with respect to independent measurements; similar to what we currently showed in terms of MODIS /AERONET AOD.**

There are two ways to improve the error characterization: i) the number of FTIR, Pandora, and MAX-DOAS observations would grow and they become permanent stations and publicly available (this is not a complaint; it is an utopia every quantitative study dealing with satellite is dreaming of) and/or ii) a more systematic approach would be taken to parametrize the errors in the validation studies; this can be done by establishing a relationship between errors and the inputs used for the retrievals. Say, one might find varying biases by changing cloud fractions such that you can reproduce the bias given a defined cloud fraction. Ultimately, a variational bias correction method [Auligné et al., 2007, https://rmets.onlinelibrary.wiley.com/doi/abs/10.1002/qj.56] can be adopted to engage those relationships (along with their uncertainty) in the inversion framework.

**We modified this part:** "We assume the constant term of errors ($e_{const}$) to be equal to 4% of HCHO total columns based on Vigouroux et al. [2020]."

*156. "we set the RMSE to 1.1x10^15 molec/cm2 in clear regions and 3.5x10^15 in moderately to highly polluted regions." This is confusing because at this point in the text we don't know what is meant by "set the RMSE". We learn later that these values will be used to populate the error covariance matrices for the inversion; please clarify that here so the reader understands what is happening.*

**Thanks, we changed it. Please see our earlier comment.**

*199-202. Does this mean that you only use the dark blue product? Please clarify.*

**Yes, we clarified it in the text.**

*The TROPOMI retrievals do not account for aerosols in the scattering weights. Yet I presume that aerosol loadings over Europe changed between the COVID and reference period. To what degree does this bias the retrieval differences and therefore the NO2 and HCHO comparisons between these periods?*

**This is an important question but not really a big issue here. Based on a very recent study from our group studying the aerosol impact on AMFs using the OMI aerosol product, Jung et al. [2019] observed negligible differences in the amount of HCHO VCD (<10%) over Europe in March-May after accounting for the impacts (including AOD, SSA, and aerosol height). The same thing applies to NO2 [Cooper et al., 2019]. The MODIS deep blue channel indicates that changes in AOD at 550 nm between 2020 and 2019 (in March-May) are <0.15:**

[Figure]

**MODIS AOD 550 nm (2020-2019)**

We speculate that the aerosol complications are more relevant for biomass burning areas, regions with large AOD like China, and retrievals that are extremely sensitive to even small amount of aerosols such as CO2 and CH4.

**To account for the reviewer's comment:**

"Aerosol effects on the scattering weights are not taken into consideration. Based on radiative transfer calculations and satellite-based aerosol products, Jung Y. et al [2019] and Cooper et al. [2019] observed small changes (<10%) in AMFs with and without considering the aerosol impacts in Europe in springtime. This tendency likely results from a low aerosol optical depth."

*236. "We nudge moisture, wind and temperature fields toward the reanalysis data used only outside of the PBL layer." Wording is unclear, as is the reason for doing this. Please clarify.*

**We included the purpose:**

"To minimize the deviation of the model from the reanalysis data, we turn on the grid nudging option with respect to wind, moisture, and temperature only outside of the PBL region. The inclusion of this option outside of the PBL is because we do not want the coarse reanalysis data washes out the relatively high-resolution dynamics".

*239. "Extensive model evaluations based upon surface observations show a striking correspondence". The model temperature bias appears to be 50% smaller in 2020 than 2019 (0.8 vs 1.2 degrees). Does this have any impact on the model interpretation of changes between years? For example, assessing changes in anthropogenic VOCs relies on distinguishing changes in biogenic emissions which depend exponentially on temperature.*

**The absolute difference only matters which is only 0.3º (-0.9 vs -1.2) on average. We were able to derive changes in anthropogenic VOCs (due to the chemical feedback of NOx on VOC) only in March where biogenic emissions are so low such that the difference in the temperature bias is negligible. This is because the average temperature and solar radiation are so low to have any significant impact on biogenic emissions. Please see the new Figure 4 including the biogenic fraction.**

*239. PBL height is a major factor for model performance in simulating AQ-relevant species. How well does the simulation capture measured PBL depths over your domain?*

**Thanks for your comment. It is indeed important. Reliable PBL observations are rare and are usually estimated based on signal processing methods applied on LiDAR attenuated backscatter values. While we did have this type of information during NASA-funded campaigns on aircrafts and over the ground, we really could not find any publicly available data over this region.**

*242. Please state the time resolution at which you are optimizing emissions. I guess there is a single 3-month mean value being derived for each grid cell but unless I missed it I don't think this is stated anywhere.*

**We had mentioned it in the inversion part, but now we elaborated to:**

 *"The inversion window is monthly meaning we have three separate correction factors in months of March, April, and May"*

*250. Please state how the Jacobian is calculated. Is there a finite difference run for every model grid cell, each tracer, and each iteration?*

**Each iteration, uniformly to all grid cells (20%. Perturbation to each grid cell). One for NOx and one for VOC. Perturbing individual VOCs is an overkill because the problem is under constrained. Perturbing each individual grid cell while keeping others constant will require 369979 forward simulations, each takes around ten days using 250 cpus. Given the computation resources we have here, that would take around 5068 years (369979 * 2. (nox and voc) * 10 days / 365 days / 2 (two runs at same time with 500 cpus). We obviously need the adjoint of the model to expand the number of state vectors which is not updated for the newer versions of CMAQ.**

"and $K_i$ $(= K(\mathbf{x}_i))$ is the Jacobian matrix calculated explicitly from the model using the finite difference method by perturbing separately $NO_x$ and VOC emissions by 20%. The perturbations are applied for each iteration."

*252. "In terms of the prior errors, we use the numbers reported in Souri et al. [2020a]." Since this is an important aspect of the inversion please briefly summarize here.*

**Sure***: "The prior errors in anthropogenic NOx and VOCs emissions are set to 50% and 150%, respectively. In terms of the biogenic emissions, the errors are set to 200% for both NOx and VOCs."*

*257. "here we iterate Eq 1 3 times." How do you know this is sufficient? As you know the emission-concentration relationship for NOx in particular is highly non-linear. Do you employ a test for convergence?*

**This number is based on two factors: first, we used the same number in our previous study [Souri et al., 2020a] over East Asia for which we found satisfactory results against satellite/observations indicating a reasonable convergence. Non-linear chemistry was a problem in that region due to large oxidation capacity and emission rates. Second, we defined certain time and computational resources for this project. The analytical inversion is extremely time (and space) consuming at a continental regional-scale.**

*275. "faster vertical mixing due to larger sensible fluxes (more diluted columns due to stronger advection in higher altitudes)". This is a little convoluted. Faster vertical mixing by itself wouldn't change the column amount, and faster winds during summer (really?) would only smear the columns.*

**If we rule out the advection, the vertical mixing does change the amount of column through changing the dry deposition rates (please see the dependency of dry deposition and vertical diffusion in Figure 10). In several places (such as the southeast US), winds are dominantly stronger within 700 hpa in *summer* due to certain synoptic conditions. Some examples based on Souri et al., 2017 are shown below (summer top, spring bottom). Please note that the Bermuda high is shifted westwardly increasing the pressure gradient between the Midwest low and itself ultimately resulting in strong low jet streams.**

[Figure]

Averaged simulated surface O₃ with BB at 06-18 CST in July 2012

Averaged simulated surface O₃ with BB at 06-18 CST in April 2012

**If the receptor is located at *A* and winds are stronger vertically, the receptor *A* will experience lower concentrations of NO2 columns. We cannot change the location of the receptor at the same time.**

**To better word this, we changed the sentence to: "**faster vertical mixing due to larger sensible fluxes (more diluted columns for a given receptor due having a greater chance of experiencing stronger winds in higher altitudes),…"

*277-280. Wording is quite awkward here.*

   **We reworded: "**This sequential decline of $NO_2$ obscures the quantitative interpretation of the satellite observations in two ways: first, as noted by Silvern et al. [2019], the free tropospheric background $NO_2$, which are highly uncertain, becomes comparable to those located in near-surface, and second, the relatively lower signal-to-noise ratios reduce the amount of information obtained for $NO_x$ emission estimates (discussed later). "

*281. "pronounced decreases", please clarify that you mean in 2020 vs 2019.*

**Added.**

*282. "In contrast, we see negligible reductions…" actually some of the regions mentioned seem to show a clear March increase.*

**We changed to increase.**

*296. "suggests an abrupt hiatus in the ongoing reduced NOx emissions". Unclear if this means the emissions went into hiatus or the reduction went into hiatus.*

**We changed is to**: *"In May, the anomaly of the tropospheric $NO_2$* suggests that the reduction in $NO_x$ emissions abruptly experiences a hiatus in central Europe (box G)."

*317. "but also stems from the fact that isoprene reactivity significantly increases by rising temperature [Pusede et al. 2015]." This is a bit oddly worded; I think you simply mean that OH is increasing seasonally along with isoprene emissions.*

**Thanks, we changed it to:** *"This signal is not only induced by the inherent temperature-dependency of biogenic isoprene emissions, but also stems* from faster isoprene oxidation through higher levels of OH [Pusede et al. 2015]."

*335-337. These evaluation statistics should be displayed in SI in a table or figures.*

**Thanks, we included relevant figures in the SI.**

"We observe a large improvement (31-45%) in the bias associated with simulated surface $NO_2$ using the posterior emissions compared to the surface measurements in many places around Europe with an exception to northeastern Germany where TROPOMI $NO_2$ observations deviates the model from the measurements (Figs S7, S8, S9 and 10). The improvements in correlation were minimal indicating that the prior location of emissions are well known."

[Figure]

Figure S7. Comparison of daily-averaged surface NO2 observations (circles) against the simulated model in different regions around Europe in March-May 2019 (baseline). The first row uses the prior emissions whereas the second is based on the top-down emissions constrained by the satellite observations through an analytical non-linear inversion.

[Figure]

Figure S8. Similar to Figure S7 but in different areas.

[Figure]

Figure S9. Comparison of daily-averaged surface NO2 observations (circles) against the simulated model in different regions around Europe in March-May 2020 (baseline). The first row uses the prior emissions whereas the second one is based on the top-down emissions constrained by the satellite observations through an analytical non-linear inversion.

[Figure]

Figure S10. Similar to Figure S9 but for different areas.

*341-344. "However, in practical terms, the magnitude of these anomalies is not as drastic as the ratio of observation to model ratio because of the consideration of observational errors and chemical feedback [Souri et al., 2020a], which always leaves some doubt about the practicality of direct mass balance methods." I am unsure what the authors are trying to say here.*

**We decided to remove this sentence because we had already emphasized this non-linear pattern in more detail in our previous studies.**

*358-360. The optimization naturally improves the simulation of HCHO with respect to TROPOMI, that is the whole point of the optimization. Does it also improve the simulation with respect to independent observations?*

**The HCHO observations used for the inversion are only based on TROPOMI. As far as we know, in-situ HCHO measurements are not available (or at least publicly available) during the period. Publicly available MAX-DOAS observations do not cover this period ([http://uv-vis.aeronomie.be/groundbased/QA4ECV_MAXDOAS/index.php](http://uv-vis.aeronomie.be/groundbased/QA4ECV_MAXDOAS/index.php)).**

*358-378. This paragraph is really unclear; I had to read it multiple times to try and parse what is being argued. It sounds like you're arguing that the chemistry changed the emissions. Please rewrite.*

**Thanks, we decided to remove this discussion based on Reviewer #2.**

*395. "Horizontal transport (shown as wind vectors) plays a critical role in explaining the spatial variations in emissions downwind." Why would wind affect the emissions?*

**We removed the discussion about Figure 6.**

*397-418. This section is all quite speculative and unconvincing. It does not appear that there is much required information conveyed here, recommend deleting.*

**We removed it.**

*409. "This in turn will provide an opportunity for the volume of air to become dispersed". Poor wording. The VOC lifetimes do not affect how a "volume of air is dispersed".*

**This part has been removed.**

*422. "Unfortunately we limit the analysis to NO2 due to the lack of routinely measured HCHO observations." The HCHO data are ultimately being used to constrain VOC emissions; so are there VOC measurements that can be used for this purpose?*

**The HCHO observations used for the inversion are only based on TROPOMI. As far as we know, in-situ HCHO measurements are not available (or at least publicly available) during the period. Publicly available MAX-DOAS observations do not cover this period ([http://uv-vis.aeronomie.be/groundbased/QA4ECV_MAXDOAS/index.php](http://uv-vis.aeronomie.be/groundbased/QA4ECV_MAXDOAS/index.php)). In case the link does not work during the review process:**

[Figure]

**QA₄ECV MAXDOAS reference data sets**

FP7-SPACE-2013-1/Project No 607405

**Introduction**

This webpage provides access to the QA₄ECV NO₂ and HCHO MAXDOAS reference data sets. The QA₄ECV MAXDOAS product consists of harmonized multi-year time series of NO₂ and HCHO tropospheric vertical column densities at a selection of 10 stations corresponding to urban, sub-urban, and rural conditions. The list of stations is the following:

| Station | Lat, Long | Class | Data Source | Time coverage |
|---|---|---|---|---|
| De Bilt/Cabauw (NL) | 52°N, 5°E | Sub-urban | KNMI | 03/2011 - 11/2017 |
| Uccle (BE) | 50°N, 4°E | Urban | BIRA | 04/2011 - 06/2015 |
| OHP (FR) | 44°N, 5.5°E | Rural | BIRA | 02/2005 - 12/2016 |
| Xianghe (CHN) | 39°N, 117°E | Sub-urban | BIRA | 04/2010 - 01/2017 |
| Bujumbura (BU) | 3°S, 29°E | Sub-urban | BIRA | 01/2014 - 12/2016 |
| Bremen (DE) | 53°N, 9°E | Urban | IUP-UB | 02/2005 - 12/2016 |
| Nairobi (KEN) | 1°S, 37°E | Rural / Urban | IUP-UB | 01/2004 - 11/2014 |
| Athens (GR) | 38°N, 23°E | Urban | IUP-UB | 09/2012 - 10/2016 |
| Mainz (DE) | 50°N, 8°E | Urban | MPIC | 06/2013 - 12/2015 |
| Thessaloniki (GR) | 41°N, 23°E | Urban | AUTH | 01/2011 - 05/2017 |

**Table 1. List of QA₄ECV MAXDOAS stations**

*440. "The surface measurements reinforce the less pronounced reduction in NO2 in northern Germany and UK, although the magnitudes are not as large as those suggested by the model." This is not clear from the figure. For example the observations suggest that decreases over the UK in April and May are quite large compared to the rest of Europe.*

**We changed it to be more specific:** "The surface measurements in March reinforce increases (or negligible changes) in NO₂ in northeastern Germany and UK, although the magnitudes are not as large as those suggested by the model."

*492-496. "This apparent discrepancy is caused by the differences in boundary and initial conditions which are not quantifiable by the process analysis and would require additional sensitivity test." Is it just the ICs and BCs, or is it that these processes being examined are not strictly independent and additive?*

**From the modeling perspective, the processes are strictly independent. As a matter of fact, if one knows the exact IC and BC contributions and add each chemical/physical components incrementally, they will be able to reproduce the concentration of ozone at a given time and location, very similar to what simulations output. Mathematically speaking, ozone is given by:**

**O3 = adv + diff + chem + emiss + drydep + cloud_chem + aerosol_chem + IC**

**Ruling out emiss, cloud_chem and aerosol_chem components, we should be able to know the difference in O3 between two years as long as IC are known. IC for this case is the beginning of April.**

*Equation 6 is incorrect (the wrong rate constant is indicated).*

**Thanks, corrected.**

*544. "This analysis strongly coincides with Lee et al. [2020] and Wyche et al. [2021] who observed roughly constant O3+NO2 concentrations over the UK before and during the lockdown 2020." With this in mind, why not actually just show the modeled Ox = O3 + NO2 change (and measured change too, if available)? This seems like the most direct way to make this point.*

**Ox is highly influenced by transport. It's difficult to pinpoint which areas have contributed to less/more Ox for a given region. We did not only want to see the general conclusion about the "partitioning between NO-NO2-O3", but also to demonstrate the rates of P(O3). The current analysis adds more information: i) where the NO-NO2-O3 partitioning is mostly occurring which is tied to titration and ii) how fast those rates are changing.**

**Additionally, those studies should have also included "NO" because Ox is not strictly defined as NO2+O3. But the measurements of NO in Europe are lacking.**

*572. "The reduced anthropogenic VOC emissions were a result of two key assumptions: the reduced NOx emissions in NOx-rich areas increased HCHO made from VOCs (evident in larger Jacobians derived from the regional model), and TROPOMI HCHO suggested a negligible difference in HCHO concentration between the two years." Again the wording here is really confusing. It appears to be arguing that changes in NOx emissions and in the ensuing chemistry changed the actual VOC emission rates. I think I know what is meant (i.e., that these factors change the emission rates that one infers for a given HCHO level) but it really needs clearer description.*

**This part has been dropped.**

*Minor edits.*

*73. "atmospheric composition" not "compositions"*

**Corrected.**

*78. "particulate matter"*

**Corrected**

*157 and 185. "clean regions" and "clean areas" rather than "clear"*

**Corrected**

*177. "mainly located" or "predominantly located"*

**Corrected.**

*437. "by only considering grid cells"*

**Corrected.**

---

## Author Comment (AC3)

*Major comments*

*This manuscript presents an inverse modeling study of NOx and VOC emissions over Europe in the spring of 2019 and 2020, based on TROPOMI NO2 and HCHO data. The focus is on the differences between the two years and on the detection of Covid-related effects. In agreement with previous studies, large NOx emission decreases are derived over most European countries in April 2020. In March and May, however, the picture is less clear, with some regions (e.g. large parts of UK and Germany in March) showing large emission increases in 2020 (Figure 4). Those changes are unrealistic and are contradicted by comparisons with surface NO2 measurements (Figure 7). The authors present the disparity between regions as a consequence of the different timing of Covid-lockdown measures over different regions of Europe, but the discussion is poor and does not present concrete arguments for the inferred patterns. Those patterns are probably related to the inability of the model to match the observed NO2 column distribution, in particular (but not only) over N-W Europe (see Figure S3). The prior model strongly overestimates NO2 over northern Germany and strongly underestimates NO2 in southern Germany and in many other regions. It would be most enlightening to examine the top-down emission increments for each year and month (and not just the differences between the two years). I suspect there will be huge disparities within several countries, especially Germany. It is highly unlikely that bottom-up emission inventory could perform so badly in terms of spatial*

**We agree with the last sentence saying that the *location* of NOx emissions inventories are relatively well known, but not necessarily that the *magnitudes* of NOx emissions provided by crude bottom-up estimates with infrequent updates are true. We need both to be right to get the simulation correct. Therefore, using a top-down estimate is useful here. The differences between NO₂ concentrations (i.e., lockdown vs baseline) simulated by the model would have been minimal, if we had not leveraged the inverse modeling. When we set up the model, the CEDS emission inventory had not projected emissions beyond the year of 2014.**

**The reviewer directly pointed at the large discrepancy between the modeled NO2 using the *a priori* and TROPOMI NO2 (Figure S11, former Figure S3) which goes to the show that the prior emissions needed a serious adjustment. The results in the manuscript are based on the a posteriori (the constrained model), not the a priori.**

**Because several points were raised by this reviewer, we broke down their arguments:**

*"the picture is less clear, with some regions (e.g. large parts of UK and Germany in March) showing large emission increases in 2020 (Figure 4). Those changes are unrealistic and are contradicted by comparisons with surface NO2 measurements (Figure 7)."*

**The model by itself is incapable of reproducing those changes in NO2 (because the same anthropogenic emissions are used for both years); we do see them because of the use of the satellite observations (mainly TROPOMI NO2 here). We need to put the motivation of this study explicitly stated in the introduction into the context: "**How representative are satellite observations at capturing surface air quality through an inversion context?" **In many places explicitly stated in the discussion, the changes suggested by the constrained model (along**

with TROPOMI NO2), such as UK and northeastern Germany in March, and eastern Europe in May, are not fully supported by in-situ measurements. It is because of this reason that we had limited the ozone analysis (the main focus of the present study) with the month of April.

There are some similarities between surface NO2 and the constrained model in March over northeastern Germany and UK. The surface measurements show little changes or higher values in 2020 compared to 2019. But these changes are obviously exaggerated by TROPOMI NO2 (and the constrained model). A screenshot of the TROPOMI NO2 anomaly map in March (bottom), the constrained model (top, right), and the surface measurements (top, left) for reminder:

[Figure]

To decipher this discrepancy we ideally would require additional data including aircraft spirals and surface spectrometers. Available data are too limited to make a sense out of the disagreement. The reviewer's concern had already been addressed in the manuscript: "Given the reasonable performance of our model at reproducing the changes observed over the surface in April, *a result of abundant samples from TROPOMI*, we only focus on this month for the subsequent analysis." **This statement is strongly in line with the concern we raised in the introduction: If the satellite observations are too uncertain (*flagged by two stringent tests, filtering out averaging kernels<0.5, explaining the partition of the information gained from the observations with respect to the prior knowledge of emissions, and comparisons to the surface observations*), we will have to drop them; so did we for the months of March and May. It is important to recognize that illustrating satellite observations can sometimes depreciate model values in some places is quite useful for the retrieval community and validation studies.**

*The authors present the disparity between regions as a consequence of the different timing of Covid-lockdown measures over different regions of Europe, but the discussion is poor and does not present concrete arguments for the inferred patterns.*

**Thanks, there are two instances in which we attribute the NOx emission differences to the different timelines of the covid-19 lockdown measures:**

1) **Moscow, Russia. We are confident in transitioning of NO2 from positive in March (2020 minus 2019) to negative in April. This is due to Russian governments' late action on imposing restrictions. The transition strongly coincided with a credible report [**https://tass.com/society/1144123**].**

2) **UK and Poland. Based on [https://en.wikipedia.org/wiki/COVID-19_pandemic_in_the_United_Kingdom#Spring_2020:_First_wave; Figure S1 in** [https://www.frontiersin.org/articles/10.3389/fpsyg.2020.579181/full](https://www.frontiersin.org/articles/10.3389/fpsyg.2020.579181/full)**, and** [https://www.bbc.com/news/uk-51981653](https://www.bbc.com/news/uk-51981653)**], the total shutdown was enforced on 18-20th March 2020; in response, we observed larger reductions in TROPOMI NO2, the top-down NOx emissions, and surface measurements in the month of April compared to March.**

**We agree that a more rigorous approach should be taken in general to fully connect the differences to the pandemic activities. Such approach should rely strictly on bottom-up information (such as synergistically combining data from cell-phone locations, traffic patterns from cameras, and mass flux measurements from industrial sectors). Satellite-based NO2 data are too crude to perform a source apportionment analysis. So we removed the discussion about the northeastern Germany.**

**To account for the reviewer's concern, we removed this part from the abstract: "." We changed this part in the abstract:** "However, NO_x emissions remain at somewhat similar values or even higher in Poland, UK, and Moscow in March 2020 compared to the baseline possibly due to the timeline of restrictions." **We removed the spatial part from the conclusion :** "Second, a large  temporal variability associated with the reduction in NO_x was evident, as each country might possibly have different timeline of restrictions."; **we smoothed this part in discussion and added more references:** "In general, the level of NO_x reduction is somewhat higher in April relative to months of March and May possibly due to spatiotemporal variabilities associated with the restrictions; for example, UK and Poland governments enforced the restrictions starting in the last week of March to the middle of April (see Figure S1 in Łukasz et al. [2020]; https://www.bbc.com/news/uk-51981653, accessed on April 2020)" **We removed this part in discussion**: ""

*"Those patterns are probably related to the inability of the model to match the observed NO2 column distribution, in particular (but not only) over N-W Europe (see Figure S3). The prior model strongly overestimates NO2 over northern Germany and strongly underestimates NO2 in southern Germany and in many other regions. It would be most enlightening to examine the top-down emission increments for each year and month (and not just the differences between the two years)."*

We have not used the a priori for any of the analysis. As we mentioned above, those discrepancies are induced by TROPOMI. The model simply is adjusting to the satellite-based observations (roughly weighted by the covariances matrix of observation ($S_o$) relative to the prior emission errors projected onto the observational space ($K^T S_e K$), please see the definition of the Kalman gain in the manuscript). *The overestimation of the emission in northeastern Germany and the underestimation in other areas are strongly constrained by TROPOMI NO2.*

To consider the reviewer's comment, 1) we included the prior vs posterior emissions of NOx in the new SI; our inverse modeling adjusted the emissions in six individual months of March, April, and May in 2019 and 2020.

[Figure]

Figure S3. The a priori and the a postteriori of the total NOx emissions for the months of March (first column), April (second column), and May (last column) in 2020.

[Figure]

Figure S4. The a priori and the a postteriori of the total NOx emissions for the months of March (first column), April (second column), and May (last column) in 2019.

**2) Surface observations (as an independent assessment) are included in the new SI, please note that the underestimation of the CEDS emissions is evident (especially in 2019), which was effectively mitigated. The northeastern Germany is an exception where TROPOMI observations exacerbated the bias. It is worth noting that remote sensing data provide limited information for optimizing emissions (e.g., Souri et al., 2016: https://www.sciencedirect.com/science/article/pii/S1352231016301315, Souri et al., 2017: https://agupubs.onlinelibrary.wiley.com/doi/full/10.1002/2016JD025663, Souri et al., 2018: https://agupubs.onlinelibrary.wiley.com/doi/full/10.1002/2017JD028009, Souri et al., 2020a,d: https://agupubs.onlinelibrary.wiley.com/doi/abs/10.1029/2019JD031941, https://acp.copernicus.org/articles/20/9837/2020/), so we should not expect the deviation between model and measurements to completely vanish. We expanded the discussion by adding:**

"We observe a large improvement (31-45%) in the bias associated with simulated surface $NO_2$ using the posterior emissions compared to the surface measurements in many places around Europe with an exception to northeastern Germany where TROPOMI $NO_2$ observations deviates the model from the measurements (Figs S7, S8, S9 and S10). The improvements in correlation are minimal indicating that the prior location of emissions are well known."

[Figure]

Figure S7. Comparison of daily-averaged surface NO2 observations (circles) against the simulated model in different regions around Europe in March-May 2019 (baseline). The first row uses the prior emissions whereas the second is based on the top-down emissions constrained by the satellite observations through an analytical non-linear inversion. All available observations are averaged within 15 km radius for each model grid.

[Figure]

Figure S8. Similar to Figure S7 but in different areas.

[Figure]

Figure S9. Comparison of daily-averaged surface NO2 observations (circles) against the simulated model in different regions around Europe in March-May 2020 (baseline). The first row uses the prior emissions whereas the second one is based on the top-down emissions constrained by the satellite observations through an analytical non-linear inversion.

[Figure]

Figure S10. Similar to Figure S9 but for different areas.

**3    )The absolute amount of CEDS is too low in many places. We should not confuse the spatial distribution with the magnitude of emissions. The reason why TROPOMI NO2 is low over northeastern Germany compared to surface measurements is unclear.**

**3)   This paper is trying to quantify the contributions of emissions changes on surface ozone based on the top-down emissions. We just need some prior values to build the a posteriori based upon. The inversion is an intermediate platform to translate the information from satellites (considering their random + systematic errors) to a model realization. We do not understand why this reviewer is concerned about the prior emissions. This paper is not about validating CEDS emission inventory. We could have used EDGAR emissions and reached to similar results (maybe with different iterations). In theory, you can even use constant emission rates throughout Europe and induce the emission changes by TROPOMI after many iterations.**

*The model compares also very poorly against TROPOMI HCHO, although this is partly due to issues with the observations. As detailed further below, the high HCHO values in April (also May) in Scandinavia and Russia might be artefacts as evidenced by comparisons of TROPOMI with FTIR data (Vigouroux et al. 2020). The authors apply a crude "bias correction" to the data (decrease by 25% the values below 2.5E15 cm-2, increase by 30% the values above 7.5E15 cm-2) but it is inappropriate as it probably increases values which are already too high (e.g. in April over parts of Russia and Scandinavia) and it leads to different corrections being applied in 2019 and 2020, thereby creating artificial patterns in the differences between the two years. Given the magnitude of the biases between the model and the satellite, the inferred top-down emission differences (Figure 5) have no credibility at all. Top-down anthropogenic VOC emissions in April appear to be much higher than in March, without justification. In the abstract and conclusions, much case is made of the small VOC emission decrease in March 2020, compared to March 2019, as if it could be related to Covid. This is highly misleading since the averaging kernel (AK, Figure 5) is lowest (close to zero) in March, i.e. the decrease is not well constrained by the data. The authors emphasize very much the importance of the NOx inversion on the results for VOCs in March, but their arguments (that VOC emissions in Spain and Italy are decreased only to compensate for the stronger VOC oxidation above cities due to NOx decreases) do not stand scrutiny. As shown by Figure S5, the inversion does increase HCHO columns over Spain and Italy in 2019, bringing the model (a little bit) closer to the data. For some unknown reason, this HCHO increase does not happen in 2020 despite similarly high TROPOMI columns. This explains the March patterns of Figure 5. There is an influence of the NOx optimization on the VOC results, but probably not the onedescribed by the paper.*

**Having an expectation for the constrained model to 100% match to the observations is incorrect because observations consist of errors. Mathematically, if AK=0, then 0 = (I-S^/S) meaning S^ = S meaning I = I-GK and because K is not zero, then Kalman gain (G) is equal to 0 meaning So >> Se, and xpost will largely be dominated by xprior. Therefore AK explains how far the emissions can be adjusted according to the observations (xpost ~ xprior + G(observation minus model), G and AK are linked together)**

**The same analogy applies to the optimal estimation of trace gases from satellite radiance. For instance, if the information of radiance on CO profile is extremely limited (AK=0), the posterior profile will be similar to the prior values.**

**We added in the methodology:**

"Not only does this method considers non-linear chemical feedback among $NO_2$-HCHO-$NO_x$-VOC by simultaneously incorporating the HCHO and $NO_2$ in the inversion framework, it also permits quantification of **A** that explicitly explains the amount of information obtained from the observation. Low **A** indicates low **G** making the a posteriori to be rather independent of the observational constraint."

**In low AK areas, we shall not expect simulations to line up with observations used for the optimization. Low AK values do not suggest that the model performance is excellent, it simply means that the information from satellites was insufficient to guide the model. So the constrained model is pretty much similar to an unconstrained model (the prior) in low AK areas. The publicly available HCHO observations are scarce in our case study. To account for this we added:** "As to VOC emissions, we observe improvements in the magnitude and spatial distribution of simulated HCHO columns after the inversion with respect to TROPOMI data over areas with reasonable amount of information (e.g., AK>0.2) (Figure S5 and S6)."

**Due to a larger number of arguments, we need to fragment this paragraph:**

*"The model compares also very poorly against TROPOMI HCHO, although this is partly due to issues with the observations. As detailed further below, the high HCHO values in April (also May) in Scandinavia and Russia might be artefacts as evidenced by comparisons of TROPOMI with FTIR data (Vigouroux et al. 2020). The authors apply a crude "bias correction" to the data (decrease by 25% the values below 2.5E15 cm-2, increase by 30% the values above 7.5E15 cm-2) but it is inappropriate as it probably increases values which are already too high (e.g. in April over parts of Russia and Scandinavia) and it leads to different corrections being applied in 2019 and 2020, thereby creating artificial patterns in the differences between the two years."*

**We will address this point in different aspects:**

1) **The crude bias correction is based on the compilation of several FTIR measurements averaged over several months (see Figure 3 in *Vigouroux et al. 2020*). The paper explicitly concluded in page 3760:** *"The negative bias over high-HCHO-level sites (biomass burning or megacities) could be due to aerosol effects."* **Negative here means TROPOMI HCHO is too low. Each individual location has a significant variability related to the bias which is not formulated in *Vigouroux et al. 2020*. This is the biggest obstacle in using the numbers reported in the validation studies. A concrete way of correcting satellite biases is to establish a relationship between the prognostics used in the retrievals and the bias. Those relationships (say a multi-linear regression involving surface albedo, cloud fraction, cloud albedo, aerosol**

**optical properties, SZA, and etc. with quantified errors) can be further used and adjusted dynamically in the inverse modeling framework, which is commonly known as "variational bias correction" in the field of meteorology. To better inform this, we included in the paper:** "Directly incorporating these numbers into an inversion model is challenging, mainly because of spatiotemporal variability in the satellite errors. Ideally, the relationship between errors and retrieval inputs (e.g., albedo, scene radiance, profiles, etc.) would be used as an additional cost function in the inversion, commonly known as variational bias correction [e.g., Auligné et al., 2007]. In the absence of such relationships, we use the biases reported in the validation studies."

2) **If one provides a certain bias over a certain point that is x% impacted by biomass burning activities, how can an inverse modeler quantitatively use that number to scale the observations? In other words, how representative a one-point comparison is at fully characterizing errors all over the scene, which may or may not be impacted by biomass burning?**

3) **Satellite errors are significantly larger in higher latitudes due to larger scattering weights (i.e., any random error will be amplified). Those errors (provided with the data) are considered in the inversion in addition to the 4% magnitude-dependent error. This is the main reason that we observed very low AK over the biomass burning activities in Russia in spite of the strong signal. This result suggests that our inversion is aware of the fact that those pixels are uncertain.**

4) **Despite some enhancements of VOC emissions over this area, the constrained model still significantly underestimated the biomass burning signal compared to Figure 5 in** *Vigouroux et al. 2020.* **This supports that the difference map is not an artifact but shows the right direction towards mitigating the large underestimation of the model; so the direction of the map is credible. VOC emissions in April 2019** *are and should be larger* **than April 2020 over those areas. The inversion did not** *overly* **correct the emissions. We added a sentence to point out this caveat:** "The inversion partly corrects for the large underrepresentation of biomass burning emissions in high latitudes occurring in April 2019 but due to large uncertainties of the retrieval over this area, averaging kernels are low. Vigouroux et al. [2020] showed FTIR HCHO columns to be around 4-6×$10^{15}$ molec/cm$^2$ in Saint Peterburgh (59.9°N), Kiruna (67.8°N), and Sodankylä (67.4°N) in April 2019. Despite some improvements over the biomass burning areas in April 2019, the model still greatly underestimates HCHO columns suggesting more well-characterized observations are needed to adjust the emissions."

**A relevant criticism to this large underestimation of model is if this would degrade the ozone analysis. April 2019 exhibited a particular synoptic condition. The predominantly high pressure system over Russia and Scandinavia effectively holds on to biomass burning contributions and prevent them from being transported to the central Europe. Moreover, the upper winds are westerly. The main findings of this study are concentrated on central Europe (box L). So we do not think those complications will impact the major conclusion of this manuscript. To account for this, we added the following figure to SI and wrote:**

"The predominately high pressure system formed over these areas (Figure S15) in April 2019 impedes the transport of the biomass burning pollution to central Europe."

[Figure]

Figure S15. The WRF-simulated mean sea level pressures in April 2020 (left) and 2019 (right).

*VOC emissions in April appear to be much higher than in March, without justification.*

**Thanks for raising this good point. We mentioned in the ozone analysis discussion that April 2020 is exceptionally warmer than the average with clear sky over central Europe. This tendency is thoroughly studied in** Ordóñez **et al. [2020], and shown in former Figure 6. As a result, HCHO levels are larger in April 2020 in central Europe compared to April 2019. From this month (i.e., April and May), biogenic emissions also kick in. TROPOMI HCHO provides good information to improve the relevant emissions, evident in AK close to 0.4. We shall not treat averaging kernels as binary values. To address this comment, we first added the averaged biogenic fraction, the a priori vs the a posteriori in SI, and added:**

"The inversion suggests larger VOC emission rates in April 2020 compared to April 2019 over central Europe. Ordóñez et al. [2020] reported ambient temperature along with solar radiation to be higher than the norm. This is primarily due to a well-developed high-pressure system over the region [Figure S] resulting in elevated HCHO columns. The top-down estimate is indicative of too low prior VOC emission rates over this area. Given the significant role of VOCs in the formation of ozone in urban settings, this correction with reasonable AK (~0.4) is crucial for precisely modeling the surface ozone anomalies (shown later)".

[Figure]

**Figure 5.** Same as Figure 4 but for the total VOC emissions. Biogenic fractions are based on the average values in both 2019 and 2020.

[Figure]

*Figure S5. The a priori and the a posteriori of the total VOC emissions for the months of March (first column), April (second column), and May (last column) in 2020.*

[Figure]

*Figure S6. The a priori and the a postteriori of the total VOC emissions for the months of March (first column), April (second column), and May (last column) in 2019.*

*In the abstract and conclusions, much case is made of the small VOC emission decrease in March 2020, compared to March 2019, as if it could be related to Covid. This is highly misleading since the averaging kernel (AK, Figure 5) is lowest (close to zero) in March, i.e. the decrease is not well constrained by the data. The authors emphasize very much the importance of the NOx inversion on the results for VOCs in March, but their arguments (that VOC emissions in Spain and Italy are decreased only to compensate for the stronger VOC oxidation above cities due to NOx decreases) do not stand scrutiny. As shown by Figure S5, the inversion does increase HCHO columns over Spain and Italy in 2019, bringing the model (a little bit) closer to the data. For some unknown reason, this HCHO increase does not happen in 2020 despite similarly high TROPOMI columns. This explains the March patterns of Figure 5. There is an influence of the NOx optimization on the VOC results, but probably not the onedescribed by the paper.*

**Inferring emissions only from satellite columns is tricky. This is especially the case for HCHO columns whose errors can be large and comparable to the absolute value of the columns in relatively colder months, and there is a substantial chemical feedback from NOx levels on HCHO formation.**

**Due to large errors of HCHO retrievals in March relative to their columns, the model is barely constrained by the HCHO columns. The large changes in VOC emissions are caused by NOx. This reviewer provided a very good point later that we can prove this feedback by running the inversion with constant HCHO levels; our inversion in this month is very close to this experiment (extremely low AKs (no Kalman gain) means no observational constraint).**

**We need to consider the errors associated with observations, the a priori, and Jacobians. We can reproduce the same amount of HCHO with varying VOCs if we change Jacobians (which are changing due to chemistry, photolysis rates, meteorology, and etc.). Likewise, the same amount of VOCs can yield different HCHO levels under different atmospheric environments. All of these complexities are considered in our inversion. The impact of NOx on the formation of HCHO is comprehensively studied in papers cited in the manuscript including Souri et al. [2020b], please take a look at Figure 10 and Figure S10 in https://www.sciencedirect.com/science/article/pii/S1352231020300820). We copy Figure S10 here because the SI is open access:**

[Figure]

*Figure S10. The sensitivity of HCHO columns to one mole/s of NOx emissions calculated by the CMAQ-DDM. The time period is May-June 2016 at 1:30 LST. The unit is molec.cm$^{-2}$*

**This figure is generated by changing one mole/s of NOx emissions uniformly over East Asia. 1 mole/s is roughly equivalent to 2.7 ton/day (assuming 90% NO and 10% NO2). The formation of HCHO is strongly dampened by increasing NO2 (due to OH suppression and formation of organic nitrates) in NOx-saturated areas. Values are around 1-3x10$^{15}$ molec.cm-2 for 2.7 ton/day. While this derivative can vary under different atmospheric conditions, we observed the same tendency over Europe. Jacobians of HCHO to VOC increase after the first iteration in March 2020. Our model is not coupled, so there is no feedback from chemistry on meteorology. The reason we do see the differences in Jacobians of HCHO with respect to VOC (occurring right after the first iteration) is solely because of the chemical feedback. HCHO observations are rather useless here (although**

they provide an upper bound for the HCHO changes, meaning HCHO columns weighted by their errors are not too different between two years, so the HCHO constraint on VOCs is minimal). Nonetheless, there is, unfortunately, a drawback in our implicit joint inversion of NOx and VOC approach, which is not including the cross relationships between NOx-HCHO and VOC-NO2, shown in color fonts (from our AGU's poster):

$$\begin{bmatrix} NO2^1 \\ NO2^2 \\ \vdots \\ HCHO^1 \\ HCHO^2 \\ \vdots \end{bmatrix} = \begin{bmatrix} \dfrac{S_{NOx}^{NO2^1}}{NOx} & \dfrac{S_{VOC}^{NO2^1}}{VOC} \\ \dfrac{S_{NOx}^{NO2^2}}{NOx} & \dfrac{S_{VOC}^{NO2^2}}{VOC} \\ \vdots & \vdots \\ \dfrac{S_{NOx}^{HCHO^1}}{NOx} & \dfrac{S_{VOC}^{HCHO^1}}{VOC} \\ \dfrac{S_{NOx}^{HCHO^2}}{NOx} & \dfrac{S_{VOC}^{HCHO^2}}{VOC} \\ \vdots & \vdots \end{bmatrix} \begin{bmatrix} NO_x \\ VOC \end{bmatrix}$$

Based on experiments we did in Souri et al. [2020a], we came into conclusion that constructing those cross-relationships in the Jacobians were not necessary. Especially if multi-sensors are used (we used both OMI/OMPS in that study); trying to co-register cross-relationships degraded the quality of the inversion if one sensor had too many gaps; moreover it is quite expensive to calculate those derivatives.

Although the non-linear feedback between NOx-VOC are implicitly considered by incrementally updating Jacobians, the lack of those cross-relationships means that we cannot explicitly quantify the amount of information gained from TROPOMI NO2 on VOC emissions (and HCHO on NOx emissions). AKs in the paper only explain the amount of information we gained from HCHO on the VOC estimate (and NO2 on NOx). Since we do not have this important piece of information right now, we decided to drop the discussion about the VOC reduction in March (abstract, conclusion, table, and discussion).

*The paper is too long and, at many instances, not clearly written. I have provided a number of suggestions for improvement, but I encourage the authors to make a general effort towards more clarity. Many sentences and entire paragraphs are given which do not add much to the discussion. For example, Figure 6 is very long to describe, but I am not sure whether it really helps to interpret the results. If it does, please make it more clear and remove unnecessary parts. On the other hand, much information about the model and methodology is incomplete or clearly wrong (e.g. the adopted errors for TROPOMI HCHO).*

Thanks for you feedback. We omit Figure 6 and the associated discussions. We also removed the discussion about the NOx impact on VOC emissions in March due to not being able to estimate cross-AK at this point. We also did our best to shorten some parts and improved the clarity.

*In conclusion, I do not recommend the paper for publication (in its present form) since its conclusions are not well supported by analysis of the data. I recommend to scale down the ambition of the paper. The HCHO (and AOD) data do not seem to help constraining the emissions. The NOx part could be interesting if presented honestly with its caveats. Sensitivity inversions would help to appreciate the uncertainties and robustness of the conclusions.*

*Minor comments*

*Abstract: very long, should be shortened*

**We removed some sentences in the abstract.**

*l 122-124 "Since vertical column densities (...° depend on assumed gas profile shape (...), we recalculate those shape factors using profiles from our (...) model": the air mass factors being a complex function of profile shapes, cloudiness, albedo, etc., more details are needed to describe how the profile shapes are taken into account.*

**AMF is defined as (taken from Souri et al., 2018):**

$$AMF = AMF_G \int_{\sigma_o}^{1} \overbrace{-\frac{1}{AMF_G} \frac{\alpha(\sigma)}{\alpha_e} \frac{\partial \ln I_B}{\partial \tau}}^{w(\sigma)} \times \overbrace{\frac{\Omega_a}{\Omega_v} \eta(\sigma)}^{S(\sigma)} d\sigma,$$

**We only re-estimated the S(sigma) based on the profiles. These profiles are iteratively updated based on the new emission estimate.**

**We added :** "Shape factors are re-estimated by calculating the ratio of the vertical column of total air to the simulated vertical column of $NO_2$ multiplied by the mixing ratios of $NO_2$ profile from the regional model [Martin et al., 2002]."

*l 156 Why the RMSE? Do you mean the assumed uncertainty on the NO2 columns from TROPOMI?*

**We improved this part for clarity:**

We assume the errors of observations originate from two main sources: i) the precision error provided with the data ($e_{precision}$) and ii) a fixed error estimated from comparisons to in-situ measurements ($e_{const}$). Mathematically, the final error is:

$$e_O^2 = e_{const}^2 + \frac{1}{n^2} \sum_{i=1}^{n} e_{precision,i}^2 \tag{1}$$

where *n* is the number of samples for a given grid and $e_{const}$ equals to $1.1\times10^{15}$ molec/cm$^2$ ($<6\times10^{15}$ molec/cm$^2$) in clean regions and $3.5\times10^{15}$ molec/cm$^2$ ($>=6\times10^{15}$ molec/cm$^2$) in moderately to highly polluted regions. These regions are defined based on the wide ranges reported in Verhoelst et al. [2021] (3-14$\times10^{15}$ molec/cm$^2$ for polluted areas).

*l 156 The values of 1.1E15 and 3.5E15 molec cm-2 seem arbitrary. Please provide better explanation of how those were derived (as they play an important role in the emission inversion)*

**The relevant paper provided these errors within a wide range (3 to 14 Pmolec cm−2 for labeling moderate-polluted areas), we found a reasonable result by choosing these numbers, please see the comparison of surface/model, and the difference maps in April against in-situ measurements. Analytical inversion is extremely time and space consuming, we cannot redo the inversion with different numbers. Again the problem with directly using the reported errors is more fundamental (mentioned earlier). The errors used in the inversion are within the reported values from the validation studies listed in Table 1.**

*l 176-178 "Vigouroux et al (...) majorly located over pristine areas and 9 MAX-DOAS stations..." : wrong. The paper concerns FTIR stations, not MAX-DOAS. Furthermore, many of the FTIR sites are in cities (e.g. Paris, Bremen, Mexico city, etc.). Please check the references you cite.*

**Thanks for your comment. We fixed the discussion:** "Vigouroux et al. [2020] expanded the validation suite by including more than 25 FTIR stations located over both pristine and polluted sites. Results from the comparison with FTIR measurements (over clean areas) also indicate a high bias, whereas those compared in polluted areas show a low bias. By compiling numbers quoted in Lambert et al. [2020] and Vigouroux et al. [2020], we correct the existing biases in TROPOMI HCHO by scaling 25% ($<2.5\times10^{15}$ molec/cm$^2$) down columns in clean areas and 30% ($>=8\times10^{15}$ molec/cm$^2$) up in polluted areas. We assume the constant term of errors ($e_{const}$) to be equal to 4% of HCHO total columns based on Vigouroux et al. [2020]."

*l 181-183 "The agreement between MAX-DOAS...": again, this paper concerns FTIR data only. Please provide the correct references to your statements.*

**Thanks, addressed above.**

*l 184-186 Please provide the precise procedure used for deriving those numbers.*

**The numbers are reported in those studies (we found a typo in the upper bound). Please see the abstract of Vigouroux et** al. *"an overestimation (+26 ± 5 %) of TROPOMI is observed for very low HCHO levels (< 2.5 × 1015 molec. cm−2 ), while an underestimation (−30.8% ± 1.4 %) is found for high HCHO levels (> 8.0 × 1015 molec. cm−2)."*

*l 186-187 A value of 4% seems extremely low and unlikely given the large biases and scatter of the FTIR-TROPOMI comparisons.*

**We added the equation regarding how we formulated the errors in the inversion (see above). 4% is indeed low, but it is based on the reported deviation of the bias in Vigouroux et. The bias varies between 1-5% on average (see the above abstract). The dominant error in TROPOMI HCHO originates from the precision errors which are provided with the data (see the new equation number 1; we also had mentioned this in the manuscript:** "The instrument covariance matrices are populated with squared-sum of the aforementioned errors based on the compilation of the validation studies and precision errors provided with the data."**). The RMSE associated with spectrum fitting of HCHO is large due to relatively week absorption of HCHO molecules in the UV-range.**

*l 188-201 The motivation for using MODIS AOD is not made clear. Clarify. If it does not bring anything, why this complication?*

**We were not satisfied with the performance of the model in terms of aerosol in the beginning, so we improved it using MODIS AOD. The photolysis rates are impacted by aerosols in CMAQ.**

*l 209-210 The assumption that the interferences are similar in 2019 and 2020 due to low photochemistry is crude. In Lamsal et al 2008, the correction factors in spring over the U.S. range typically between 0.4 and 0.7. Since CMAQ calculates the interefering species (PAN etc.), why don't you apply the correction proposed by Lamsal et al.? It is a rough correction but it would be better than no crrection at all.*

**Thanks, we used to correct this complication in our former studies (Souri et al., 2016; 2017; 2018), but after realizing how poorly models can perform in terms of NOy in colder months (see Figure 7 in Travis et al., 2020 based on the ATom campaign), we started having doubts on whether this correction can be beneficial. Ideally we would need to constrain or at least validate NOz in the model before applying the correction. NOz observations are lacking during the case study.**

**To clarify, we have added:** "Additionally, the correction needs a careful evaluation of the model with regards to the NO$_z$ family whose measurements are not available in this case study"

*l 228 Do you use gridded maps of the emission factors or PFT distributions in conjunction with the emission factors from Table 2 in Guenther et al 2012?*

**PFTs are the fraction of land/use land covers based on the Community Land Model (CLM). They also are used for scaling LAI because MEGAN uses LAIv (scaled by the fraction of vegetation). The emission factors are estimated by the new information of PFT and the PFT-specific values in. Guenther et al., 2012.**

To clarify, we added: "The biogenic emission factors are estimated based on the PFT-specific information provided in Guenther et al. [2012]."

*l 230 Does the model include soil and lightning NO emissions? The use of fertilizers could be a significant source in spring.*

**MEGAN v2.1 also estimates the soil NOx emissions based on J.J. Yienger and H. Levy II, Journal of Geophysical Research, vol 100,11447-11464,1995 (see soilnox.F subroutine in the model). We added: "**Soil NO$_x$ emissions are estimated by Yienger and Levy, [1999].**".**

**In terms of lightning NOx, we turned on the inline calculations which uses the convective precipitation rates and cloud bottom/top layers. Flash counts were not available during the project. We added**. "Lightning NOx emissions are based on in-line calculations involving convective precipitation rates and cloud vertical distributions. Lightning NOx emissions are not constrained in the model."

**For constraining the lightning NOx emissions, we will definitely need the profile information.**

*l 230 Is the diurnal cycle of anthropogenic emissions taken into account? Regarding the biogenic emissions from MEGAN, are diurnal and day-to-day variations included? What VOC species are emitted (besides isoprene)?*

**Unfortunately, CEDS does not provide the diurnal scales. While we adopted some US EPA-based diurnal correction for mobile sources in Souri et al., 2020a over East Asia, we realized that those crude approximations might not always be true. Furthermore, TROPOMI cannot constrain diurnal information. The NO$_2$ comparisons and analysis are all daily-averaged.**

**On the other hand, we do consider hourly-basis biogenic VOC and biogenic NOx emissions in the model based on MEGAN.**

**We added:"** Anthropogenic emissions are based on the Community Emissions Data System (CEDS) inventory in 2014 [Hoesly et al., 2018]. Diurnal scales are not considered."

"Hourly-basis biogenic emissions are processed by the offline standalone"

**MEGAN produces many different VOCs including isoprene, monoterpenes, xylene, ethanol, methanol, CH4, acetaldehyde, ethene, ethane, toluene, paraffin, and direct formaldehyde. MEGAN has a processor to convert those emission reported in Guenther 2012 paper to CB05 mechanism. We added : "**The biogenic VOCs include a wide range of compounds including isoprene, monoterpenes, aromatic VOCs, and methanol."

*l 247 Is **y** a collection of monthly NO2 columns or daily NO2 columns? Is the model sampled at the satellite overpass time? Specify the temporal tolerance window.*

**Monthly-basis. Not only the model is sampled at the same but also the Jacobians and emissions (state vectors) are sampled at the same time and over "qualified pixels". TROPOMI provides a time variable for each row used for co-registration. Everything is**

synched. **We do not need a temporal tolerance; each pixel has a definitive time. We added:** "The model outputs along with Jacobians and emissions are spatiotemporally co-registered with the observations." **and** "where *y* is bias-corrected monthly-averaged TROPOMI NO₂ and HCHO observations,"

*l 257 Why three times? How do know whether this is sufficient?*

**This number is based on two factors: first, we used the same number in our previous study [Souri et al., 2020] over East Asia for which we found satisfactory results against satellite/observations indicating a reasonable convergence. Non-linear chemistry was a problem in that region due to large oxidation capacity and extremely large emission rates. Second, we defined certain time and computational resources for this project. The analytical inversion is extremely time (and space) consuming at a continental regional-scale.**

*l 263-267 The rationale for this assimilation of MODIS AOD is not clear.*

**We addressed this before.**

*l 275 The faster vertical mixing should generally lead to higher NO2 columns due to the higher sensitivity of TROPOMI to NO2 at higher altitudes. Stronger advection does not change much when averaged over a sufficiently large area. Clearly, the increased photochemical activity is by far the main reason for lower NO2 columns in later months.*

**First, If the receptor is at the location *A* and winds are stronger vertically, the receptor at *A* will experience lower concentrations of NO2 columns. We cannot change the location of the receptor at the same time. To be more specific: "faster vertical mixing due to larger sensible fluxes (more diluted columns for a given receptor due having a greater chance of experiencing stronger winds in higher altitudes),"**

**Second, the larger sensitivity of TROPOMI NO2 to upper levels is only relevant for SCD. The primarily reason of using AMF is to scale the column based on the sensitivity. VCD is defined as SCD/AMF. Scattering weights increase by altitude, so even a small amount of NO2 in the free troposphere largely increases AMF scaling down total columns. In this part of paper we are pointing at fundamentals, not what can go wrong with the profile.**

*l 282 "we see negligible reductions..." : there is no reduction at all. There is a significant increase in these regions (boxes B, C and D). Rephrase.*

**Thanks, we changed it to increase.**

*l 289 "northern Germany is associated with less populated areas": quite an extraordinary statement. Please look at population density maps. Please focus on relevant information, e.g. the timing of the lockdown, besides meteorological variability. When did lockdown measures take effect in Germany, France, Italy, etc.?*

**We removed the discussion about the northeastern Germany.**

*l 308 A detection limit of 7E15 cm-2 is not "very low", since it is higher than the TROPOMI columns at most locations in March-May (Figure S5)*

**Detection limit can be significantly lowered down by coadding pixels over time/space. So we should not directly compare this value to the monthly-averaged data. This detection limit is magnificent compared to former sensors for such a single small pixel.**

*l 315 The reference Karlsson et al. 2013 does not inform on the occurrence of biomass burning in 2019*

**We also added a NASA link ([https://earthobservatory.nasa.gov/global-maps/MOD14A1_M_FIRE](https://earthobservatory.nasa.gov/global-maps/MOD14A1_M_FIRE)).**

*l 315 Over St Petersburg, the FTIR HCHO column in April 2019 is about 4.2E15 cm-2 (Vigouroux et al. 2020), a factor of 1.6 below the TROPOMI column. A similar overestimation is found at Kiruna. In May, the discrepancy is even higher at Scandinavian sites. Clearly, TROPOMI data over Northern Europe and Russia in spring need to be considered with extreme caution. The "dipole anomaly" (line 319) might very well be an artefact (at least quantitatively)*

**We discussed this before. Our model is largely underestimated there and TROPOMI HCHO provided little information due to large precision errors associated with high latitudes. We did not overcorrect the emissions.**

**As for the month of May, biomass burning contributions subsided (please see https://earthobservatory.nasa.gov/global-maps/MOD14A1_M_FIRE). The anomaly is strongly related to surface temperature (biogenic). We added:** "We revisit the pronounced dipole anomaly of dominantly biogenic VOC emissions in May. In this month, the biogenic VOCs dominate. Our model suggests that ambient surface temperature differences between Russian and central Europe are more than 7°C, possibly inducing a strong dipole anomaly in biogenic emissions."

*l 317 "the fact that isoprene reactivity significantly increases by rising temperature": the OH-rate constant actually decreases at higher temperatures. The chemical lifetime of isoprene is always short enough that it is oxidized close to the emission area. Nevertheless, there is a longer delay in winter/spring before oxidation products like MACR and MVK get oxidized and form HCHO. But this should not play a significant role compared with the temperature-dependence of biogenic emissions. Note furthermore that over Russia and Scandinavia, where coniferous trees are dominant, monoterpenes (not isoprene) might be the main biogenic precursors of HCHO. Are those emissions (and their subsequent chemical oxidation) considered in the model? If not, what could be the consequence of their omission?*

**We changed it to** "but also stems from faster isoprene oxidation through higher levels of OH [Pusede et al. 2015]."

**Monoterpene is generated by MEGAN (added in the model description) and is included in Cb05:**

| | | 0.967*C2O3 | | |
|---|---|---|---|---|
| R149 | TERP+O | 0.150*ALDX + 5.12*PAR | 3.60E–11 | 20 |
| R150 | TERP+OH | 0.750*HO2 + 1.250*XO2 + 0.250*XO2N + 0.280*FORM + 1.66*PAR + 0.470*ALDX | 1.5E–11 @ –449 | 20 |
| R151 | TERP+O3 | 0.570*OH + 0.070*HO2 + 0.760*XO2 + 0.180*XO2N + 0.240*FORM + 0.001*CO + 7.000*PAR + 0.210*ALDX + 0.390*CXO3 | 1.2E–15 @ 821 | 20 |
| R152 | TERP+NO3 | 0.470*NO2 + 0.280*HO2 + 1.030*XO2 + 0.250*XO2N + 0.470*ALDX + 0.530*NTR | 3.7E–12 @ –175 | 20 |

*Section 3.2 Before discussing the top-down emissions, the paper should discuss the performance of the a priori model against satellite data. I find striking that the model fails at reproducing many prominent features of NO2 column distributions. Why is CMAQ NO2 so high along the coasts of Germany and Holland whereas it is notably too low e.g. over southern Germany? Over Ukraine and other regions, the model is too low by a very large factor (>4). The paper should show not just the top-down emissions but also the emission increments and discuss whether those increments have any plausibility. I have serious doubts on that matter. The a priori emission distribution (from CEDS) might have some uncertainty but cannot be completely wrong.*

**We have comprehensively addressed this in the major comment.**

*l 328 "elsewhere": elsewhere in the paper or in a further study? I would guess that these aerosol changes have only limited impacts on NO2 and HCHO. If so, the impact of AOD assimilation should be either briefly mentioned or dropped entirely from the paper. If not, it would be interesting to discuss more in detail.*

**We needed to improve the quality of aerosol mass in our model. The photolysis rates are impacted by aerosol optical thickness which directly/indirectly impacts NO2, HCHO, and ozone. Unfortunately, we had not saved photolysis rates with/without AOD adjustment analysis for the whole time period (with the inversion) to show the differences of the impacts. As a modeler, we have to provide all necessary adjustments made to the model.**

*l 335-336 "large reduction (...) in the bias associated with simulated surface NO2": why not show this, e.g. in the Supplement?*

**Thanks, we added in the major comment.**

*l 338-339 "the discrepancies between the simulated tropospheric NO2 columns versus TROPOMI are largely mitigated by the inversion": only in region with highest emissions, not at all elsewhere.*

**Yes, this exactly points out the definition of AKs. Low AKs means xposteriori is roughly similar to xpriori.**

*l 342 "because of the consideration of observational errors": but the choice of NO2 column errors was pretty arbitrary (as far as I understand). It could be useful to show inversion results adopting alternative choices of those errors and other setup parameters.*

**We addressed this earlier. Those numbers are within the reported values.**

*l 343 During summer and even in spring (at least in southern Europe), the feedback would be: if NOx increase, then OH increase, then the NOx lifetime decreases, implying a larger NOx emission increase is needed to match the NO2 enhancement. Therefore, it does not seem obvious that chemical feedbacks would decrease the magnitude of the anomalies. Please clarify, or drop the mention of chemical feedback.*

**That's a very good point, We did some experiments in (https://www.sciencedirect.com/science/article/pii/S1352231020300820), Figure 10. We changed NOx and VOC in a very actively photochemical area in summer over Seoul where abundant VOC emissions were present (very similar to the southern Europe), OH as you mentioned increased by NOx in VOC-rich environment. But OH leveled off after certain points. So this chemical feedback is very subject to initial values of NOx and VOC (to be more specific reactivity of VOCs). Because we have discussed the feedback in more detail in our previous papers. We decided to remove this sentence.**

*l 344 "some doubt the practicality of direct mass balance methods": at least, such methods provide a direct answer independent on assumptions regarding uncertainties.*

**We do not wish to treat the satellite observations error free and totally ignore the chemical feedback. The mass balance methods are more appropriate for precise in-situ measurements relative to emission errors, and relatively inert species like CO2 or CH4.**

*Table 2: The absolute differences of top-down NOx emissions are not really useful and could be dropped.*

**Regulatory agencies are often more interested in the absolute changes compared to the percentage, so we wish to provide this information.**

*l 358 As for NOx, a discussion of the model performance is needed for HCHO, before discussing the inversion. In addition, the a priori VOC emissions should be shown for the 3 months. Generally speaking, there is a huge underestimation of the model against TROPOMI HCHO (Fig S5). That might be partly due to biases in the data (see above regarding FTIR vs TROPOMI comparisons) but should clearly be mentioned. My guess is that the model would underestimate the FTIR HCHO columns at sites like Paris and Bremen. In any case, the large uncertainties in TROPOMI HCHO make the inversion results unreliable (except maybe at low latitudes in May). The differences "Lockdown-Baseline" (Figure 5) are even more uncertain. I think they should not be shown at all as they might mislead the reader.*

**We need to account AKs when we compare the data to the constrained model. We addressed this earlier. We added the prior VOC emissions, the biogenic fraction and more**

discussion about the potential errors associated with model. Again, we should not treat AKs as binary, we do see decent information in April in central Europe. Readers should consider AKs. That is the main advantage of an analytical inversion over a numerical way.

*l 366-367 "This tendency, which is striking, mainly stems from the indirect impacts of the reduced NOx emissions on HCHO": the reasoning is overly simplistic. E.g. over Spain, the largest change is not seen over Madrid but in an area to the west of the city. Over Italy as well, the changes are spread over wide areas. Furthermore, in April the VOC emissions are found to increase quite a lot over cities like Paris, Rome, Milano, etc. Obviously the patterns are primarily dictated by the large differences between TROPOMI HCHO and the model, despite the large HCHO uncertainty adopted in the inversion.*

**We addressed this in the major comment.**

*An additional worry concerns the seasonality of top-down emissions in 2019. According to Fig 5, VOC emissions in April are considerably higher than in March over most countries. The retrieved emission patterns indicate primarily anthropogenic emissions. How can this be justified?*

**We addressed it in the major comment. Biogenic emissions have an hourly temporal scale. CEDS has a monthly scale. We believe most of this signal is biogenic due to very favorable atmospheric conditions in April 2020. For the same reason, we observed 48% of MDA8 ozone enhancements were due to biogenic+meterology (natural variability, see the middle panel in Figure11). Additionally, the reactions of RO2+NO linked to VOCs are large (Figure 13). The relatively large AK (~0.4) suggests there are fruitful information in this area from TROPOMI HCHO.**

*Section 3.3.1 could be shortened. Get to the point!*

**We removed the discussion about the former Figure 6.**

*l 440 In March, surface NO2 is higher in 2020 than in 2019 according to the model (not the data), consistent with the column changes (Fig 4). This confirms the suspicion that the NOx emission changes in these regions are unreliable due to large model errors.*

**TROPOMI NO2 suggested those tendencies. The model by itself does not have the knowledge to make those changes. This can be an observation problem. We addressed this in the major comment.**

*Figure 8: is this given for 2019 or 2020? I suggest providing both years, but in the Supplement instead of the main article*

**It's the averaged. We added : "**Figure 8 depicts the average number of days that TROPOMI was able to sample on in both years.**".**

**We added both years in the SI. Please note how clear the sky is in April 2020 (the reason of elevated HCHO and 48% of ozone based on our model).**

[Figure]

*Figure12. The number of good quality (qa_flag>0.75) TROPOMI tropospheric NO$_2$ days observed at 15×15 km$^2$ in 2019. These numbers are heavily affected by cloudiness.*

[Figure]

*Figure13. The number of good quality (qa_flag>0.75) TROPOMI tropospheric NO$_2$ days observed at 15×15 km$^2$ in 2020. These numbers are heavily affected by cloudiness.*

*l 566-568 "large spatial and temporal variability associated with the reduction in NOx was evident as each country might have different level and timeline of restrictions": however, the discussion of this aspect is poor in the paper. "emissions decreased in April rather than March in some portions of UK, northern Germany...": do you really mean that the UK and Germany both showed significant regional differences in the lockdown measures? I doubt very much that it was the case, but if it is, it should be discussed and better justified.*

**We addressed this earlier.**

*l 569-571 "we showed that anthropogenic VOC emissions over Paris (...) decreased in March (...) achievable through jointly using NO2 and HCHO observations" as noted above, this is very doubtful. You have not made your case that the VOC emission changes are due to NOx emission changes. For that, you should realize a separate inversion using only TROPOMI HCHO and compare with the standard inversion.*

**We addressed this earlier.**

*Technical/language comments*

*l 32 "estimate of the NO2 reduction is underestimated": rephrase*

**Changed to** *"*Comparisons against surface monitoring stations indicate that the constrained model underrepresents the reduction in surface NO$_2$. This underrepresentation correlates with the TROPOMI frequency impacted by cloudiness."

*l 32 "a picture that correlates with the TROPOMI etc.": unclear*

"This underrepresentation correlates with the TROPOMI frequency impacted by cloudiness."

*l 37 "TROPOMI HCHO sets an upper limit for HCHO changes such that the chemical feedback (...) reveals a non-negligible decline..." : unclear. That a feedback reveals a decline doesn't make sense.*

**We removed this part.**

*l 44 "Results of integrated process rates of MDA8 surface ozone": unclear*

**We changed it** *"*The model suggests that physical processes (dry deposition, advection and diffusion) decrease MDA8 surface  ozone in the same month on average by -4.83 ppbv, while ozone production rates dampened by largely negative $J_{NO2}[NO_2]-k_{NO+O3}[NO][O_3]$ become less negative, leading ozone to increase by +5.89 ppbv."

*l 51 "capture the essential character changes..." essential in what sense? Unclear.*

**We removed it.**

*l 57 "has exponentially become more polluted during previous decades": wrong over Europe*

**We changed this:** *"*Earth's atmosphere has substantially become more polluted since the industrial era in comparison to its original environmental condition [Li and Lin, 2015], thus any abrupt hiatus in anthropogenic (man-made) emissions"

**We simply want to say the anthropogenic-emissions are large.**

*l 59 "impulsive and sweeping" : not clear what is meant*

**Changed to** *"immediate* impact"

*l 78 "particulate matter" (drop the s)*

**Removed.**

*l 100 Why the indentation?*

**Unclear.**

*l 143 low spatial resolution (remove hyphen)*

**Removed.**

*l. 154 "while considering" unclear*

**This part has been rewritten:** *"In the case of NO$_2$, we uniformly scale up the satellite tropospheric columns by 25%. This bias estimate is derived by first assuming a 37% low bias in the columns over polluted regions as reported by Verhoelst et al. [2021]. In turn, this low bias can be mitigated somewhat by the application of high spatial resolution profiles in the air mass factor calculation, such as the ones used in this study. Table 1 summarizes the results from several TROPOMI validation studies at specific locations that calculated NO$_2$ using model profiles with higher spatial resolution than the operational TROPOMI (1º×1º) profiles (see Table 1 columns "Modification" and "Modified Bias"). In these studies, modified columns show increases ranging from 0 - 25%. Based on these results, we assume a low bias of 37% can be mitigated by ~12% through the use of high spatial resolution profiles, for a resulting total low bias of 25%. This bias is likely not valid over pristine areas, where validation studies show lower biases in TROPOMI NO$_2$ [Verhoelst et al., 2021, Wang et al., 2020, Zhao et al., 2020]; nonetheless, we previously observed in Souri et al. [2020a] that the low signal-to-noise ratios of those column amounts resulted in small changes in the top-down emissions."*

*l 156 The sentence should make more clear that "clear" is <6x1015 molec/cm2 and polluted is above that level. Use the proper symbol for >=*

**We changed it "clean".**

*l 176 "Those biases oscillates around 8x1015 molec/cm2": completely unclear.*

**We removed it.**

*l 177 "majorly" -> mainly*

**Changed.**

*l 236-237 The sentence "We nudge moisture (...) data used only outside of the PBL layer" is a bit ambiguous,*

**We clarified it:** *"To minimize the deviation of the model from the reanalysis data, we turn on the grid nudging option with respect to wind, moisture, and temperature only outside of the PBL region. The inclusion of this option outside of the PBL is because we do not want the coarse reanalysis data washes out the relatively high-resolution dynamics."*

*please rephrase*

*l 237 "PBL layer" is redundant*

**Changed.**

*l 240 the correspondence is good, not striking.*

**We disagree. The performance of the model, especially in terms of U and V are striking in the modeling world.**

*l 255 are assumed diagonal*

**Corrected.**

*l 277 "unintended" is weird. NO2 columns have no intention. Rephrase.*

**Changed to** *"This sequential decline of $NO_2$ obscures the quantitative interpretation of the satellite observations in two ways: first, as noted by Silvern et al. [2019], the free tropospheric background $NO_2$, which are highly uncertain, becomes comparable to those located near-surface, and second, the relatively lower signal-to-noise ratios reduce the amount of information obtained on $NO_x$ estimates (discussed later)."*

*l 278 "are first the free-tropospheric region complication": what does this mean? Not clear at all.*

**See above.**

*l 279 "a barrier to obtaining high amount of information from the sensor..." unclear, rephrase.*

**See above.**

*l 296 "suggests an abrupt hiatus in the ongoing reduced NOx emissions": unclear*

**Changed to** *"the anomaly of the tropospheric $NO_2$ suggests that the reduction in $NO_x$ emissions abruptly experiences a hiatus…"*

*l 300 Why "potential"?*

**Thanks, removed.**

*l 302 "leading to striking HCHO column patterns with large variations" does not tell anything, please remove.*

**Unclear (?), but removed.**

*l 303 "higher chance": is it really a matter of chance? Rephrase.*

**Changed to** *"it is easier to single out"*

*l 304 "looking at": rephrase*

**Changed to** *"it is easier to single out anthropogenic-derived HCHO concentration by HCHO measurements made in wintertime"*

*l 312-313 "are below the detection limit (...) to relate them to the lockdown...": lousy wording, please rephrase*

*(e.g. remove the last part of sentence"*

**We removed it.**

*l 313 "nonetheless TROPOMI sets an upper limit of these changes": not useful*

**We removed it.**

*l 363 "we surprisingly observe": weird wording*

**We removed it.**

*l. 426-427 "ignoring spatial representivity function to directly compare point measurements...": unclear, rephrase*

**Changed to** *"not accounting for spatial representivity function when it comes to directly comparing two datasets at different scales (i.e., point measurements vs the model grids);"*

*l 430 "then are then"*

**Unclear**

*l 433 "heterogenicity" --> "heterogeneity"*

**Changed.**

*l 440 "The surface measurements reinforce the less pronounced reduction in NO2 in northern Germany and UK": unclear.*

"The surface measurements in March reinforce increases (or negligible changes) in NO₂ in northeastern Germany and UK, although the magnitudes are not as large as those suggested by the model."

*L 458-460 The sentence "This tendency potentially is driven (...) has drawn much attention" is grammatically incorrect.*

**Thanks, changed to** "This tendency potentially driven by ozone chemistry"

*l 461 "The challenge is to simulate a model": unclear*

**Changed to** "The challenge is to set up a model"

*l 463 "essential character": unclear, rephrase*

**Removed.**

*l 475 "namely as": delete "as"*

**Removed.**

*l 523 In Equation (6), the rate constant should be k(O3+NO), not k(OH+NO2+M)*

**Thanks, changed.**

*l 560 Remove comma at end of sentence*

**Changed.**

*l 1009 "explain" --> "describe"*

**Changed.**

*Table 1 hyphen in MAX-DOAS*

**Changed.**

*Table 2: too many significant digits are given.*

**We removed one digit.**

---

## Author Response (AR2)

The authors did address many of my comments, and I thank them for that. The paper now provides a much more complete description of what the model and the inversions do, and this facilitates the interpretation of the results. The paper also new comparisons with surface NO2 data (Fig. S7-S10).

Still, regarding many concerns, I remain unconvinced, as explained below. The authors keep insisting in their response that "the results in the manuscript are based on the a posteriori, not the a priori" as if the large (sometimes huge) model biases were simply irrelevant. It is obvious from Fig. S11-S14 that both the prior and the posterior model fail to match the observations at most locations except the most polluted. Over many regions (e.g. Ukraine, etc.), the model-data difference for NO2 is systematic and exceeds the TROPOMI error (~1.1E15, Verhoelst et al.). This implies serious issues in the model and/or in the data (e.g. in the bias correction). Things are even much worse regarding HCHO. If we are clueless as to the causes for such discrepancies over moderately polluted regions, why should one trust the results in very polluted areas? True, the AKs provide an indication regarding where and when the inversion results are most reliable (if we accept the hypothesis that model and data are not too biased). However following that guideline, one would have to accept as very credible the NOx results for Germany in March (Figure 2 and Figure 4) indicating a strong emission increase in Northern Germany (in 2020 with respect to 2019) and an emission decrease in Southern Germany. This discrepancy between regions in the same country is an obvious artefact, as the authors implicitly admitted by removing the discussion on that region. This is a "COVID-19 paper" and the reader should be given some clues regarding patterns of emission changes which are obviously wrong. No need for cell-phone, traffic or industrial data for that. The paper does not provide any clue, probably because of the too many issues with the data (largely due to cloudiness, Fig. S16-17) and especially with the model. For example, the very wrong distribution of NO2 columns in the model over Germany (Fig S11) should have prompted the authors to try to explore its possible causes instead of relying exclusively on the power of inverse modelling. Maybe the inversion is correct, but how does it help anyone if we don't understand why?

**Thanks for your comment.**

**To make sure that we are on the same page, we need to explain the Bayesian inversion here. The goal of the inversion is NOT to *exclusively* match the model to the observations (TROPOMI). The cost function follows a quadratic shape consisting of two terms:**

$$J(\mathbf{x}) = \frac{1}{2}(\mathbf{y} - F(\mathbf{x}))^T \mathbf{S}_o^{-1}(\mathbf{y} - F(\mathbf{x})) + \frac{1}{2}(\mathbf{x} - \mathbf{x}_a)^T \mathbf{S}_e^{-1}(\mathbf{x} - \mathbf{x}_a)$$

**The left-hand side tries to minimize the differences between the observation and the model (F(x)), and the right side uses the prior knowledge as a pseudo-observation (or more precisely, an expectation, because the Bayesian inversion tries to find the maximum likelihood of P(x|y); the mathematical part is explained in https://www.sciencedirect.com/science/article/pii/S1352231016301315 and Rodgers 2000).**

**There is a competition between these two terms. If the error of prior knowledge (projected to the observational space) is very small compared to that of observations ($K_i \mathbf{S}_e K_i^T$ << So), the estimation will be rather independent of the first term. This is clearly described in the**

Kalman gain shown in our manuscript (So>>Se will make G almost zero). For example, if we are given super accurate values of NOx emissions observed by flux measurements (kg/s) over a chimney and someone asks us to verify it using model+satellite data, it will be very unlikely for the top-down emission values to be different than the prior estimate no matter how large the differences between satellite and model are. This tendency manifests in low AKs. One may say: why doesn't your constrained model match with satellites observations in this scenario (or within a sigma value of the satellite observations)? The first question we should ask is: why should it match if the prior knowledge (flux in-situ data) is more certain? Why should we degrade our estimation with garbage? The fundamental reason behind using the Bayesian inversion in all type of data assimilation/inversion frameworks is to prevent the *"garbage in-garbage out"* problem.

Uncertain observations have a less of chance to impact the emissions because the prior knowledge is competing with them. It is intuitively clear that any instrument should have a larger uncertainty (relative to their absolute values) in low/background conditions. For instance, fitting the cross-sections to the satellite radiance will be much harder (and prone to larger errors), if the molecular absorption of our targeted gas is weaker (the reason why HCHO observations are noisier than $NO_2$, and why TROPOMI $NO_2$ columns in rural areas are less credible than over polluted ones). It is because of this reason that we see a larger discrepancy between the constrained model at rural areas with respect to NO2 columns compared to polluted areas. Is this a drawback from Bayes' theorem perspective? Absolutely not, this is in fact its power. As a matter of fact, the inversion is a solid way to extract good information (variance) from the data considering the probability distribution of x (P(x)) and P(y|x), all together.

We could have narcissistically inflated our result by increasing the emissions errors or scaling down the observational errors to artificially line up our model with the observations over rural areas. Is this ever logical in the sense that our prior knowledge is completely wrong and/or satellites are so precise? The same problem applies to the biomass burning area in April 2019. The reason why the model seemingly failed to reproduce those values was because it understood that TROPOMI columns were too uncertain over that region and especially in high latitudes (due to considering TROPOMI's uncertainty column variable that were considered as e_precision (in eq1) in addition to 4%). There are "*horror stories*" expressing the consequence of not fully accounting for biomass-induced aerosol optical properties in AMF calculations (which can result in either too low or too high AMFs depending on single scattering albedo). There might be other problems with TROPOMI HCHO. We also do not want to rule out the problems with the model that may provide biased Jacobians over that area (discussed later). Overall, we think having a model not being strongly constrained by controversial observations over that area is better (safer). Kalman gain is so low over that area such that scaling up and down the columns are not going to make a noticeable difference in the a posteriori (assume post = prior + G(model-OBS); G (Kalman gain)=*epsilon*, now scale OBS by 1.2 or 0.8, the post will be resilient to the change).

Now back to reviewer's comment on the discrepancy between the constrained model and both TROPOMI and surface measurements in non-polluted areas. For example April 2019:

[Figure]

First, the discrepancy between Post and bias-corrected TROPOMI (what is shown above), has nothing to do with the bias correction factors. The bias-corrected TROPOMI was used for the inversion, so if *y* was scaled by something different (or nothing at all), the a posteriori would follow that direction ((y-F(x) + bias)*G).

Second, the inversion is not a least-squares estimation meaning that we have a competition between prior knowledge and the observational constraint. Even the outdated mass balance method partly accounts for this competition. Please take a look over Martin et al., 2002; before they suggested that new_NOx = old_NOx * (satellite column/model column), they mentioned how to derive the final answer thorough performing a weighted average (combining the estimates derived from the mass balance method and the prior knowledge):

$$\ln E = \frac{(\ln E_f)(\ln \varepsilon_a)^2 + (\ln E_a)(\ln \varepsilon_f)^2}{(\ln \varepsilon_a)^2 + (\ln \varepsilon_f)^2} \qquad (1)$$

But the fact this equation had buried in the beginning of their paper has made so many scientists unintentionally forget to make some assumptions about the errors.

Third, if one wants to directly compare X (TROPOMI) and Y (the constrained model), they are required to consider the variance of X in the comparison through a Monte Carlo method, or simply a weighted chi-sq minimization. Pixels with larger uncertainty should have smaller weights in the comparison. So aiming for 100% match between the model and the TROPOMI is flawed. This is one of the primary reasons why we are using an inversion here. Granted, any inversion system will naturally get contaminated with the model parameter errors (discussed later) meaning the estimates are simply the added information on top of an ignorant model.

Here, the reason why we still see some differences in Post and TROPOMI over rural/suburban areas is because the prior emission has played an important role. We could have loosed the prior constraint (which is set to 200% for biogenic emissions, say we could have set it to 400%) to see a bigger change (and closer values) compared to TROPOMI. But that is just one hypothesis, one realization. We shall not forget: "All models are wrong but some are useful" said George Box. We just want to learn some tendencies from the model. Who knows how uncertain MEGAN soil parametrization is? We had been transparent about all numbers going into the model/inversion since the beginning of the process. Based on these numbers, the presented results are the optimal estimates of

emissions given prior/observation errors, which ultimately helped us to get a decent anomaly maps in April over the central Europe due to less cloudiness.

**So many arguments are provided in one paragraph so again we need to fragment them:**

If we are clueless as to the causes for such discrepancies over moderately polluted regions, why should one trust the results in very polluted areas?

**The results over polluted areas are relatively more credible because the TROPOMI has a larger weight to constrain the model compared to the prior estimation (discussed above). The generalization of the fact that we do not see a good agreement (which needs to take into account the variance of TROPOMI and the prior) over less certain columns so we cannot trust the stronger (more certain) signals disregards the concept of Bayesian inversion. We had provided the independent measurements in our original manuscript for a reason. In retrospective, we wish we could have FTIR/MAX-DOAS HCHO observations (if available) to do the same thing for HCHO changes.**

However following that guideline, one would have to accept as very credible the NOx results for Germany in March (Figure 2 and Figure 4) indicating a strong emission increase in Northern Germany (in 2020 with respect to 2019) and an emission decrease in Southern Germany.

**As we mentioned in the review process, we had verified them with scrutiny: AKs and surface measurements.**

**First, without using any model or the inversion, TROPOMI (Figure 2) is suggesting those semi-artifacts (although we should recognize that there are some similarities between surface anomalies and TROPOMI over northeastern Germany; and we still believe the smaller reduction in NO2 levels over northeastern Germany suggested by surface measurements is because it's less urban thus less impacted by the reduced mobile emissions). We also do see the semi-artifact patterns without the bias-correction (shown later). Second, it is true that AKs suggest that TROPOMI was able to provide reasonable information on the emissions relative to the prior knowledge (which is subject to errors; the true AK requires the exact TROPOMI and model parameter errors), but there are four important complications that we must consider:**

**i) we cannot expect that having very few number of observations (sometimes down to 4 days out of ~30 days in a month) from TROPOMI will help us to capture the full picture of emissions over the surface (surface measurements are based on all days); so we have a temporal representativity issue and ii) we do not know the exact statistics with respect to TROPOMI errors over the area for two different years. There could be an issue with the prior profile, or any sort of uncertainty resulting in relatively too high TROPOMI NO2 in 2020 over some areas iii) could this be due to column/surface decoupling issue that are usually not resolved in CTMs; the PBL is parametrized in this model (because the spatial resolution of the model is much coarser than 100-300 $m^2$ where you can leverage LES). iv) could it be due to not fully capturing lightning in the CMAQ model (and TM5) resulting in high values of TROPOMI NO2 in 2020, since too low AMFs can cause too high VCDs? All**

**of these can happen but none of these can be easily proven without real data. To our best knowledge, publicly data are not available. This study is trying to portray what we can do given our current knowledge on satellite validation, surface networks, and model parametrization. It is not about enhancing our knowledge on atmospheric chemistry from the limited data during the pandemic. The paper concludes with: "**Unless a comprehensive air quality campaign targeting COVID-19 related lockdown is available, we recommend that the impact of lockdown on air pollution should be examined through the lens of well-established models constrained by publicly available data, especially those from space in less cloudy environments."

**At this point, without having real data, all we can do is to articulate past issues related to models/observations:**

**In Section 3.1 we had pointed that :"** However it is crucial to note that these maps are based upon sporadic clear-sky pixels that might obscure the full portrayal of emissions changes happening throughout the period (discussed later)."

**we added some limitations:**

The constrained model correlates reasonably well with the changes observed by the surface measurements in April, but it fails to reflect those in March and May.

The surface measurements in March reinforce increases (or negligible changes) in $NO_2$ in northeastern Germany and UK, although the magnitudes are not as large as those suggested by the model (and TROPOMI $NO_2$ columns). A number of factors can contribute to these large discrepancies: i) the surface measurements were present throughout the month of March, whereas TROPOMI data were frequently absent due to cloudiness resulting in some degree of temporal representativity issues; ii) the statistics used for the TROPOMI bias-correction may not always hold true, since each individual pixel can deviate from the norm of the reported biases; iii) the shape of $NO_2$ profiles simulated by the WRF-CMAQ can sometimes be uncertain due to errors in the PBL parameterization or the difficulties with resolving the non-hydrostatic components (where vertical motions are comparable to horizontal ones) [e.g., Pouyaei et al., 2021]; this complication can result in unrealistic changes in the columns.

**In conclusion:**

"Third, the changes in $NO_x$ emissions suggested by TROPOMI $NO_2$ and the constrained model over northeastern Germany in March and Eastern Europe in May were unrealistic, possibly due to observations and/or the model issues."

For example, the very wrong distribution of NO2 columns in the model over Germany (Fig S11) should have prompted the authors to try to explore its possible causes instead of relying exclusively on the power of inverse modelling.

**We would need aircraft spirals and surface spectrometers to investigate that. Again, we see the wrong distribution of NO2 columns from TROPOMI, which means that there can be an issue with the observations.**

That being said, the authors have accounted for many of my concerns and updated the text accordingly. The authors realized that the NO2 results are less reliable in March and May, which is why the ozone analysis is restricted to April, as it should. They state to their defense that "remote sensing data provide limited information for optimizing emissions [many references]" which I find contradictory since emission optimization is precisely the methodology adopted in this paper, and the paper provides in great detail relative and absolute differences (2020-2019) of top-down emissions (Figure 4 and Table 2) despite those limitations.

**It is true that satellites can sometimes provide limited quantitative information (like HCHO in higher latitudes or NO$_2$ in rural areas) or depreciate the model analysis (like northeastern Germany in March, and eastern Europe in May), but the motivation of this study is not to blindly advertise their utilization. Our major motivations stated in the introduction:**

*"The motivations of this study are to determine the capability of a regional model constrained by satellite HCHO and NO$_2$ columns to capture near-surface pollution, and if local ozone production rates are the driving factors for heightening ozone pollution during the 2020 lockdown. In other words, what chemical and physical processes are associated with the elevated*
*ozone? How representative are satellite observations at capturing surface air quality through an inversion context? Is meteorology the primary factor in shaping elevated ozone as suggested by Ordóñez et al. [2020]?"*

**This paper is an ozone study. The surface ozone levels are function of emissions, meteorology, transport, and etc. To perturb the emissions, one may cut mobile emissions by 50% (like what most covid-19 modeling studies have been done) or try to provide top-down estimate using real data (satellites). We chose the latter and as a result, we found a decent spatially-varying anomaly in NO2 in April. This helped us to achieve a decent anomaly in term of ozone (which is extremely hard to get from models; please take a look over Figure 2 in https://agupubs.onlinelibrary.wiley.com/doi/10.1029/2020JD034213, April 2020 O3 (the middle right over Europe); this paper is written by great scientists, but their model over the Europe is not as responsive as our model (and OBS) to the changes in emissions/meteorology). We broke down each individual physiochemical model process to understand why surface ozone is higher in 2020.**

The new figures S7-S11 with the comparisons with surface NO2 data are interesting but I wonder about their significance. There is an overall bias reduction (31%-45%, although the precise meaning of this range is not made clear) but, despite the TROPOMI NO2 constraint, a large systematic underestimation of modelled NO2 remains with respect to the data. Why is that? Could this be due to representativity issues? Or interferences in the NO2 measurements? Note that daily averages are strongly influenced by night-time chemistry. For a proper evaluation of TROPOMI-constrained model values, it would be preferable to sample the data as well as the

model in the early afternoon (say between 12 and 15 LT), since night-time chemistry is not well constrained by TROPOMI. In addition, it would be useful to summarize the comparison over the different subregions of Fig. S7-S11 in a table (including mean bias, mean absolute deviation and correlation coefficient). The authors state in their response "The absolute amount of CEDS is too low in many places" as explanation for the model underestimation. This is not a comparison of CEDS with NO2 data. There might be issues in the model (PBL mixing, chemistry) or in the data (interferences, representativity issues) which should be acknowledged first before holding the emissions as responsible.

**We actually found a very recent study done by Sun et al., 2021 ([https://acp.copernicus.org/preprints/acp-2021-268/](https://acp.copernicus.org/preprints/acp-2021-268/)) who found the same inventory to be significantly lower than other top-down estimates; See figure 11 in their paper. Yes, chemistry, PBL, and the NOz interference partly play a role, but they can't fully be responsible for such a large underestimation of NO2 in the model. We have extensively used the WRF-CMAQ in our previous studies and we have never observed anything even close to these large negative biases [Souri et al., 2016; Souri et al., 2017; Souri et al., 2018; Souri et al., 2020] except for emission offsets happening episodically in petrochemical industries. We strongly believe the emissions are the primary culprit.**

**We added more explanation and added the statistics in the supplementary for the selected regions. We need to focus only on daily-averaged data to minimize the impact of discounting for the diurnal mobile emissions due to the lack of the diurnal factors in the CEDS inventory.**

**A major philosophical question arises here: why should point measurements absolutely match to the grid data? Point is an element of space, but model (and satellite) grids (at best) represent the average values. This is a fundamental issue in our community. We just recently tackled this taboo (https://drive.google.com/file/d/1a7qc_YHU3zP_pp1U4GfomgV0e6AZv7T7/view?usp=sharing). Vigouroux et al. may be interested in this study as their statistics could have been largely impacted by the spatial representativeness.**

**We also forgot to mention in the previous review that we did not see such a large deviation of NO2/(NOy) factors portrayed by Lamsal et al., 2008, from 1.0 in our previous studies over Texas and CONUS. The NO2 correction factors were topmost 10-20% in September 2013 over Houston, which is far lower than 60-80%(!). Perhaps, the chemical mechanisms have been much improved since 2008. The coefficients may need to be readjusted with newer models. Moreover, the spatial representivity factor is hugely impacting Lamsal's results. $NO_2$ values are spatially heterogenous making the spatial representivity error massively large.**

**To account for the reviewer's comment:**

**We added Table S3 and S4.**

**We added:**

We observe an improvement in the statistics associated with simulated surface $NO_2$ using the posterior emissions compared to the surface measurements in many places around Europe with an exception to northeastern Germany where TROPOMI $NO_2$ observations deviates the model from the measurements (Figs S7, S8, S9 and 10; Tables S3, S4). The large underestimation of the model in terms of surface $NO_2$ concentrations is most likely due to the underestimation of CEDS inventory [e.g., Figure 11 in Sun et al., 2021]. However, it is worth noting that the disagreements between the model and the surface measurements do not solely reflect the uncertainty in the emissions. A major complication arises from the fact that point measurements represent concentrations locally, whereas the model grids ($15 \times 15$ $km^2$) are (at best) the average of infinitesimal points integrated over the grid space. Essentially, no one should expect that these quantities will completely line up, unless one transforms the point measurements to the grids (i.e., rasterization) by carefully modeling the spatial auto-correlation (or semivariograms) of the point data [Souri et al., 2021]. Additionally, there is uncertainty about the chemical mechanism used in the model. In particular, Souri et al. [2017] observed a large overestimation (~ factor 4) of daily-averaged total nitrate ($HNO_3 + NO_3^-$) in the CB05/AERO6 mechanism despite moderately reasonable nitrate ($NO_3^-$) simulations. This was attributed to a large overestimation of $N_2O_5$ hydrolysis rate [Bertram and Thornton, 2009] which is the primary loss pathway of $NO_x$ in low photochemically active regions [Shah et al., 2020]. The interferences from the $NO_z$ family on the surface measurements might be still present in springtime in midlatitudes (~10-30%) [Lamsal et al., 2008]. Last but not the least, the PBL parametrization controlling the level of vertical mixing rates has errors primarily due to soil moisture not being observationally constrained in the model [Huang et al., 2021].

The authors complain that they "do not understand why this reviewer is concerned about the prior emissions. This paper is not about validating CEDS (...) We could have used EDGAR emissions and reached similar results (...) [or] even used constant emission rates throughout Europe and induce the emission changes by TROPOMI". This is wrong. The striking similarity between prior and post NO2 columns (Fig S11-S14) indicates clearly that both TROPOMI and the prior determine the solution, and therefore the choice of the prior does matter a great deal. Even in areas with high AKs where the inversion is mostly driven by the observations, the changes in the patterns of the emissions induced by the inversion must be questioned: are those real or could they be related to issues in the model or in the observations? I'm not asking you to solve these issues but only to consider and discuss them in a more balanced way.

**Thanks for your comment. We both agree and disagree. If we set NOx emissions constant in the entire region, it is wrong to set the prior errors to 50%; it should be ~10000%. This will automatically simplify the problem to an iterative joint mass balance method. The final result may not be as good as considering the prior knowledge (the height of injection, rates, speciation with regards to VOCs, …), but the final estimate will be much better than a constant rate (i.e., AK~1 almost everywhere). From an optimization perspective, this is a successful case. But if we are aiming for the exact rates and to reduce the computational costs, it is obvious that we should have reasonable prior values. We already addressed these problems in the first comment.**

Regarding the HCHO inversion I still believe that the results of the inversion have very little value given the high differences between the model and TROPOMI. If they think that the emission differences 2020-2019 from the inversion are significant, please provide a quantitative estimation of the uncertainties on the retrieved emissions.

The authors insist that the high TROPOMI HCHO values in April in Northern and Eastern Europe are not too high, citing the negative bias over high-HCHO level sites reported by Vigouroux et al. 2020. The hypothesis that the negative bias is due to aerosol effects is what it is, a hypothesis. In April 2019 over Saint Petersburg (a megacity right in the big HCHO plume on Figure S13), TROPOMI is overestimated by about 35% based on comparisons by Vigouroux et al. Same thing over Sodankyla and Kiruna. Those direct measurements inside the HCHO hot spot are much more relevant than speculations about the possible role of biomass burning. Your manuscript should acknowledge that TROPOMI HCHO is very probably overestimated in that area, for reasons unknown. I do not dispute the fact that the direction of the emission increment in April 2019 goes in the right direction. Of course it does. But you do not need a sophisticated inversion system to infer that emissions were higher than the model prior in April 2019.

**We decided to move the VOC and HCHO part to the supplementary material as there are not independent data to verify the model/TROPOMI. We cannot remove this part entirely because the VOC emissions are constrained (especially in lower latitudes where the signal is strong). The results must be reproducible, therefore we need to inform readers about all modifications applied to the model including VOCs. We need to keep the results in the supplementary part for different reasons:**

i) **Ozone chemistry is a function of NOx and VOC emissions, even though TROPOMI HCHO (not only the inversion) didn't provide significant information on emissions in midlatitude/high latitude regions, we do see a good amount of information in lower latitudes (Mediterranean basin for example).**

ii) **The atmospheric lifetime of ozone can reach to several weeks. So it is important to constrain the relevant emissions in large areas as much as we can. This is more critical for April 2020 when the high pressure system over the central Europe has extended to lower latitudes implying that there are strongly regional background contributions.**

iii) **There are some changes on VOCs in central Europe in April 2020. The results won't be reproducible if these changes are not applied (especially if we consider the larger sensitivity of ozone production rates to VOCs in NOx-statured areas). The results must be reproducible, so the analysis on VOC/HCHO should be included somewhere.**

iv) **We do not polarize the results. Each pixel should be treated as an independent instrument with different uncertainty.**

**We briefly mentioned the results in abstract/conclusion:**

The observational constraint on VOC emissions is found to be generally weak except for lower latitudes.

Fourth, we observed a weak observational constraint on VOC emissions from TROPOMI HCHO except for lower latitudes.

We added in the supplement:
It is worth noting that the TROPOMI bias-correction factors used here based on Vigouroux et al. [2020] are not necessarily correct over this area possibly due to snow cover, the profile shapes, or non-linear aerosol impacts on AMFs (see Figure5 in Vigouroux et al. [2020]).

**The title has changed to weigh down the VOC part:** Unraveling Pathways of Elevated Ozone Induced by the 2020 Lockdown in Europe by an Observationally Constrained Regional Model using TROPOMI

The authors did not address my comment that the bias correction might lead to different corrections being applied in 2019 and 2020, thereby creating artificial patterns in the differences between the two years.

**We added that (above):** ii) the statistics used for the TROPOMII bias-correction may not always hold true, since each individual pixel can deviate from the norm of the reported biases

**FYI: NO2 results without the bias correction; please note that those positive values over C and B are still present. With these results, the underrepresentation of the model in terms of the NO2 reduction in April would have worsened.**

[Figure]

**And for HCHO without the bias correction:**

[Figure]

**We don't believe the TROPOMI bias-correction has impacted the conclusions drawn from our analysis. It worked in the favor of capturing NO2/NOx in April. Again, over the biomass burning area, the Kalman gain is so low that it doesn't really matter if the correction factors or +30% or -30%. Also, we did not conjure up some numbers out of nowhere. The correction factors are based the recent validation studies.**

I strongly recommend to drop that part on HCHO and the VOCs which, despite relying on a sophisticated inversion scheme, cannot do better than a simple visual inspection of model results and observations. For the NOx part, I recommend strongly to drop Table 2 and reword many parts (including discussion, abstract and conclusions) in order to convey the known limitations and uncertainties of inverse modeling (as acknowledged by the authors, see above).

**Mentioned above. The abstract is already too long to include them.**

Minor comments:

- Thanks for the clarification on the AMF and the profile shapes. How does that relate to the averaging kernels (in the definition of e.g. Eskes and Boersma 2003, www.atmos-chem-phys.org/acp/3/1285/) used by other groups to derive total columns from model profiles?

**To be able to use "averaging kernel" variables in TROPOMI data, we need to multiply those values by AMF. This quantity will be identical to box AMF (or sometimes wrongly called scattering weights in some literature; box AMF = AMFg*SW; final AMF = box AMF*SF). See eq 7.2 in https://sentinels.copernicus.eu/documents/247904/2476257/Sentinel-5P-ATBD-HCHO-TROPOMI.pdf/db71e36a-8507-46b5-a7cc-9d67e7c53f70?t=1625507823781**

- Thanks for the clarification on the observation error. Note that the TROPOMI precision estimation might actually contain non-random parts. Your inversion system does not account for model errors. Can some crude estimate be provided for those? How could their omission impact the results?

**Propagating the model error parameters (such as winds, PBL, clouds and etc.) to the final estimation requires a fully explicit calculation of Jacobians (here linking columns to that specific parameter, and finally columns to emissions) which is computationally burdensome and sometimes not possible (do we have the 4D error of wind vectors or cloud microphysics for example?); from Rodgers, 2000:**

$$\mathbf{S}_f = \mathbf{G}_y \mathbf{K}_b \mathbf{S}_b \mathbf{K}_b^T \mathbf{G}_y^T \qquad\qquad (3.18)$$

**G is the column-emission relationship, Kb describes the column-model parameter relationship. Sb is the covariance matrix of the model error parameter.**

**That's an oversight which we had touched upon in Souri et al., [2020]. So, we essentially tend to under-predict the errors in the top-down estimation because of treating the model parameters as perfect. An alternative way to quantify these errors is to run ensemble of models with different parametrizations/initializations/reanalysis data. That would beautifully provide a set of solutions for the estimates and the resultant anomaly maps (like ozone), which in turn, it would help us with detecting outliers. We have the scripts ready to do that (see https://agupubs.onlinelibrary.wiley.com/doi/abs/10.1029/2019JD031941), and our first attempt for this study was to follow that approach with different emissions/parametrization (https://twitter.com/AmirHSouri1/status/1271485748915519491?s=20). But we just could not afford the computational cost at some point, so we had to switch our inversion to what we did in Souri et al., 2020.**

**To account for the reviewer comments:**

"An important caveat with this inversion system is that we do not take the model parameter error (such as errors in chemistry, cloud microphysics, and PBL) into account. To properly estimate the forward model parameter errors, one needs to calculate the sensitivity matrix of the columns to the model parameters combined with the sensitivity matrix of the columns to the emissions (*K*) [Rodgers, 2000]. The former calculation is computationally expensive. Moreover, the spatiotemporal varying model parameter errors may not be known in detail. The consequence of disregarding the model parameter errors is the overconfidence in the top-down estimates (i.e., overestimations of AKs)."

- The HCHO TROPOMI product provides random and systematic error estimates, why not using those? The 4% seems too low considering the large variability among the different sites in Vigouroux et al.

We addressed in the supplement: We assume the constant term of errors ($e_{const}$) to be equal to 4% of HCHO total columns based on Vigouroux et al. [2020]. The precision error ($e_{precision}$) is populated with the column uncertainty variable provided with the data.

- What are the implications of neglecting diurnal variations of anthropogenic emissions?

**For ozone? It depends on the underlying chemical conditions. Our former studies over industrial areas in Texas in summertime revealed that it was sometimes important (by sometimes we mean on very stagnant and photochemically active day) to have the right timing of mobile emissions if we want to replicate the pick of ozone in NOx-sensitive areas. We should recognized the fact that not all anthropogenic sectors have diurnal cycles. But we are unsure if this matters when it comes to the springtime ozone over Europe. The chemical conditions are majorly NOx-saturated, and O3 titration through NOx is prevalent. So having a fixed emission rate over a diurnal variability would decrease/increase the titration during daytime/nighttime making ozone slightly higher in daytime, and lower in nighttime. We are unsure if this is critical for MDA8 on the monthly basis in low photochemically active areas. It could have been problematic if we had studied the changes in diurnal shape of ozone due to covid-19.**

- Table 2: please provide uncertainties if you provide numbers which you think could be used by regulatory agencies. There is ample evidence that these numbers should be taken with great caution. Or delete this table if you cannot estimate the uncertainty.

**Using eq.4, we can provide errors for these estimates, but those are theoretical. To provide an exact error, we will need eddy covariance flux measurements. Therefore we removed the table.**

- Seasonality of top-down VOC emissions in 2019: I reiterate my comment. The retrieved patterns indicate primarily anthropogenic emissions, with hot spots in Ruhr and Rhine Valleys, Southern Holland, London, etc. Those are not biogenic emission hotspots. The inversion system is simply unable to bring useful information on the emissions, except that the total VOC emissions were much higher than the prior (and than their 2020 counterparts) in Northern Europe (which is of course very obvious from TROPOMI).

**The inversion system used for this study is based on a work presented in Souri et al., 2020. This system significantly improved HCHO over east Asia using the same model (WRF-CMAQ):**

[Figure]

The inversion does not work in *mysterious ways*. The reason that we do not see such an agreement in this study is primarily because of large errors of TROPOMI in less photochemically active areas over Europe (please compare these columns to those in Europe) or the inability of model to provide sensitive Jacobians over certain areas (due to clouds, chemistry, and etc.). Another important caveat with comparing two datasets (here post vs TROPOMI) is that we must consider their variance. Pixels with higher uncertainty (larger variance) located over higher latitudes have a smaller weights in the comparisons.

[Figure]

As for the enhancements of TROPOMI HCHO (and VOCs) in some urban areas in 2020 with respect to 2019, we would need speciated VOC measurements to understand why. HCHO is too crude to determine the main reason. One hypothesis is that the number of VOC compounds in CEDS is too limited such that the model had to increase the anthropogenic emissions to compensate for. Another reason could be due to uncertainty in the yield of HCHO in CB06 mechanism. We added in the supplementary:

"However, the reason behind of the enhancement of VOCs over several urban areas such Paris and Po Valley is not fully understood. This can be caused by the errors in the chemical mechanism or the limited VOC compounds provided by the CEDS emission inventory."

**The reviewer provided very constructive and remarkable comments, which we have taken to heart; as a result, we believe our study has become stronger and are hoping this reviewer will find the manuscript merit publication in ACP.**